# Warming assessment of the bottom-up Paris Agreement emissions pledges

Yann Robiou du Pont [1] & Malte Meinshausen [1,2]

Under the bottom-up architecture of the Paris Agreement, countries pledge Nationally Determined Contributions (NDCs). Current NDCs individually align, at best, with divergent concepts of equity and are collectively inconsistent with the Paris Agreement. We show that the global 2030-emissions of NDCs match the sum of each country adopting the least-stringent of five effort-sharing allocations of a well-below 2 °C-scenario. Extending such a self-interested bottom-up aggregation of equity might lead to a median 2100-warming of 2.3 °C. Tightening the warming goal of each country's effort-sharing approach to aspirational levels of 1.1 °C and 1.3 °C could achieve the 1.5 °C and well-below 2 °C-thresholds, respectively. This new hybrid allocation reconciles the bottom-up nature of the Paris Agreement with its top-down warming thresholds and provides a temperature metric to assess NDCs. When taken as benchmark by other countries, the NDCs of India, the EU, the USA and China lead to 2.6 °C, 3.2 °C, 4 °C and over 5.1 °C warmings, respectively.

[1] Australian-German Climate & Energy College, University of Melbourne, Parkville, 3010 Victoria, Australia. [2] Potsdam Institute for Climate Impact Research (PIK), Telegraphenberg, 14412 Potsdam, Germany. Correspondence and requests for materials should be addressed to Y.R.d.P. (email: yann.rdp@climate-energy-college.org)

Since the adoption of the United Nations Framework on Climate Change Convention (UNFCCC)[1] and its objective to stabilize greenhouse gas (GHG) concentrations to avoid dangerous global warming, most countries have committed to limiting GHG emissions through domestic measures or support of mitigation action abroad. Informed by literature on effort-sharing approaches, the international community has long discussed the operationalization of equity following the UNFCCC principle of Common But Differentiated Responsibilities and Respective Capabilities (CBDR-RC) to drive national emissions allocations[2,3]. The failure to agree on a top–down mechanism to derive binding national emissions targets for all countries led to a bottom-up situation where countries should pledge NDCs of highest possible ambition[4,5]. While the quest for a common understanding of what is a fair effort-sharing continues, rapidly falling technology costs of renewables and increasing mitigation co-benefits shift the attention away from effort-sharing considerations[6]. However, current bottom-up NDCs do not add up to a global ambition consistent with the joint temperature goals[4,7–9]. A 5-year stocktake requires all countries to pledge enhanced actions and support[4].

The quantification of national emissions levels consistent with both Paris Agreement's mitigation and equity goals relies on contentious interpretations of distributive justice[2,8,10]. Scientists, non-governmental organizations and government experts have suggested multiple effort-sharing approaches to derive equitable national emissions allocations[2,8,11–22]. While not all countries use indicators that favour their equity argument in their communication[10], a common definition of equity is unlikely to be adopted since countries generally tend to support interpretations of distributive justice that best serve their self-interest and justify their negotiating positions[23–26]. Developed countries who committed to take the lead in reducing emissions and mobilizing finance for developing countries[4] often submitted NDCs that do not match the concepts of equity that they publicly supported[27] and leave the Green Climate Fund poorly funded[28]. Their NDCs often imply a status-quo in terms of global emissions shares[27], while most of the very ambitious NDCs are from smaller developing countries (https://paris-equity-check.org/).

The UNFCCC does not specify whether its principles and the CBDR-RC refer to distinct principles or to a single operationalization of equity[29]. A way to reconcile this ambiguity is to combine multiple dimensions of equity using weighting factors[12,14–17,22] and per-capita income thresholds[14,16] in a single effort-sharing approach applicable to all countries. Alternatively, effort-sharing approaches can be combined in a differentiated manner where countries follow different equity principles. A recent study allocated emissions to each country using the least-stringent of two equity allocations[11]. The global level of ambition of each equity allocation was then set by a diversity-aware leader so that the sum of all countries' allocations matches 2 °C-consistent levels[11]. Under that methodology, countries follow different equity approaches that are applied under different warming thresholds, which may be considered unfair by Parties to the UNFCCC.

In the present study, we use a single aspirational warming threshold that is lowered consistently until the sum of all bottom-up emissions allocations, where each country follows the least-stringent effort-sharing approach[11], aligns with 2 °C-consistent levels. Countries are thereby assumed to follow different effort-sharing approaches applied to a common global virtual warming threshold. Ultimately, this hybrid approach follows a bottom-up combination of equity allocations consistent with a common top–down warming threshold, which arguably reflects the hybrid nature of the Paris Agreement[10,30].

The hybrid combination of equity approaches does not constitute an equitable operationalization of the CBDR-RC principle where all countries seek to maximize absolute gain[31] by agreeing on a common approach of equity. Rather, it reflects national preferences for relative gain[32]—i.e., a country's inclination to measure the fairness of its contribution to the global mitigation effort by looking at other countries' efforts—rather than for domestic indicators alone. Despite claims that discussions of justice are irrelevant or dangerous in a post-Paris world, equity is fundamental for climate policy research[33,34] and scientific analyses on equitable burden-sharing can be influential on the UNFCCC processes[5]. However, the absence of agreement on an unanimous operationalization of the CBDR-RC should not be used as an excuse for inaction[3] and should not leave the international community without a metric reflective of current agreements to assess the ratcheting-up process. The multiplicity of equity concepts results in a wide range of emissions allocations for countries and regions[35] that is sometimes used as an uncertainty range by non-experts. In a recent climate case, the District Court of The Hague ruled[36,37] that the Dutch government has to reduce 2020 emissions to at least the least-ambitious end of the range recommended by the IPCC-AR4 for the Annex I country group based on multiple equity allocations from 16 studies[38]. The court did not pick an approach of equity and ruled for the minimum effort consistent with international treaties in light of commonly reviewed science. While the multiplication of climate litigations cases against governments[39] (http://climatecasechart. com/) can contribute to the ratcheting-up process, systematic court decisions that governments must follow the least-ambitious end of an equity range would be insufficient to achieve the Paris Agreement. As a first step, this paper models such a bottom-up situation where each country follows the least-ambitious of five effort-sharing approaches representative of the quantified IPCC categories[35]. As a second step, it models the hybrid approach consistent with the current compromise where each country chooses an equitable effort-sharing approach to determine its effort but cannot directly use that approach to influence other countries' effort.

Overall, this study presents an operationalization of the current agreement to disagree on equity concepts to achieve a common temperature goal. The hybrid approach with its bottom-up combination of equity concepts reflects the pledge-and-review architecture of the Paris Agreement and provides a metric for the ratchetting-up process. The results of this study inform on the adequacy of the emissions targets contained in current NDCs with the Paris Agreement.

## Results

**Projecting a self-interested approach of effort-sharing.** The first step of this study is to model a self-interested bottom-up allocation of emissions from Integrated Assessment Models' (IAM) scenarios reflective of the Paris Agreement goals (excluding emissions from land-use and international shipping and aviation, see Methods). We derive this bottom-up allocation of a 2 °C-scenario—with a likely ( > 66%) chance to stay below 2 °C until 2100 (RCP2.6, ref[40])—and a 1.5 °C-scenario—with a median ( > 50% likelihood) 2100-warming below 1.5 °C (Methods)—using five effort-sharing approaches representative of the five categories quantified in the IPCC fifth assessment report[8,27,35]. These five categories include notions of capability to pay (CAP approach), equality with the dynamic Equal Per Capita (EPC) approach, responsibility-capability-need with the Greenhouse Development Rights (GDR), historical responsibility with the Equal Cumulative Per Capita (CPC) and national circumstances regarding current emissions levels with the

grandfathering approach (also named Constant Emissions Ratio or CER). The grandfathering approach, a status-quo approach that allocates equal emissions mitigation rates to all countries, is considered unfair[12,41] and not openly supported by any country but implicitly matches many developed countries' targets[8], which they often declare as fair[42]. Under the complete bottom-up modelling setup (Supplementary Table 1), each country follows the approach, from the complete set of five effort-sharing approaches, that yields the greatest cumulative emissions over the 2010–2100 period (Fig. 1a, Supplementary Fig. 1). The complete bottom-up attribution of the least-stringent effort-sharing allocation to each country under the 2 °C-scenario (Fig. 2a, Supplementary Table 2) aims at representing the currently discordant equity debate[11,23,25,43].

Coincidentally, the trajectories of these complete bottom-up 1.5 °C-scenario and 2 °C-scenario align with the aggregated high and average (average of high and low) NDCs assessments[44], respectively (Fig. 2b). In other words, the global pledged effort matches that of a world where each country follows the least-stringent vision of effort-sharing for their circumstance. Extending such a self-interested situation throughout the century would result in median-warmings of 2.0 °C and 2.3 °C in 2100, respectively higher than the 1.5 °C and well-below 2 °C thresholds. For reference, the well-below 2 °C-scenario of this study results in 1.7 °C median 2100-warming[45]. We then increase the ambition of the complete bottom-up allocation to be consistent with a top-down temperature threshold. We find that the

complete bottom-up allocations consistent with more stringent aspirational global scenarios that limit warming to 1.1 °C and 1.3 °C result in global emissions consistent with 1.5 °C and well-below 2 °C, respectively (Fig. 2, Methods, Supplementary Figs. 1, 2). Under such a self-interested situation, each country should use an aspirational target of 1.3 °C when calculating its fair share to effectively stay well-below 2 °C, and 1.1 °C to effectively return to 1.5 °C. The resulting temperature gaps between the virtual targets and effective warmings reflect the necessary strengthening of global temperature aspirations to compensate for the disagreement on effort-sharing (Fig. 2b). The national emissions trajectories resulting from the hybrid allocation add up to the targeted IAM scenario (Fig. 1b) and can be met through combination of domestic mitigation, internationally traded mitigation outcomes[4] and financial contributions. As a result, the technical feasibility of the hybrid approach follows that of the underlying IAM scenario.

**A hybrid approach to ratchet-up self-interested ambition**. The second step of this study derives an equity-based metric, reflective of the CBDR-RC principle[1,3,4], to assess the ambition of current and future NDCs under the Paris Agreement mitigation goals. We quantify the CBDR-RC hybrid approach to align with countries' preferences for equity approaches that are based on principles of equality (EPC), responsibility (CPC) and capability (CAP)[3]. Each country is attributed the equity approach with the

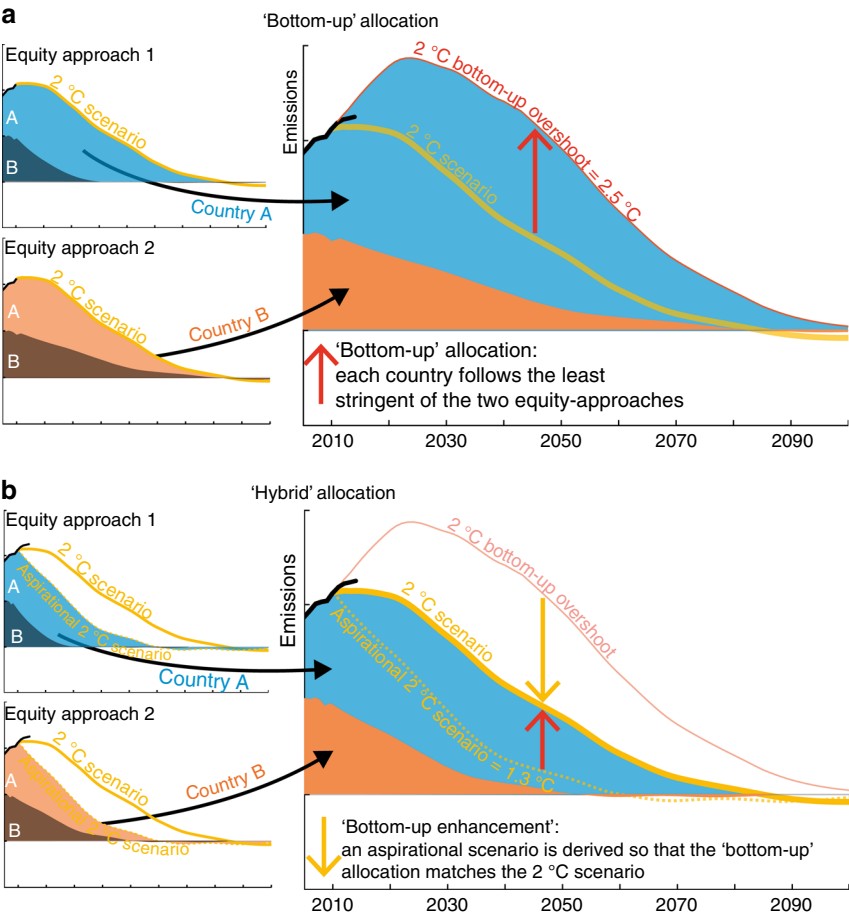

**Fig. 1** Schematic description of the bottom-up and hybrid allocations of global emissions scenarios. **a** Under the bottom-up allocation, each country adopts the least-stringent equity approach. As a result of this self-interested allocation, the targeted 2 °C scenario is overshot. **b** An aspirational scenario is created so that its overshoot under the bottom-up allocation matches the originally targeted 2 °C scenario. Each country individually adopts the least-stringent equity approach of the aspirational scenario in order to collectively achieve the originally targeted 2 °C scenario

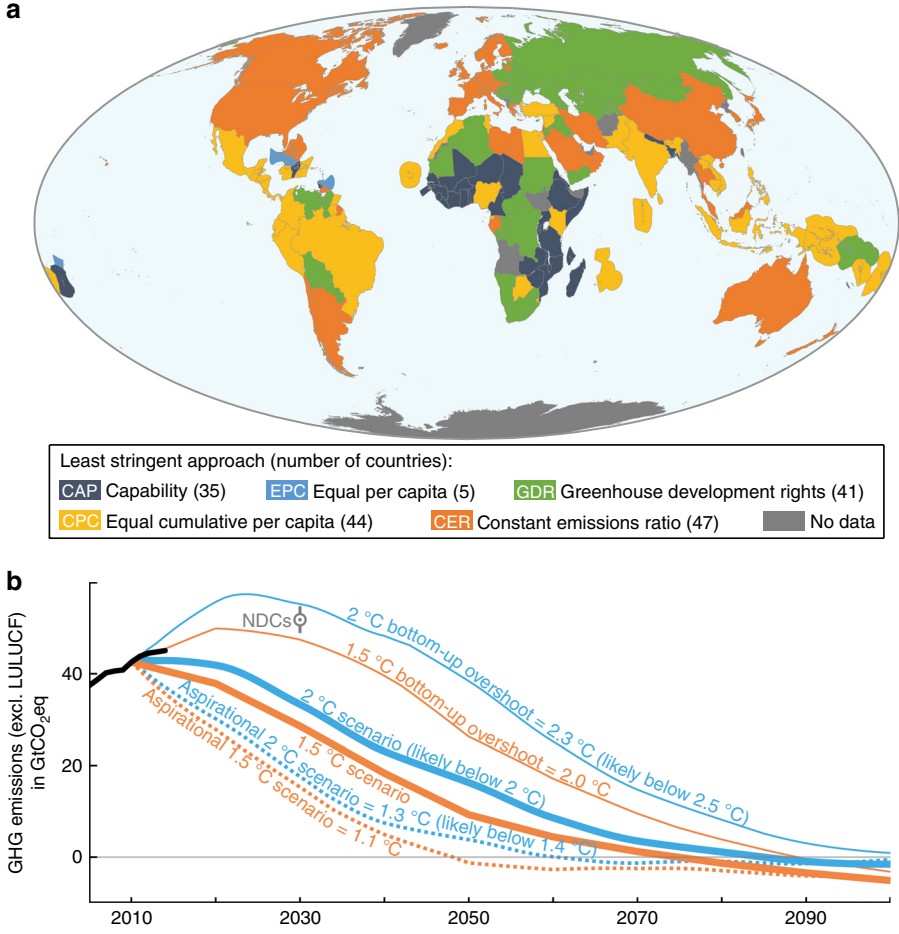

**Fig. 2** Complete bottom-up and complete hybrid allocations of global emissions scenarios consistent with the Paris Agreement. **a** Least-stringent of five effort-sharing approaches, that with the highest cumulative emissions between 2010 and 2100, for each country under the complete bottom-up allocation of 2 °C-scenario. **b** Scenarios towards well-below 2 °C and 1.5 °C (excluding land-use and bunker emissions) shown with their corresponding complete bottom-up allocation and aspirational scenarios and the NDC assessment[44] range (grey)

least-stringent 2030-emissions, excluding the grandfathering and GDR approaches (Methods, Supplementary Fig. 3, Supplementary Table 3). Current national emissions ratios, at the root of the status-quo grandfathering approach[8,12,41], also influence near-term allocations of any continuous allocations, reflecting some of the national circumstances mentioned in the Paris Agreement[4]. The GDR approach is based on principles of capacity and responsibility[16] that are covered in the CAP and CPC approaches, respectively. The GDR approach relies on hypothetical projections of Gini indices and business-as-usual emissions (here downscaled from RCP8.5, see Methods) that can lead to large variations in emissions allowances. These variations can be more determined by input assumptions on counterfactual baselines than by the effort-sharing principles themselves (https://paris-equity-check.org/). We therefore present a combination of the EPC, CPC and CAP approaches in a CBDR-RC hybrid setup that enables the derivation of an NDC warming assessment tool applicable to all, with self-differentiation. We also provide in the Supplementary Information results under hybrid setups that include the GDR, which represents a right to development, and that use the five effort-sharing approaches (Methods, Supplementary Fig. 4, 5 and 6).

At the national level, the current NDCs of the G8 countries (including the 28 EU countries) and China imply higher 2030-emissions than even the most favourable of the three equity approaches applied to the 2 °C-scenario. In other words, their

NDCs are less ambitious than the CBDR-RC bottom-up allocation of the 2 °C-scenario, unlike the other economies as a group (Fig. 3a).

The objective is here to combine multiple approaches of equity to achieve the Paris Agreement long-term mitigation goals. A method commonly used is to average or blend[22] multiple equity approaches. This method postulates a joint agreement on a common yardstick of fairness that does not reflect individual countries' views on equity. The hybrid approach modelled here avoids the use of subjective weighting factors across multiple equity approaches. The national allocations of the CBDR-RC hybrid approach happens to yield similar results to the average of the three equity approaches' allocations[8] for G8 countries and China (Fig. 3b). Compared with using an average, the CBDR-RC hybrid allocations are greater for most Least Developed Countries (LDC), but lower by 19 percentage-points of 2010-emissions for Brazil and by 16 percentage-points for India. The variability across national results does not show strong regional trends. Achieving allocations under the CBDR-RC hybrid setup with a likely chance to stay below 2 °C implies raising the NDC's (using the average assessment of ref. [44]) ambition by around 38 percentage-points for the USA, 29 for the EU28 and 89 for China (Fig. 3c). Aiming at 1.5 °C, rather than 2 °C under the CBDR-RC hybrid approach requires one additional percentage-points from the G8 countries and China, ten for the other economies altogether, and more for LDC (Fig. 3d).

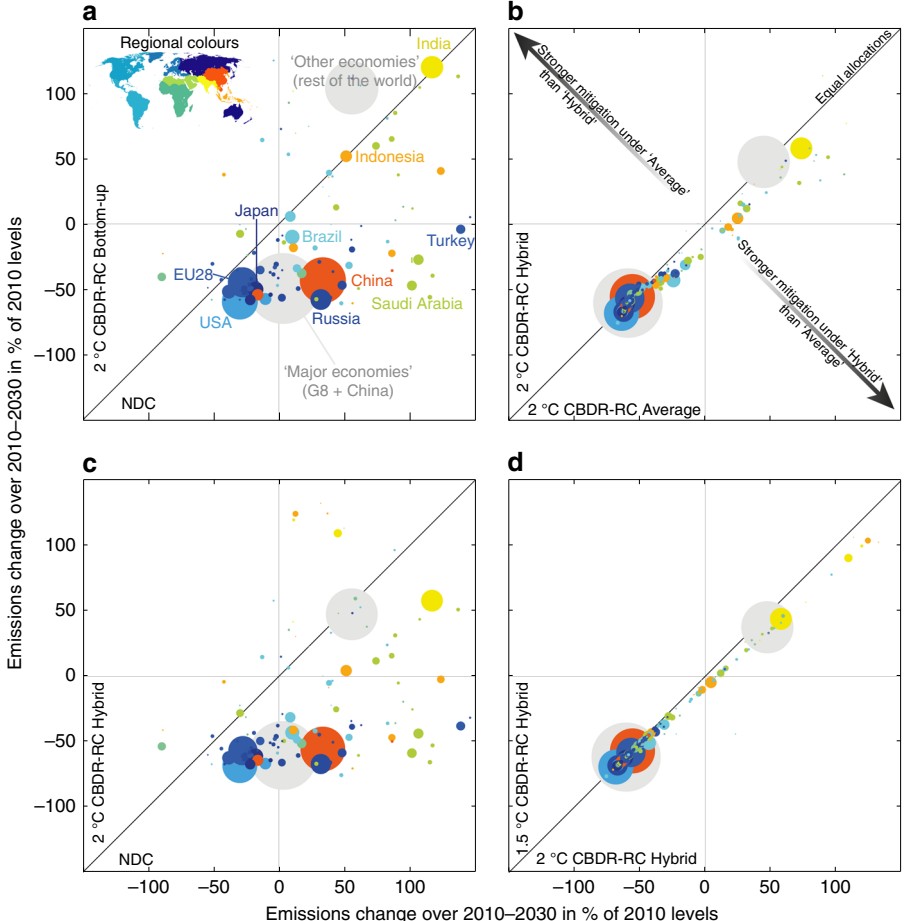

**Fig. 3** Comparison of emissions changes over 2010-2030 across diverse combinations of equity and with NDCs. The CBDR-RC hybrid setup combines three equity principles: equality, responsibility and capability. **a** Comparison of the CBDR-RC bottom-up allocation of the 2 °C-scenario and countries NDCs. **b** Comparison of the CBDR-RC hybrid allocation and the average of the three equity allocations, under the 2 °C-scenario. **c** Comparison of the CBDR-RC hybrid allocation of the 2 °C-scenario and countries' NDCs. **d** Comparison of the CBDR-RC hybrid allocation of the 2 °C-scenario and 1.5 °C-scenario. Discs' sizes are proportional to 2010-emissions level. Colours indicate countries' world regions, and the major economies (G8 + China, larger disk) and the other economies (all countries other than G8 countries and China, smaller disk) are shown in grey. The NDC evaluation follows the average evaluation of ref. [44]

**Assessment of NDCs on a global warming metric**. To relate the ambition of national climate pledges to levels of global warming, we compare countries' NDCs to the CBDR-RC hybrid allocations of global scenarios with median 2100-warmings ranging from 1.2 °C to 5.1 °C (Methods, Supplementary Fig. 7, 8, Supplementary Table 4). We find that the NDCs of Canada, China and Russia are less ambitious than their CBDR-RC hybrid allocations even under the least ambitious global emissions scenario available, with 5.1 °C of warming in 2100 (Fig. 4). While the NDC of China appears very unambitious, and if recent analysis is correct to suggest a continues decline of Chinese emissions[46], China would then be peaking emissions well ahead 2030 and could significantly ratchet-up its NDC. India's current policies also appear on track to outperform its NDC[47]. When taken as benchmark by other countries, the NDCs of India, the EU, Brazil, the USA, Japan and China lead to warmings of 2.6 °C, 3.2 °C, 3.7 °C, 4 °C, 4.3 °C and over 5.1 °C, respectively. The aggregated emissions of other economies are aligned with a 1.7 °C median-warming. A think-tank report for a subset of 32 countries (http://www.climateactiontracker.org/) finds similar results for the NDCs of USA, Russia and the Philippines, higher warming assessments for Ethiopia and Indonesia, and lower warming assessments for Australia, Brazil, Canada, China, Japan, India and

the EU. The differences are largely due to the different methodology that simply assesses the compatibility of a country's effort with the 2 °C and 1.5 °C thresholds based on the number of effort-sharing approaches that its NDC aligns with http://www.climateactiontracker.org/. However, the progresses of Brazil on deforestation are not accounted in the present study as land-use emissions are excluded. Warming assessments and emissions trajectories until 2100 under the CBDR-RC hybrid approach applied to the 1.5 °C-scenario and 2 °C-scenario for all available countries are in Supplementary Data 1 and can be visualized at: www.paris-equity-check.org/warming-check.

**Discussion**

Because of the range of modelling assumptions in IAMs, multiple global scenarios with similar 2030-emissions values feature a range of 2100-warmings. Combining this scenario-related uncertainty on the link between global 2030-emissions and 2100-warming with that of the NDC assessment, we obtain a range of possible warming assessments for each NDC under the CBDR-RC hybrid approach (Fig. 4a, Supplementary Fig. 9, 10). Here, we use a second-degree fit to convey the relationship between the 2030-emissions allocations under the CBDR-RC

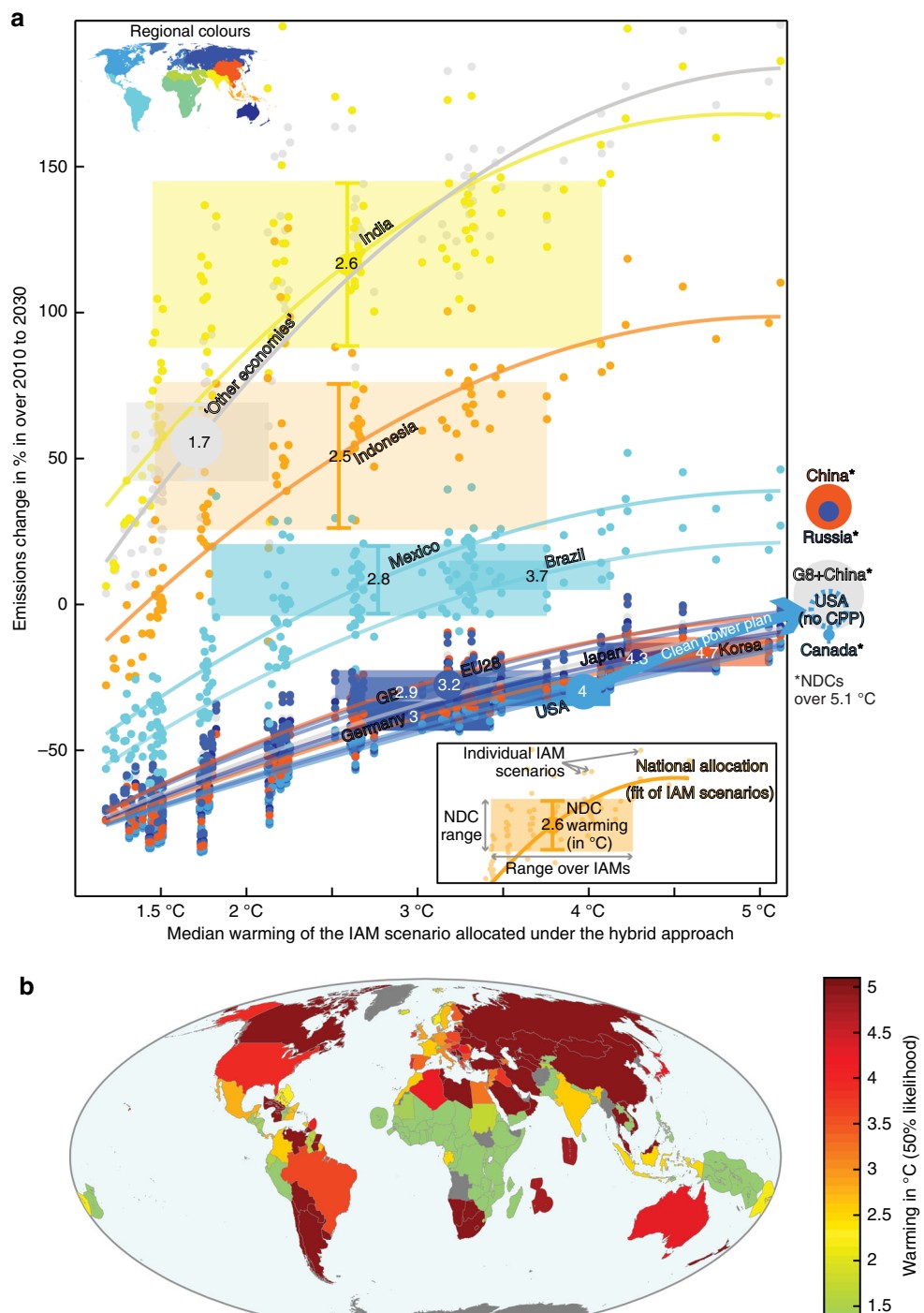

**Fig. 4** Global warming responses under the CBDR-RC hybrid approach following NDCs' ambitions. **a** For the top fifteen 2010-emitters, 2030-emissions as a function of median 2100-warming above pre-industrial levels under the CBDR-RC hybrid approach (coloured lines). The CBDR-RC hybrid setup combines three equity concepts: equality, responsibility and capability. Discs indicate the average NDC assessment and their sizes are proportional to 2010-emissions. Vertical ranges (coloured rectangles) indicate NDCs' evaluation ranges[44]. The horizontal uncertainty ranges (coloured rectangles) over the warmings of the IAM scenarios (coloured dots) that lie within the vertical NDC assessment range. Colours indicate countries' world regions (see map inset). **b** Global warming assessment (50% likelihood, compared with pre-industrial levels) of average NDC ambitions for 169 countries as calculated in panel **a** (maps with high and low NDC quantifications in Supplementary Figs. 9 and 10). The assessment ranges from 1.2 °C to 5.1 °C, NDCs outside this range are not differentiated. Small island developing states are represented by their maritime zones

hybrid approach and the 2100-warming of the corresponding global scenario (Methods). Choosing alternative scenarios sub-sets, or alternative representations of the scenario-induced uncertainty range would therefore affect the NDC warming

assessments but would not substantially change countries' NDC warming ordering.

Under the hybrid combination, the equity approach followed by a country influences the amount of emissions available for

other countries. Results therefore strongly depend on the set of effort-sharing approaches included in hybrid approach modelling. Using five approaches (complete hybrid setup), representative of the five effort-sharing categories quantified in the IPCC-AR5, results in lower temperature assessments of developed countries' NDCs and consequently in higher temperature assessments of other countries' NDCs (Methods and Supplementary Fig. 5). The inclusion of the GDR favours mostly Eastern European countries, Australia and South Africa, partly due to the influence of relatively high business-as-usual (BaU) emissions (Methods and Supplementary Fig. 6). Additionally, an indirect source of sensitivity arises from the sensitivity of the underlying effort-sharing approaches to their input data and parameters[48,49]. While this sensitivity is not linked to the hybrid combination of effort-sharing approaches, the warming assessments and ranking of NDCs' ambitions can be affected by political choices, such as the choice of period to account for historical emissions, convergence periods and technical assumptions regarding projections of population, GDP and BaU emissions (Methods).

The UNFCCC and Paris Agreement do not indicate how to operationalize the CBDR-RC and countries supposedly build their NDCs based on their own understanding of fairness, often self-interested[23–26]. As no single definition of fairness emerges from current NDCs[10], we quantify a new combination of equity concepts that reconciles the bottom-up pledge and review architecture of the Paris Agreement with its top-down mitigation goals. The resulting metric provides a warming assessment of countries' NDCs under the current regime and can inform the Talanoa dialogue and ratchetting-up process without hypothesizing an international agreement on a single approach of equity. This hybrid combination of countries' least-stringent equity approaches is also relevant to climate cases where the court only rules for the least-ambitious end of an equity-based range[50]. We find that most of the Least Developed Countries have NDCs consistent with the Paris Agreement goals. However, the NDCs of most developing countries appear insufficient, as those of developed countries who yet agreed to take the lead in reducing emissions and mobilizing finance to support mitigation in developing countries[4]. The hybrid approach should not be interpreted as an endorsement of moral subjectivism, and only a commonly accepted operationalization of equity would result in a fair and enduring mitigation[33]. The hybrid approach serves as an additional benchmark, reflective of the current Paris Agreement architecture, to assess whether NDCs are indeed fair and ambitious under the ratcheting-up process and avoid the most unfair outcome: unmitigated climate change.

## Methods

**Global scenario selection**. The 2 °C-scenario is RCP2.6 (Ref.[51]), the only of the four IPCC-AR5 Representative Concentration Pathways that offers a likely chance of limiting global warming to 2 °C (and results in a 1.7 °C median-warming at the end of the century[45]). The 1.5 °C-scenario in this study is the average of the 39 scenarios selected in Ref.[8] to have both net-zero GHG emissions before 2100, including emissions from Land Use, Land-Use Change and Forestry (LULUCF) and international shipping and aviation, and results in a median-warming below 1.5 °C in 2100. Warmings are expressed in comparison with pre-industrial levels. Two scenarios are from the IPCC-AR5 database (hosted at the International Institute for Applied Systems Analysis and available at: https://tntcat.iiasa.ac.at/AR5DB) complemented by 37 scenarios of refs[52–54]. These scenarios from IAMs represent a commonly used framework to discuss global mitigation under various Shared Socioeconomic Pathways (SSP) that model possible futures with different equity settings[55]. However, many technologic assumptions used in these scenarios can adversely impact vulnerable populations, depending on their implementation (for example land-based mitigation to achieve negative emissions[56]). The global emissions scenarios used to derive the range of 2030-allocations under the hybrid approach are from the SSP database (85 emissions scenarios with temperature assessment, hosted at the International Institute for Applied Systems Analysis and available at: https://tntcat.iiasa.ac.at/SspDb). The 2100-warming median assessments of these SSP-scenarios range from 1.7 °C to 5.1 °C. These are complemented by lower emissions scenarios

from ref[54] (36 emissions scenarios) whose 2100-warming median assessment ranges from 1.2 °C to 1.5 °C.

The relationship between national 2030-emissions levels and the 2100-temperature responses presented in Fig. 4 is derived from a representative sub-selection of global emissions scenarios. We standardize the data across both dimensions (2030-emissions, excluding LULUCF and bunkers and 2100-warming) and derive the third-degree polynomial fit (Supplementary Fig. 7). Using a second-degree polynomial fit would result in a plateau where high global warming hardly depends on 2030-emissions levels. We then select a subset of scenarios with the least standardized distance to the fit, starting at the lowest 2100-warming and every 0.5 °C (nine scenarios, Supplementary Table 4, Supplementary Figs. 7, 8). The USA's policy projection for 2030 is 6.74 GtCO2eq without the Clean Power Plan taken from http://www.climateactiontracker.org/.

**Global scenario preparation**. We used and extended the Potsdam Real-time Integrated Model for the probabilistic Assessment of emission Paths[19,57] (PRIMAP) to model allocations approaches. The database contains population, GDP and GHG emissions historical and projected data from composite sources as detailed in ref[27].

The aggregation of Kyoto–GHG emissions follows the SAR GWP-100 (Global Warming Potential for a 100-year time horizon), consistently with the reporting under UNFCCC (http://unfccc.int/ghg_data/items/3825.php).

The national emissions allocations derived in this study do not cover the LULUCF sector. Emissions from the LULUCF and from international shipping and aviation are removed from the global scenarios before allocating their emissions across countries using the methods and data indicated in ref[8].

For RCP2.6 and the 85 SSP scenarios, we subtracted $CO_2$ emissions from LULUCF. For the 36 scenarios of ref[54]. (including the 1.5 °C-scenario) where no specific LULUCF emissions are available, we subtracted the $CO_2$ emissions that do not come from fossil fuels combustion.

The historical emissions of Fig. 2b are from PRIMAP[19,57] until 2010 and follow the growth rates of ref[58] until 2014.

**Hybrid allocation**. We name complete bottom-up allocation of a global scenario the allocation to each country of the least-stringent of the scenarios calculated under the CAP, EPC, CPC, GDR and CER (grandfathering) approaches. The modelling and parametrization of these five approaches follows that of ref[8,27] (and their Supplementary Information) with the same limitations regarding the data missing for 27 countries and territories. Similar modelling to the EPC is also named per-capita convergence[11], equity[12] or similar. The EPC dynamically shares the emissions of the global scenario across countries based on their projected population trajectories, and thus it does not result in equal cumulative per capita emissions (i.e. equal cumulative emissions over cumulative populations). Comparing two countries with equal given cumulative population, a country with increasing shares of the global population will have lower allocations under decreasing global emissions scenarios, and higher allocations under increasing global emissions scenarios, than a country with decreasing shares of the global population.

Under the hybrid approach, every country picks the least-stringent approach, in terms of cumulative emissions over 2010–2100 (Fig. 2) or 2030-emissions levels (Figs. 3 and 4), while staying below a warming threshold. The modelling of the hybrid allocation consists in iterative steps to derive a global aspirational pathway whose bottom-up allocation matches any chosen emissions scenarios from IAM.

The iterative process starts by calculating the difference D(1) between the chosen IAM scenario (IAMscenario) and the bottom-up allocation of that chosen IAM scenario BU(IAMscenario). We then build a first aspirational emissions scenario A(1) that is IAMscenario discounted by half the calculated difference D(1)/2.

$$A(1) = IAMscenario - (BU(IAMscenario) - IAMscenario)/2. \qquad (1)$$

The following step consists in calculating the difference D(2) between IAMscenario and the bottom-up allocation of the new aspirational pathway A(1). We then build a new aspirational emissions pathways A(2) that is A(1) discounted by the difference D(2)/2:

$$A(2) = A(1) - (BU(A(1)) - IAMscenario)/2 \qquad (2)$$

These steps are repeated iteratively until BU(A($n$)) = IAMscenario or until A($n$ + 1) = A($n$):

$$A(n + 1) = A(n) - (BU(A(n)) - IAMscenario)/2 \qquad (3)$$

Note that A(0) = IAMscenario. Supplementary Fig. 2 shows the national emissions allocations of the aspirational 2 °C-scenario under the five selected effort-sharing allocation approaches (Supplementary Figs. 2a–e) and under its complete bottom-up allocation which is also the complete hybrid allocation of the original

2 °C-scenario (Supplementary Fig. 2f). The most favourable approach for a country may differ when considering a different global scenario, and therefore may also change over the iterative process (Supplementary Fig. 3).

The allocations of country-groups presented in this study (EU28, G8 + China or the rest of the world) are calculated based on the least-stringent approach of each of their members individually, rather than the least-stringent approach for the country group.

We update the CPC modelling approach[8] when applied to global emissions scenarios with positive emissions in 2100 to avoid national positive emissions after a period of negative emissions. The CPC approach then derives national ratios of the global emissions scenarios that are positive in 2100. These national ratios are a linear interpolation between 2010-emissions ratios and the 2100-ratios that result in equal cumulative per capita emissions. The impact of high historical per capita emissions has therefore a lower impact on 2030-emissions than under the CPC modelling applied to global scenarios with negative 2100-emissions. The lesser equity stringency on 2030-emissions aligns with the lesser global stringency. For example, the influence and importance of equitable allocation is lower when applied to business-as-usual scenarios. The accounting of historical emissions since 1990 and the autonomous energy efficiency improvement index are similar in both CPC setups.

Under the CBDR-RC hybrid setup used in Fig. 4, the hybrid approach is based on countries' least-stringent of three equity approaches only (CAP, EPC and CPC), following their 2030 allocations. Using this methodology, the least-stringent approaches applied to the 2 °C-scenario (RCP3PD), corresponding to a CBDR-RC bottom-up situation, are shown in Supplementary Fig. 3a. The least-stringent approaches of the aspirational 2 °C-scenario, corresponding to a CBDR-RC hybrid approach, are shown in Supplementary Fig. 3b. Only few countries have different least-stringent approaches under the CBDR-RC bottom-up and CBDR-RC hybrid cases.

The comparison between a complete hybrid (attributing the least-stringent of all five effort-sharing approaches over the 2010–2100 period, as used in Fig. 2) and a CBDR-RC hybrid (attributing the least-stringent of only the CAP, EPC and CPC equity approaches in 2030), is shown in Supplementary Fig. 4e. The current NDCs of China and Russia imply higher 2030-emissions than even the most favourable effort-sharing approach applied to the 2 °C-scenario, under a complete bottom-up approach (Supplementary Fig. 4a). The national allocations of the complete hybrid approach happen to yield similar results to the complete average of the five effort-sharing approaches' allocations[8] for most G8 countries (including the 28 EU countries), and China (Supplementary Fig. 4b). The variability across national results is greater than under the CBDR-RC hybrid (Fig. 3b). Reaching the complete hybrid allocation, including the status-quo grandfathering approach, implies raising the NDC's ambition by around 30 percentage-points of 2010-emissions for the USA and the EU28, and 77 percentage-points for China (Supplementary Fig. 4c). Aiming at 1.5 °C, rather than 2 °C, under the complete hybrid approach requires five additional percentage-points from the G8 countries and China, 20 for the other economies altogether, and more for LDCs (Supplementary Fig. 4d).

**Discussion on the monotony and uncertainty**. The bijectivity between NDCs ambition and their temperature assessment relies on the strict monotony of the relationship between global scenario's 2030-emissions and 2100-warming. We selected nine global scenarios every 0.5 °C to achieve such strict monotony at the global level. The 2030 emissions levels dependency to the NDC temperature assessment of Fig. 4 is a second-degree polynomial fit based on the allocations derived from the selected nine global scenarios. A second-degree polynomial fit smoothens the variability while preserving the greater sensitivity of national 2030-emissions allocations at lower 2100-warmings.

For 36 countries, the relationship between 2030-emissions and 2100-warming is non-strictly monotonous. In each case, the local maxima are at 4.8 °C or more, higher than their NDC assessments of the corresponding countries. The national emissions allocations at high temperature are only indicative for these countries. In the absence of interpolation, local maxima are found for 14 countries and are at lower temperature than the NDCs of only: Iraq, Trinidad and Tobago, Jordan, Brunei Darussalam and the Maldives.

The allocation of high global emissions pathways using effort-sharing approaches reflects equitable contributions to high global warming. However, the allocation of global BaU scenarios results in national scenarios that are, de facto, no longer business-as-usual. The range of 2030-emissions levels from global BaU scenarios reflects a range of modelling assumptions rather than a range of ambitions. The high-warming allocations derived in this study can indirectly be used to assess NDCs. Clearly, though, the equity allocations of BaU scenarios, taking into account the impacts the world is facing at global warming of 3.9 °C or more (Supplementary Table 4), cannot represent an equitable outcome[28].

The uncertainty resulting from the various NDC quantifications is shown in Fig. 4 by the height of the NDC ranges. The high assessment reflect the most optimistic end of the assessment range, including the application of conditional targets[44]. The low NDC assessment takes the least optimistic assumptions of unconditional targets. The maps of global warming responses under the CBDR-RC hybrid approach associated with high and low NDC assessments are shown in Supplementary Figs. 9 and 10.

**Choice of equity approaches**. The IPCC-AR5 presented quantifications of emissions reduction following five effort-sharing categories (Chapter 6, Figure 6.28 ref. [35]), which reflect combinations of three underlying equity principles: responsibility, equality and capability (Chapter 6, Table 6.5 ref. [35]). In addition, the IPCC-AR5 presented a category (but did not present a quantification) based on responsibility only, as proposed by Brazil in 1997, that derives emissions goals without allocation (IPCC-AR5 Chapter 6, table 6.5, refs. [20,35]). The historical responsibility of countries for their past emissions is modelled here with the equal cumulative per capita approach (CPC) that uses only historical emissions and population data to calculate countries' emissions allocations. The GDR approach also uses historical emissions and accounts for countries historical responsibility. Other approaches of distributive justice (e.g., based on sufficiency[59], or using the Human Development Index) and other metrics (e.g., accounting for consumption-based emissions or exported emissions), not currently used in the IPCC report or under the UNFCCC, are not modelled here but would bring useful perspectives that could be integrated in the hybrid approach.

The sensitivity of effort-sharing approaches to their input data and parameters indirectly affects the results of their hybrid combination and thus of the NDCs' warming assessments presented in Fig. 4. The sensitivity of the effort-sharing framework used in this study[8] was studied under a range of parameters consistently across the five effort-sharing approaches[48,49]. The sensitivity analysis was performed by quantifying 3 to 81 combinations of parameters, depending on the approach. Because of their flexible parameterization, the most sensitive approaches are the CPC (sensitive to the period covering historical emissions) and GDR approaches (also sensitive to business-as-usual emissions projections and internal parameters, such as wealth threshold and responsibility-capability ratio[16]). The GDR approach is included in the complete hybrid quantifications, but not in the NDC assessment of Fig. 4. In addition to the allocation approaches uncertainty, various exogenous assumptions have equity implications. For example, an earlier (later) convergence date for the EPC and CAP approaches, and earlier (later) starting date to account for historical emissions is expected to favour—result in a lower NDC warming assessment—developing (developed) countries compared with the current NDCs.

The selection of effort-sharing approaches to derive countries' least-stringent allocations directly influences the hybrid allocations of all countries. Removing the least-stringent approach of a country group would penalize these countries that would have to follow a more stringent approach and would consequently favour all other countries.

The complete hybrid allocation, which uses the five effort-sharing approaches, results in lower temperature assessments of developed countries' NDCs and consequently in higher temperature assessments of other countries' NDCs (Supplementary Fig. 5). The assessment of China's NDC is still higher than the temperature scale range.

The choice of the five effort-sharing approaches includes the grandfathering approach (CER) that is only implicitly supported by some countries through their pledges. The grandfathering approach represents a status-quo in terms of equity where all countries conserve their share of global emissions and mitigate at a common rate, that of the global scenario. The GDR approach, categorized as a responsibility-capability-need[35], preserves a right to development through the allocation of mitigation requirements, although the link between the objective right to development and the selected implementation criteria is subjective[11,16,60]. Excluding only the grandfathering approach, and including the GDR, represents four equity approaches representative of the four key equity principles dimensions: responsibility, capacity, equality and the right to sustainable development (IPCC-AR5 WG3 Chapter 4, Section 4.6.2, ref. [24]).

The modelling of the GDR approach relies on business-as-usual (BaU) emissions that countries do not mutually recognize. The BaU emissions used here are downscaled from RCP8.5 resulting in allocations substantially higher than other allocations for Eastern European countries and Australia. Compared with the CBDR-RC hybrid setup presented in Fig. 4, the inclusion of the GDR approach in the hybrid setup results in a lower temperature assessment in favour of the NDCs of Eastern European countries, Australia and South Africa, and higher temperature assessment disfavouring India, Brazil, Mexico and Indonesia (Supplementary Fig. 6). The GDR approach was designed to allocate mitigation efforts to individuals with income above a certain threshold. The share of a country's population above the income threshold is derived using the Gini index of inequality[16]. While the GDR is complex method accounting for more indicators than most approaches in the literature, its reliance on hypothetical BaU emissions and Gini projections that are not commonly agreed indicators results in an important sensibility that cannot be easily resolved. The notions of responsibility and capacity that the GDR is based on are conveyed by the equal cumulative per capita (CPC) and capability (CAP) approaches, respectively.

Removing both the GDR and the grandfathering approaches from the hybrid allocation (Fig. 4) leaves equity approaches that rely on measurable population and GDP data that can be updated over time.

**Warming assessment of the global scenarios**. The evaluation of the warming resulting from the bottom-up scenarios requires the calculation of their GHG compositions (Fig. 2). The Equal Quantile Walk (EQW) method[61] is used to derive a multi-gas scenario that is needed by the simple climate model MAGICC[62,63] for

the evaluation of the temperature response. Land-use $CO_2$ is taken directly from the target 2 °C-scenario and 1.5 °C-scenario. The probabilistic temperature projections of the complete bottom-up multi-gas scenario is constrained by historical global mean temperature observations and a priori estimates of uncertain model parameters, such as climate sensitivity[64]. We use MAGICC version 6.8 and climate sensitivity distribution from ref [40].

The aspirational scenarios are of purely numerical nature and have no underlying economic assumptions. These scenarios, which include bunkers but exclude land-use emissions, show a steep decline in the first half of the century with minima in 2070 (aspirational 2 °C-scenario, Supplementary Fig. 11a) and 2060 (aspirational 1.5 °C-scenario, Supplementary Fig. 11b) and very low emissions throughout the second half of the century. The EQW is based on older scenarios that did not have negative $CO_2$ emissions. The EQW thus cannot model negative fossil $CO_2$ emissions. However, these negative emissions are necessary to reach the low emissions levels in the second half of the century. We calculate multi-gas emissions scenarios consistent with global aspirational scenarios given in $CO_2$-equivalent units as the aggregation of all Kyoto–GHG.

A dataset of full-gas 1.5 °C scenarios[54] is harmonized to 2010-emissions[19]. Harmonization is carried out for the global aggregate Kyoto–GHG value excluding land-use emissions but including bunkers emissions. Bunkers emissions are included because for most IAM scenarios they are not available as independent time series but included in the world total and regional time series. The harmonization factor linearly converges to unity in 2040, and scenarios are unchanged onwards. The Kyoto–GHG harmonization factor is used for all substances.

From the harmonized database, we select the ten scenarios with the least mean-square distance to the aspirational scenario over the 2010–2100 period. Absolute distances are used because relative distances are not meaningful to compare positive and negative values, which is inevitable when using low emissions scenarios with negative emissions. The average across the ten selected scenarios is taken for each gas individually. In order to match the aspirational scenario, only the $CO_2$ emissions levels are reduced, which corresponds to additional mitigation implemented in the fossil $CO_2$ sector. Indeed, fossil $CO_2$ emissions can be mitigated more profoundly than other GHG (e.g., methane from agriculture). Carbon dioxide is the only gas where prototypes for large-scale negative emissions technologies exist (even though costs, side-effects and acceptability of large-scale projects are uncertain). Conversely, additional fossil $CO_2$ emissions are assumed where the aspirational scenario is higher than the average of the selected scenarios. Land-use $CO_2$ is taken directly from the target 2 °C-scenarios and 1.5 °C-scenarios.

The resulting multi-gas aspirational scenarios are then used for probabilistic temperature assessment as described above for the bottom-up scenarios.

**Code availability**. The code developed for this study is based on the PRIMAP modelling environment, developed at the Potsdam Institute for Climate Impact Research (PIK) and is not publicly shareable. Requests for code will be jointly considered by the authors and the PRIMAP research team.

## Data availability

The data that supports the main findings presented in Fig. 4 is available in Supplementary Data 1 and can be visualized on the interactive website http://paris-equity-check.org/warming-check. The rest of the data is available from the corresponding author upon reasonable request.

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

### Acknowledgements

We gratefully acknowledge the work of modellers behind the IPCC-AR5 emissions scenarios. M.M. is supported by the Australian Research Council (ARC) Future Fellowship (grant number FT130100809). Great thanks to Peter Christoff, Joeri Rogelj and Anita Talberg for their comments on the manuscript, and to Johannes Gütschow and Louise Jeffery for their help with the PRIMAP database and with MAGICC.

### Author contributions

Y.R.d.P. led the study, performed the calculations and designed the figures. M.M. suggested the study and updated and managed the composite PRIMAP database. Both authors contributed to discussing the results and writing the paper.

### Additional information

**Competing interests:** The authors declare no competing interests.

