## [Peer Review File · Nature Communications]

Reviewers' comments:

Reviewer #1 (Remarks to the Author):

Dear editors:

Upon reflection, I can't quite decide between the two forms of rejection that your guidelines leave me with:

- Reject, but indicate to the authors that further work might justify a resubmission
- Reject outright, typically on grounds of lack of novelty, insufficient conceptual advance or major technical and/or interpretational problems

Rather than strain to choose one or the other, I will attempt to explain myself.

Comments:

On line 5, we begin with "five effort-sharing allocations of emissions scenarios to achieve the Paris Agreement." This is curious because while the authors repeatedly refer to "five" approaches, and cite the IPCC as their source, the IPCC itself (see page 458 of AR5) actually categorized effort-sharing proposals into six classes.

I don't want to belabor this point, but I do want to say that something was lost in the translation. It has a lot to do with historical responsibility, which is not just a matter of resource sharing. And it has a lot to do, as well, with the "right to development." In any case, the issue here is notable because this paper is based upon the authors' leverage of their set of five approaches, as if it were representative of the entire range of oft-cited, well-supported, and influential effort-sharing / equity / allocation approaches, and indeed as if this set was universally accepted as representing a coherent equity approach. Neither of these conditions, however, is true.

Beginning on line 19, the core idea of this paper is introduced. This idea, while interesting, is frankly odd. Basically, the authors claim that all countries will use the allocation principle that reflects their interests, and that it is, therefore, both illuminating and politically helpful to grant that they will do so, and then to observe that if they were to also agree to a much stronger global temperature target than they have in fact agreed to, the Paris temperature objectives would come more closely into reach.

On line 32, they claim that the literature on effort-sharing approaches is "rich." I do not agree, but cannot today take the time to explain why. I want to stipulate this point, however, because it's relevant to the central point of the authors' method – that countries will agree to act as if their disagreements about equity are immaterial to their willingness to accept far more ambitious targets than they have thus far asserted in their NDCs. Rather, I believe that such a willingness, if it ever comes, will necessarily involve a reexamination of the equity challenge.

On line 47, we get the citation that supports the claim that countries "generally support interpretations of distributive justice that justify their negotiating positions." I will not belabor this point, but I will say that citation 22 is an interesting but polemical paper, while 23 is a blithe reference to an AR5 chapter that could as easily be cited to support the opposite claim. Do note, too, that the situation here is more complex than the authors aver. See for example Winker et. al., Countries start to explain how their climate contributions are fair: more rigour needed, Int. Environmental Agreements, accepted 27 October 2017, which offers empirical evidence "that countries do not necessarily use equity indicators that favour their equity argument." This paper is also interesting as evidence that the equity debate is still in early stages (hardly "rich") and that, to the extent that it is reflected in the NDCs, it is not reflected in a manner that is captured in the limited set of approaches that is the basis of this paper.

For the quick review of the stakes here, see AR4, WG4, Chapter 4, especially 4.6.2.1

I found the paragraph beginning with line 48 to be unhelpful. It touches on a number of issues, but it does not raise what would seem (given the authors' overall strategy) to be the main point, which is the assumption that nations simply take equity positions that suit their interests within a decision space defined by author's five approaches, and will hold those position steady while adjusting to the demands of a massive and extremely challenging increase in global ambition.

The para that begins on line 63 is a critical one. First of all, I do not agree with the authors' claim that their exercise here is not normative. In fact, I find this claim to be so unsupportable that it makes me lean toward the "reject outright" recommendation. In a nutshell, I believe that their method is strongly skewed in favour of wealthier, higher-emitting countries.

Consider the sentence that begins on line 64, a sentence that strikes me as problematic, if not simply wrong. Briefly, and despite citations 26 and 27, the issue is not merely "national preference." The issue is also "relative fair shares" – the real core of the effort-sharing debate – and it inevitably raises normative challenges that should not be perfunctorily dismissed, especially not by way of blithe claims to be avoiding normative judgment.

Ultimately, what is at stake here is a neo-realist claim that we should finesse normative judgment. In fact, the claim that counties will do just this when faced with the overwhelming imperatives of climate stabilization. In effect, the authors are arguing that the "agreement to disagree on equity concepts to achieve a common temperature goal" is key to finding a path forward. It may be – I don't claim to know – but I do not find the argumentation here, the method, to be particularly convincing. In particular, I do not see any defense for the claim made at the end of this para, that the authors' "hybrid approach" can inform the PA's ratcheting's mechanism. Have I missed it?

Line 77. Again, the reification of the "five IPCC effort-sharing categories." (Note that they have been promoted). What I find particularly interesting (line 82) is the admission here that the CER approach (popularly known as "grandfathering") is widely considered to be unfair, because just a bit later (line 117) we are told that "current national emissions ratios, at the root of the controversial CER approach, also influence near-term allocations of all other approaches." Which is to say that CER itself is tossed out along the way (and I applaud this move), though GDR is tossed out along with it, because it is "redundant." The result is that EPC, CPC and CAP, a group that is heavily influenced by CER (unfair CER) is taken as the core of the assessment methodology, going forward.

Line 119. By the way, no convincing explanation of why the GDR approach is "redundant" is ever given. What is said is that "The GDR approach blends the concepts of historical responsibility and capability covered by the CPC and CAP approaches, respectively," but this misses the whole point of the framework, as I understand it, which is NOT a per-capita framework, but rather one that is based upon "the right to development," and which treats that right in explicitly progressive terms. In fact, as I understand it, modeling GDR – which by the way was long ago renamed the "climate equity reference framework" – by way of a parameterization that reduces it to a per-capita system is explicitly to caricature it. Which is to say that it is NOT redundant with EPC, CPC, or CAP and that to paint it as being so just raises the question of how GDR was modeled by the authors in the first place. In any case, given that the authors are leaning so heavily on the supposedly canonical nature of their five categories, the further simplification of the "five" into "three" seems a real methodological stretch.

The real point in all this, I suppose, is that the "hybrid" method that the authors are speaking for in this paper, while it may be interesting, is not well served by being erected upon the fragile foundation of the "five approaches." As they themselves note (line 145), results "strongly depend on the set of effort-sharing approaches included in 'hybrid' approach modeling." Which is exactly why, to properly evaluate the usefulness of the approach, it would be nice to see it modeled in a

fair way, one that in fact takes account of all the equity challenges discussed in AR5, rather than just the ones that remain after the cascading reductions that seem to lie behind this paper.

Finally, let me note (line 152) that the authors statement that “The emissions trajectories quantified in this study offer an alternative effort-sharing scheme that reconciles the bottom-up pledging architecture of the Paris Agreement’s with its mitigation goals. The resulting equity-based metric acknowledges countries’ self-interested positions and does not hypothesize an international agreement on a unique approach of equity.” Let me, in particular, say that I find the logic here, or at least the desire, to be quite understandable. Self-interest, after all, is a major factor in all this. Having said this, let me add that I see no reason to believe that this paper, as it stands, will be helpful. For one thing, it cleans up the “five” by dismissing the CER approach, but carries the CER idea forward under other names. And it caricatures and problematically dismisses a promising approach (GDR) as a per-capita approach, when in fact its great strength is that it is explicitly not a per-capita approach. Not, by the way, that I think GDR is without its problems. It’s hardly the answer to the riddle of history, but still. This is all supposed to be about fairness, right?

Reviewer #2 (Remarks to the Author):

Peer review of “Temperature assessment of the bottom-up Paris emissions pledges”

Reviewer: Daniel Horen Greenford

Disclaimer: Don’t mean to be rough with comments so if it seems I’m harsh at times, it’s only me being terse and opinionated. Things that I feel especially strong about have been emphasized and repeated. My angle is one of ethics side so gave most criticism and comments on that, and less weight on the technical side, which I hope other reviews have addressed more thoroughly. Hope these comments are helpful to your work! Good job! Lot’s of great insights here! Hope this gets out soon and there’s good media coverage of it. Feel free to contact me for further explanation or to talk over some points. I have chosen to disclose my identity, in keeping with the journal’s philosophy of transparency. Apologies for the delay, busy time of year! Also sorry if this is a bit of a mess and for some repetition. Think I got most of what I had to say out in this sprawl of points. Keep up the great work.

General comments:

Overall excellent work. The paper is very interesting and informative, but needs to be tightened up for clarity. Understandable but a bit cumbersome and flow not as illustrative of main themes as it could be. I have some suggestions as how to make it more accessible to a broader audience and if I get what points you’re trying to make, how to make them jump out more clearly and explicitly. I also am unsure of the best way to structure the narrative here. In my opinion, the most interesting result is how much warming would be if all countries followed paths of analogous ambition to a chosen country. If I understand correctly this is what you show when you talk about what “prolongating bottom-up NDCs of a given country would lead to.” Correct me if I’m wrong, but if this is what you’ve down, that’s the most important insight for me. I think the more technical analysis of showing what temperatures lead to successful hybrid allocations is also super interesting but less important insofar as it is pretty tough to see countries following this approach (not to criticize too severely the applicability of your work to realpolitik because pretty much all work in this field, including my own is likely never making into actual policy), so the real

contribution is insight into how to conceptualize ambition and how far countries actions are from the ideal. I think the other realization that targets would have to be much lower if countries would all pick the effort-sharing method that benefited them the most is interesting but the way you frame it seems to suggest that countries would be convinced by this, and I'm pretty skeptical of this. It is a bit like trying to tell your son that he can have whatever allowance he wants but you'll only give him \$5 and he has to wait for inflation to make it \$50 for example. Maybe not a precise analogy but the same logic applies of making someone think they are choosing something/winning but in reality you're just telling them what they want to hear and then giving them the share that's due to them in first place (not exactly how your hybrid works, I know, but pretty close, looking at)

One thing that bothers me, and I'm not certain I'm interpreting this correctly: Is your "hybrid approach" just a way of tricking countries into thinking they have agency? Giving them the false impression that they're choosing what's best for them (they are to begin with, we assume, and not even picking the least stringent of equity methods for current NDC derivations) when in actuality you are just tweaking the temperature target so it works. Kind of like sleight of hand to make everyone feel like they're coming out on top. If this was proposed, would nations really think it was fair? What would you be telling them? If I'm reading this right, I see "so everyone pick the metric that gives you the biggest share of the carbon budget, then we lower the temperature target such that we get to the stated global objective." If I were a nation who was told that, I'd be scratching my chin a bit and ask, "hey are you sure this is a winning deal for me, aren't you just leading me to believe it is but then tweaking the target so it works and hence cutting my share just the same?" Of course the share derived with your hybrid approach may be bigger for certain countries and smaller for others (do you have some numbers showing how much the difference is between a chosen equity metric and the hybrid outcome? This is figure 2 panel b, right? Showing the difference between hybrid and average, but does average include CER?)

So the top-lining messaging is very important and I think this can be clearer. My main two takeaways from this paper are:

1) This is what temperature rise would be if all countries followed analogous ambition to a chosen/specified country's ambition, e.g. if all countries acted in a manner of analogous ambition to China, we'd warm the planet by 5.1degC. Am I interpreting this correctly? This is in my opinion the most important takeaway and should be more plainly worded for a more general audience, try to be clear in the opening paragraph and abstract for sure. A lab-mate who is in climate adaptation did not get that from reading the abstract, so not accessible enough. I would suggest something like (instead of line 24-26): "We also (I would say 'also' since this is another significant finding in your study and it seems to be lumped into the previous part but I'm confused as to whether the NDC analysis is the same as the hybrid result with higher temperature thresholds) explore what temperature would be induced when using each country's NDC as the specified level of ambition for a bottom-up approach to global mitigation efforts. In other words, we find that if all countries acting with a level of ambition of a single chosen country, emissions would result in a given amount of warming. We find that the NDCs of India ... respectively." I think the added sentence or two of explanation is necessary.

2) If all countries pick the effort-sharing approach that benefits them the most, i.e. gives them the largest share of the remaining carbon budget / amount of allowable cumulative emissions, then the world would need to pursue a global warming threshold target lower than the Paris Agreement's window, and find that in order to reach 1.5 and 2 degrees, the "aspirational" target would have to be lowered to 1.1 and 1.3 degrees.

Also, does your analysis based on Climate Action Tracker's work use a similar approach? Is your estimate of the impact of abandoning the Clean Power Plan novel or just quoting their work? If the latter, it doesn't belong in the opening paragraph, but it seems like you take their initial work and then quantify the warming impact so this is pretty interesting. Curious as to how they do it, and generally how you and Meinhausen have come up with pathways for NDCs based on somewhat

ambiguous descriptions of policy. I intend to read more about this, but generally very important and impressive work that's invaluable to the community so thanks very much for this!

I can't stress this enough: I would NOT include CER in the "average" since it is clearly not equitable. Please EXCLUDE CER from your "average", so only use the 4 methods: GDR, CAP, EPC, CPC. And to be an ethical purist, I would argue that equal per capita is NOT an equitable outcome, it's a compromise at best, since it disregards historical emissions, and the historical overuse of the atmosphere or climate debts or however you prefer to frame it, it still is not a truly just outcome. It's a "let's wipe the slate clean and start from here outcome." And what are/is the convergence date(s)? Is that where the range for EPC comes from or that's just the range of uncertainty from the NDCs, right? In your previous work you use a 30 year convergence period. That the same parameter here? CPC on the other hand does take historical emissions into account so I think that's just. GDR does blend history fault with present responsibility based on available wealth (what they call "capacity"), and is very nuanced and I think very equitable regardless of most parameterization, and I'm not too family with CAP but if it's similar to Kemp Benedict's method for deterring "capacity" I support it. If it's a simpler version of that, it seems just conflating financial capacity with GDP, it's a much worse metric than GDR's capacity so even if it's an IPCC standard, it doesn't add anything. I understand the appeal/benefit of going with established categories and methods so I would stick with that, but I have to voice my grievances with this effort sharing approaches even if this is not the paper to debate them. That said, I still think you have the creative license to exclude ones you think are bunk or not equitable so if you find my arguments convincing please do dispense with EPC and maybe CAP and just keep GDR and CPC. Or as a compromise, have another "average" that is only those or for a broader one that isn't pure equity and take the average of the four. Still have to be hard on the no CER "average" being represented, but if you do want to include the original one with it, be clear about it. It is analogous to something like Raupach and colleagues' "blended" equity setting (Nature Climate Change, 2014), which is the middle ground between totally unfair and climate justice since it's just the average of CPC and CER. The average of the five will be much more equitable than that since there are more equitable allocations than non-equitable ones to shift the average. I actually tried taking the data from your Nature Climate Change paper and removed CER to see how much of a difference it made and it's not huge but still significant.

Title:

Abstract

Main messages/take homes

If I get this (sorry for being repetitive in the comments by the way, helps me rephrase the ideas and clarify my thinking and you can correct me at any step where I'm misreading the study)

The takeaways are:

- 1) What would the temperature target need to be to meet 1.5 or 2 deg C if all countries picked the equity principle (including CER) that gives them the largest share of remaining emissions.
- 2) What would the rise in global temperature be if all countries acting in accordance with analogous level of ambition to a chosen country (or policy change).

General points about study:

- I'm interested to know what the hybrid approach looks like without CER. Can you just exclude it as an option? It really isn't part of the suite of equitable choices and so let countries "pick" any other approach, just not grandfathering.

Line-by-line:

17-19. This is what you found in your previous paper? Sounds like original research when in present tense. Should be "In a previous study, we found" if so. Also, sounds like you mean that current NDCs align with least stringent effort sharing method, and add up to 2.3 degC, but this isn't what ref [4] found (more like 3-4 degC). Needs to be clearer that this is a novel finding when you "prolongate" bottom-up approaches. Also, don't use the word "prolongate," try something simpler with equivalent meaning like "extending" or "extrapolating." Or maybe you meant "elongate?" Since prolongate is a more arcane way of saying "prolong" and refers to either extending the duration of time or distance in space. Do you mean : "When extending/extrapolating the current level of ambition for each country, we derive a "bottom-up" estimate of global mitigation resulting in 2.3degC of warming by 2100 (50% chance)." I also wouldn't use the word "equity" since you are referring to countries picking the least stringent effort sharing method which for developed countries is always grandfathering (CER).

23. So this hybrid approach is from ref[1]? Or based on theory proposed by Mace? Haven't read this article yet. If it's your method, make sure to note that it's novel.

24. To make this clearer: "By creating this range of top-down emissions allocations, we are able to infer which global temperature change matches which NDC. This is effect is equivalent to assessing what temperature change would be brought on if all countries acted with analogous ambition to a specified country's level of ambition (as defined by the NDC). For example, we find that if the world acted in alignment with the NDCs of India, the EU..., warming would result of 2.6degC,..., respectively." If I am interpreting this finding correctly, this is a clearer way of communicating the result (which is in my option the most interesting/profound insight form this study, since it informs the public and policy makers what the level of ambition an NDC amounts to in temperature change).

25. Curious as to the methodological approach of getting from Climate action Tracker's work to this finding. Will see if you explain in methods.

41. Cite something on the upcoming Facilitative Dialogue (UN document)

44. Insert "have" between "experts" and "suggested"

47. Change "justify their negotiating positions" to something a bit more clearly about self-interest. Their negotiating positions are symptomatic of their self-interest. I know you say this later in the text so not absolutely necessary to change if you prefer this framing, but I just wanted to point out that it's helpful to be explicit about the politics. Could say "...justice that best serve their self-interest and likewise justify their..."

49. Remove "indeed" and replace with: "One way/approach to reconcile this ambiguity is to combine multiple concepts of equity into a single metric using weighting factors / by taking a weighted average of each approach / using a weighted average. Weighting allows for parameterization that can be adjusted to reflect the level of importance placed on any single approach."

51. Ref [19] sounds like similar research, if yours is an extension of their method, connect/relate the two by adding something like: we extend this methodology by including the full suite of IPCC effort-sharing approaches and [whatever else you feel you do above and beyond their work]."

52. Why is ref [19] after leader and not end of sentence, necessary to cite again, seem to be talking about the same paper as last lines, not that crucial but maybe better way to cite? Something for the editor perhaps.

51-56. OK I'm having a bit of trouble seeing the difference between their work and yours. So

you're saying that they set temperature targets for each country differently so that they do their fair share based on each of two chosen effort sharing metrics or the weaker of the two /. Better one for them? And you also let countries pick the least stringent effort sharing metric but lower the temperature threshold for all countries until they meet 2 (or 1.5) degrees?

59. So did they do this in ref[19]? And you now did it for your set so the only difference is the number of choices they had? Maybe you just need to make the description of ref[19]'s study clearer. Work on that last paragraph. I'd be happy to take another look at that and anything else I've said could be clearer after you edit again.

57-62. Again just be clear where ref[19]'s research ends and yours begins. After reading this over once and reading it again, I know, but on a first read it may be confusing to some. Maybe have a paragraph laying out the hybrid approach from "A to Z" rather than defining it implicitly in line

61. Don't know if this is nitpick and that maybe the style here is supposed to be more journalistic, but a suggestion nonetheless. If others have found this reads well and they see the definition then feel free to ignore.

65. So are you getting into Pareto optimality here? Not sure but maybe this is worth elaborating here or maybe those references are enough and this isn't the place to go further into this. Maybe there's a less technical / more intuitive way of describing this optimality. Essentially, all countries agree that they want to limit warming to 1.5 to 2 degrees, (the absolute gain being saving civilization from possible destabilization and collapse), then they also want to get the most out of any arrangement supporting this for themselves. These two are generally in direct conflict and lead to a tragedy of the commons. This kind of research can be seen as a way of regulating the commons when parties agree that they want to protect the commons and will hold each other accountable for overuse of the atmosphere while being allowed to vie for additional benefits to themselves. Maybe add a line making this more concrete, about how this looks like in effect. Not sure if it's appropriate in a paper like this but I might add something like: We start with the assumption that all countries support the 'absolute gain' which is the safeguarding of human civilization, since any individual economy's welfare is contingent/predicated upon the geopolitical stability of the global community, then are allowed to vie for whatever additional benefits they can garner/secure for themselves, the 'realtime gain,' without undermining the absolute gain.

68. I like the "agree to disagree" framing

73. Mention how your choice of scenarios is commensurate r made to represent/ quantify the Paris Agreement's targets. Like "we pick two temperature scenarios that we feel best quantify/capture the meaning of the PA's 'well below 2C' and 'pursue efforts to limit warming to 1.5.'"

78-82. So here I'm going to chime in about "equity." I don't feel that EPC is equitable since it doesn't include historical consideration. I also don't think CER should be included at all or maybe have another average without it. Sure it represents a choice, but is it a choice we want to offer nations? Sure you may argue, they're picking it de facto right now anyway, but they're not, it just happens that the poetically-derived targets countries are picking from themselves coincidentally fall in line with this approach. For all intents and purposes they could pick any metric out of the sky, e.g. we won't be able to decarbonize and actually need a bigger share than our current ratio since we need more revenue from carbon intensive industries since some other country outdid us on some low carbon sector and now that's failed." I mean you can make up anything to justify/argue you need more emissions. I think it's impossible to not be normative or a bit prescriptive here. I see the appeal of proposing a "non-normative" framework here, and I think CER is the furthest out you'd even consider countries being able to justify taking for themselves, so in this sense it's accurate as a depiction of reality, but it's tricky. If you are going this route, it's imperative that you remind the reader, this isn't a policy portal, we're not legitimating the continued colonialism of poor nations by wealthy ones and the perpetuation of this kind of

international oppression, we're simply asking the question: "what would you have to tell countries to shoot for if they picked their current NDC-level of ambition." So in this sense it's a bit of trickery. You're saying sure pick what you like, then aim for this temp instead of 1.5 or 2.

82. Remove "in the literature" since it is plainly unfair, then cite the literature that supports that (those should be fine)

83. Change "is reflective of" to "matches"

84. Do they really declare their contributions as "fair?" Haven't read this work of yours, but surely this definition of fairness is contingent on how other countries are conducting themselves. Maybe add a line about that. "These countries claim that their efforts are fair, but only in the context of the rest of the global commutates efforts. This definition of fairness is dynamic and bound to become more stringent as countries progressively ratchet up their own ambitions. Fairness is always seen asa. Relative measure, in a 'I only want to do as much as my neighbour /the next country does' kind of way." (Make less colloquial, of course)

85-87. So here is the rest of your explanation of your hybrid method, I see it's a bit spread out, if you understand what I'm trying to get at. Consolidating this earlier in the text with the definition of your method may be helpful to the reader. Again, if others feel it makes sense in this order, feel free to disregard.

91. Again, don't like the word "prolongating." Unless this is a technical word that's absolutely necessary and says something more accurate than e.g. extrapolating, I'd change it.

92. So this implies that at the moment, may countries are proposing NDCs that are even laxer than the CER allocation? (Citing "Paris Agreement's need a boost..." paper) So this could be a good thing to point out explicitly. Maybe add a brief sentence to state explicitly.

102. Again just a note to provide an average with CER excluded for this too. It's in there right?

105. I think you note somewhere else to remind the reader that Brazil's obligation would look different (have to be more ambitious) if you included LULUCF

116-117. change "excluding the redundant..." to "excluding the inequitable CER and redundant GDR approaches." Also I strongly prefer GDR to CAP since it uses a much better way of calculating CAP than CAP does itself. I would opt to dispose of CAP in favour of using GDR with $R=0$ and $C=1$. (Will make another note in Methods)

121-22. Why "orthogonal," do you mean that these dimensions are independent of each other? If so, I would say they're not, they're very much a combination and reflection of each other/ two-sides of the coin. Capacity was gained through the accumulation of wealth derived from the bringing of fossil fuels, which is historical fault (or what his refereed to responsibility). I would use another word or don't qualify it at all. Seems like you're trying to make a math analogy here, like you're looking for the basis of a set of linear equations. Don't know if this works well for equity-based policy. Way more messy than linear algebra!

122-123. What's this tool, the Paris Equity Check website? Or this methodology you use to see what warming would be like if all countries acted with similar ambition as one chosen country's NDC pledge? Don't really get this sentence.

132-134. So are you suggesting that Brazil would get an even bigger share if deforestation were included (because the rate of deforestation has slowed? Please clarify. I would not say that since that's like warding them for being less bad, when rewards should only be additive, not less subtractive. Like in Canada, the new Conservative leader has said things like since we have so

much forest on our territory we should be entitled to emit as much as we like since our forests offset it. He forgets that forests don't belong to anyone really and your territory is coincidental. In my opinion, countries shouldn't be allowed to claim decreases in emissions from not destroying their forests, but in keeping with the philosophy of good stewardship, they should be penalized by being allocated increases in emissions when they deforest their land. Brazil deforesting less than a previous should not grant them extra emissions allowances. But if we were to include LULUCF for everyone, and Brazil was reducing deforestation, it would make room for more emissions. Also maybe this should be at line 105?

141-142. So you're saying that the order of NDC-determined T changes doesn't change when accounting for uncertainty in how you quantify NDC pledges (uncertainty due to ambiguity in how pledges are worded/described)?

149-151. Don't quite get this. Can you elaborate on "contentious future reference emissions"? Does this mean that there's a lot of uncertainty regarding what emissions will look like for these countries? How does Australia, a developed nation with high capacity, benefit in a similar manner to eastern European nations and South Africa? There all quite different.

159-161. Needs work. You trying to say they're especially inadequate since these are the same countries who championed deep and rapid decarbonization and climate finance to support mitigation in developing countries, and their slacking is therefore especially egregious/hypocritical? Like: not only are developed nations not pledging NDCs in line with the PA but they also have rhetoric that says they would be the leaders on rapid mitigation. (and as a footnote, also are not living up to their rhetoric on climate finance contributions). In other words: "The disconnect between lofty/ambitious rhetoric and lack of action makes the latter even more reprehensible."

Minor point on style: you put quotations inside punctuation. This deliberate? Typically one puts the quote outside but I think this is becoming a style preference. I sometimes think it's nicer and more representative of the meaning of a sentence. Does nature have a style guide/preference?

201. typo = "substracted" change to subtracted

218. When you say "iterative" do you mean you repeat the calculation for every year? If so, I wouldn't use the term iterative since it refers to (in my mind, at least) numerical solving. This was confusing.

277. Indeed, a 3 to 4 degree world would have highly inequitable distribution of impacts. Maybe cite Althor et al., 2016 Scientific Reports. "Global mismatch between greenhouse gas emissions and the burden of climate change"

282. But you said that approaches are picked for individual countries?

283. Including ones in the country group?

292. "same rates" as what they were doing already, not the same as in they all mitigate at a given rate. Maybe reword to clarify.

337-338. Citation to substantiate "mitigated relatively easily compared to other GHG." Is it that simple? Some study must elaborate on this? I'm personally curious.

Figures

Like the figures, no major criticisms or recommendations. Maybe add a point on how the diagonal in figure 2 represents matching NDC and equitable ambition.

Methods

To summarize my thoughts as I wrote comments line by line:

- 1) I think it would be good to dispose of CER, or if being used to inform reader that it happens that current ambition often matches a CER share, then great, but maybe include an average with it excluded.
- 2) I strongly prefer GDR to CAP since it uses a much better way of calculating CAP than CAP does itself. I would opt to dispose of CAP in favour of using GDR with $R=0$ and $C=1$. CAP doesn't capture actual capacity (as discussed by Kemp-Benedict in his justification for his Gini parametrized wealth distribution with development threshold) it's subject to being misrepresented, since GDP/capita hides/asks the excessive wealth of the super rich and the aggregate wealth of lots of people living in abject poverty. I'm reminded of the old adage that illustrates how averages can be very misleading — the average person has one testicle.
- 3) I know you use "cost-optimal" pathways, but I'm of the camp that cost-optimal pathways are highly contentious as they don't really account for future damages, and of course heavily privilege the wealthy and alive over the poor and the yet-to-born. I'm also wary of the magnitude of negative emissions in these pathways and would like to see what mitigation would be required to avoid negative emissions a) entirely and b) with some more technically-supported limit. I know there is a literature review on the technical limits and feasibility of large scale deployment (see Smith et al. 2016 NCC. "Biophysical and economic limits to negative CO2 emissions"). Maybe there would be some clear numbers there to use. I am currently looking into this for my own work on making decarbonization pathways for Canada that respect these constraints.
- 4) Convergence dates: I see you use a 30 year convergence date (as being politically viable) so in the non-notative spirit of this paper, maybe best to stick to that. Is this supported by other research? Could you include a scenario or a range provided by picking a faster convergence?
- 5) Cumulative emission period: Can you do the same thing for 4) but for the cumulative emissions period? Maybe try one at 1960 and even 1850 but that may be excessive, and unnecessary since the bulk of emissions is from mid 20th century onwards. See Matthews, 2016. Nature Climate Change. "Quantifying historical carbon and climate debts among nations" for rationale. Just the argument of "we knew beyond a reasonable doubt" (1990) and "intent matters but it's not everything" (1960).

Congratulations on the submission and good luck with the revisions!

- Daniel

Reviewer #3 (Remarks to the Author):

The manuscript "Temperature assessment of the bottom-up Paris emissions pledges" introduces a new method to evaluate the individually NDCs and can inform the ratchetting-up process of the Paris Agreement.

Interestingly, the paper follows the architecture of the Paris Agreement, with a global mitigation goal and bottom-up pledges. The paper presents an operationalization of an 'agreement to disagree' on equity concepts to achieve a common temperature goal. Each country can choose an equity approach to determine its effort. The least stringent equity approach are chosen for each country when allocating global scenario emissions to each country. This acknowledges the countries' self-interested positions. The global scenarios, used to allocate global emissions to countries, are adjusted such as the temperature goal is reached (1.5/2.0 degrees or to follow IAM scenarios).

The topic of the paper is NDCs and global temperature targets. A method to assess the ambition of the current NDCs is presented in the paper, and would be of interests in a wide community.

I feel that the paper is not very well presented and some clarifications are needed before possible publications. My main comments are related to Figure 3. See below.

Here are my specific comments:

Line 14-16: "Current NDCs align,..." consider rephrasing.

Line 17: Which emission scenarios are you referring to?

Line 73: "First,..." What is the "second/then"?

Line 96: Do you mean that the global temperature goal should be lower than 1.5? Consider rephrasing.

Line 133: "Our assessment finds lower warming assessments for Ethiopia and Philippines" Lower than what? The previous assessment in ref 5? These two countries could then be included in the list at Line 132.

Line 213-217: Here all Figures 1-3 are related to "while preserving a likely change to limit global warming to 2 degrees". The figures are related to both 1.5 degree, 2 degrees and the different temperatures in Fig. 3. This need to be rephrased. To be clearer, maybe include a reference to Fig 3 after these words: "The modelling of the hybrid allocation". It would be useful if you are clearer in the description of the method if you are talking about the 1.5/2 degree scenario and the IAMs results used in Fig. 3. That would help the reader a lot.

Line 258-260: I do not understand this sentence or how Fig. 3 show this. Please explain.

Line 261-263: Consider rephrasing.

Line 313-314: Is aerosol forcing also one of the uncertain model parameters? What is the range of the aerosol forcing?

Other comments:

In the figures, warming of X degrees. Relative to what? What is the reference period? 1750 or 1850 or another period? Please specify.

Related to the scenarios, what is the start year of the mitigation? I presume it is before 2017. Since we are at the end of 2017, how will it affect the results?

Line 195: The aggregation of Kyoto-GHG emissions follows the 'SAR GWP-100'. Why do you still use SAR GWP and not AR5 GWPs?

LULUCF CO₂ emissions are excluded from the analysis, but the other Kyoto GHGs are included? What about aerosols and other short lived components?

Comments to the Figure:

Fig 1a: Hybrid allocation, Equity approach 2: Impossible to read "Aspirational 2°C scenario".

Figure caption line 468: "Small island developing states by their maritime zones." Does this sentence belong to Fig. 1?

Fig 2: Suggest to add "(rest of the world)" below "Other Economies" as (G8+China) is added below "Major economies". Rest of the world are used in the figure legend and in the main text.

Fig 3: I am struggling a bit with this figure and the method used for making it. Below is my comments.

Why do you include the map when the name of the countries are written? Do you need the map? Is it only for visualization? E.g. Mexico and Brazil have the same color, but in the method used to derive the global temperature response they are treated as separate countries, right?

You say in the text that you use 9 scenarios with 2100 warming of 1.2°C to 5.1°C. In the figure caption you write: "Coloured dots represent allocations under each of the 121 global scenarios". And further in the caption on line 484 you write: "coloured lines and patches range over the interpolations of degree one, two and three (Methods)" I am struggling a bit here. Are both the lines and the patches determined by the range over the interpolations of degree one, two and three? The patches are not determined by the individual scenarios (121 global scenarios) consistent with the range of the NDC? I am struggling a bit on the method here, and it would be

useful with more clarification.

Figure 3a is messy around the disks of China, Russia, Canada etc. As I understand, these countries NDCs does not correspond to any of the IAM scenarios. Maybe extend the x-axis with a > 5.1 to make this clearer. And I do not understand the dotted lines from these disks towards the main part of the figure.

It would be very useful if the figure caption of Fig. 3b could be extended. So far only: "Global warming responses (median assessment) following NDC ambitions." And in the figure "Warming in degrees" One or two sentences what have actually been done (maybe related to Line 216-217).

"Other countries": Use the same definition as in the rest of the figures.

Maybe specify in the figure caption that the darkest green color and the darkest red color are outside countries with NDCs that correspond to global mean temperature change below or above the scenario range. Or use an array in the color bar. Now the color bar is rounded at the ends covering several temperature levels.

I like that the figure show the large spread in median warming of the scenario, and as mentioned in the text, the emission change over the period 2010 to 2030 can lead to large spread in the 2100 warming. Maybe you can include two map figures in the supplement with the max and min scenario warming.

Comments to the supplementary:

The title of the supplementary text is not equal to the title of the main article.

Line 34 and 40: grew -> gray

FigS5: How does this map plot show uncertainty?

Supplementary table: Columns D to OH contain the national 2030 emissions levels associated with the global 2100-warming temperature of Row 1 using the 'Hybrid approach'[in GgCO₂eq]. Is the unit correct? It should not be in %?

Reviewers' comments:

Reviewer #1 (Remarks to the Author):

Dear editors:

Upon reflection, I can't quite decide between the two forms of rejection that your guidelines leave me with:

- Reject, but indicate to the authors that further work might justify a resubmission
- Reject outright, typically on grounds of lack of novelty, insufficient conceptual advance or major technical and/or interpretational problems

Rather than strain to choose one or the other, I will attempt to explain myself.

Thanks a lot for providing a review, which greatly helped improving the manuscript. The reviewer has brought a perspective that was missing in this work (and perhaps previous work). We have tried to integrate the reviewer's suggestions, even though some disagreement remains, and reviewed the language, which was unclear and could lead to misunderstandings. We hope that the contribution of this manuscript is now clearer, not as a new equity approach, but as a suggestion for combining dissonant approaches and assessing current NDCs' ambition while countries progress on agreeing on a common definition of fairness.

Comments:

On line 5, we begin with "five effort-sharing allocations of emissions scenarios to achieve the Paris Agreement." This is curious because while the authors repeatedly refer to "five" approaches, and cite the IPCC as their source, the IPCC itself (see page 458 of AR5) actually categorized effort-sharing proposals into six classes.

Thanks a lot for the comment, our manuscript was indeed imprecise. We understand that this critique applies partly to the claims of a previous publication, which itself seeks to follow an IPCC categorisation of effort-sharing approaches. The manuscript discussed presently uses the modelling of this previous publication as a basis and should state the limitations of this modelling. The five effort-sharing approaches used in the first part of this paper correspond to the five effort-sharing categories quantified in the latest IPCC report in Figure 6.28. We here correct the manuscript to avoid any misunderstanding.

We correct the following sentences to:

"using five equity emissions allocations representative of the five effort-sharing categories quantified in the latest IPCC report (Robiou du Pont et al., 2017; Robiou du Pont, Jeffery, Gütschow, Christoff, & Meinshausen, 2016)." Lines 105-107

and

"Using five approaches representative of the five effort-sharing categories quantified in the IPCC-AR5 results in lower temperature assessments of developed countries' NDCs and consequently in higher temperature assessments of other countries' NDCs (Methods and Supplementary Information)." Lines 196-199

and

“The ‘hybrid’ combination of five equity approaches is representative of the five burden-sharing categories quantified in the IPCC-AR5.” Lines 376-377

I don’t want to belabor this point, but I do want to say that something was lost in the translation. It has a lot to do with historical responsibility, which is not just a matter of resource sharing. And it has a lot to do, as well, with the “right to development.” In any case, the issue here is notable because this paper is based upon the authors’ leverage of their set of five approaches, as if it were representative of the entire range of oft-cited, well-supported, and influential effort-sharing / equity / allocation approaches, and indeed as if this set was universally accepted as representing a coherent equity approach. Neither of these conditions, however, is true.

Thanks for this comment. This paper indeed relies on an existing effort-sharing framework (Robiou du Pont et al., 2017), and its contribution is the suggestion of a novel methodology to combine equity approaches. The hybrid approach at the core of this publication can be applied to other equity modelling frameworks. We understand that the effort sharing approaches presented in the IPCC report do not represent an exhaustive list of interpretations of equity or distributive justice. Following this comment, we have added a paragraph detailing the choice of the five effort-sharing categories and have also presented to the sixth category that is discussed but not quantified in the IPCC AR5. We also discuss the existence of other approaches of equity.

“The IPCC-AR5 presented quantifications of emissions reduction following five effort-sharing categories (Chapter 6, Figure 6.28 ref. (Clarke et al., 2014)), which reflect combinations of three underlying equity principles: responsibility, equality and capability. In addition, the IPCC-AR5 presented a category (but did not present a quantification) based on responsibility only as proposed by Brazil in 1997 that derives emissions goals without allocation (IPCC-AR5 Chapter 6, table 6.5, ref. (Clarke et al., 2014; Höhne, den Elzen, & Escalante, 2013)). The historical responsibility of countries for their past emissions is modelled here with the equal cumulative per capita approach (CPC) that uses on historical emissions and population data to calculate countries’ emissions allocations. The GDR approach also uses historical emissions and accounts for countries historical responsibility. Other approaches of distributive justice (for example based on sufficiency approach (Arneson, 2013), or using the Human Development Index) and other metrics (for example accounting for consumption-based emissions or exported emissions) not currently used in the IPCC report or under the UNFCCC are not modelled here but would bring useful perspectives that could be integrated in the ‘hybrid’ approach.” Lines 359-371

The right to development is modelled here through the GDR approach. We added a statement:

“We also provide in the Supplementary Information results under a ‘hybrid’ setup that includes the GDR, which represents a ‘right to development’, and that uses the five effort-sharing approaches.” Lines 169-171

Beginning on line 19, the core idea of this paper is introduced. This idea, while interesting, is frankly odd. Basically, the authors claim that all countries will use the allocation principle that reflects their interests, and that it is, therefore, both illuminating and politically helpful to

grant that they will do so, and then to observe that if they were to also agree to a much stronger global temperature target than they have in fact agreed to, the Paris temperature objectives would come more closely into reach.

The manuscript does not discuss what countries will do and what allocation principle they will use. This manuscript models a hypothetical situation where countries use the allocation principle that reflects their direct interest (excluding the global interest of a global cooperation). The manuscript indicates that the contributions of many countries align with such a situation, but the manuscript does not speculate on the future positions of countries. We have clarified the language throughout the text.

On line 32, they claim that the literature on effort-sharing approaches is “rich.” I do not agree, but cannot today take the time to explain why. I want to stipulate this point, however, because it’s relevant to the central point of the authors’ method – that countries will agree to act as if their disagreements about equity are immaterial to their willingness to accept far more ambitious targets than they have thus far asserted in their NDCs. Rather, I believe that such a willingness, if it ever comes, will necessarily involve a reexamination of the equity challenge.

The term “rich” was referring to the literature available since the 1990s and the 40+ studies mentioned in the IPCC AR5. However, we agree with the authors that the term “rich” is subjective, and have removed it from the manuscript:

“Informed by literature on effort-sharing approaches, the international community has long discussed the operationalization of equity following the UNFCCC principle of Common But Differentiated Responsibilities and Respective Capabilities (CBDR-RC) to drive national emissions allocations(UNFCCC, 2012; Winkler & Rajamani, 2014b).” Lines 28-31

We agree with the second point raised here regarding the necessary re-examination of the equity challenge. With this paper, we are trying to make a contribution to the debate on distributive justice applied to emissions allocations. This paper shows that the absence of consensus on burden sharing does not imply the absence of an equity-based metric to assess the ambition of current mitigation pledges, as they are framed under the UNFCCC. We agree that the equity challenge should not be restricted to its current framing under the IPCC and UNFCCC, even though this manuscript is not tackling that issue. Finally, we agree that the equity challenge stretches beyond mitigation and think that work is urgently needed to link with adaptation and loss and damage. We have expressed this position in the manuscript:

“Despite claims that discussions of justice are “irrelevant or dangerous in a post-Paris world” (Robert Keohane quoted in ref.(Klinsky et al., 2017)), equity is fundamental for climate policy research(Dooley, Gupta, & Patwardhan, 2018; Klinsky et al., 2017) and scientific analyses on equitable burden-sharing is influential on the UNFCCC processes(Mace, 2016). However, the absence of agreement on an unanimous operationalisation of the CBDR-RC should not be used as an excuse for inaction(Winkler & Rajamani, 2014b) and should not leave the international community without a metric reflective of current agreements to assess the ratcheting-up process.” Lines 74-79

On line 47, we get the citation that supports the claim that countries “generally support interpretations of distributive justice that justify their negotiating positions.” I will not belabor this point, but I will say that citation 22 is an interesting but polemical paper, while 23 is a blithe reference to an AR5 chapter that could as easily be cited to support the opposite claim. Do note, too, that the situation here is more complex than the authors aver. See for example Winker et. al., Countries start to explain how their climate contributions are fair: more rigour needed, Int. Environmental Agreements, accepted 27 October 2017, which offers empirical evidence “that countries do not necessarily use equity indicators that favour their equity argument.” This paper is also interesting as evidence that the equity debate is still in early stages (hardly “rich”) and that, to the extent that it is reflected in the NDCs, it is not reflected in a manner that is captured in the limited set of approaches that is the basis of this paper.

For the quick review of the stakes here, see AR4, WG4, Chapter 4, especially 4.6.2.1

Thank you for this perspective. We do agree with the comment that citation 22 is polemical and it actually argues to some extent against the need for discussing equity at the negotiations. We do not support this message but wanted to bring an alternative narrative, which was brought to us during the review process of another manuscript. We can remove it if necessary. Regarding reference 23 (the IPCC AR5), we based our interpretation on the following quote (IPCC AR5, Chapter 4, Section 4.6.2.1, page 317) “Because there is no absolute standard of equity, countries (like people) will tend to advocate interpretations which tend to favour their (often short term) interests (Heyward, 2007; Lange et al., 2010; Kals and Maes, 2011)”.

We understand that many countries show leadership and have made pledges that go beyond any of the quantifications of equity presented here (mostly Least Developed Countries, as can be seen at: <http://paris-equity-check.org/> which uses the same effort-sharing framework as this study). We amended the sentence to better reflect the IPCC wording and include the reference suggested in the review:

“While some countries do not use indicators that favour their equity argument in their communication(Winkler, Höhne, Cunliffe, & Maria, 2017), a common definition of equity is unlikely to be adopted since countries generally tend to support interpretations of distributive justice that best serves their self-interest and likewise justify their negotiating positions(Averchenkova, Stern, & Zenghelis, 2014; Fleurbaey et al., 2014; Lange, Löschel, Vogt, & Ziegler, 2010; Tørstad & Sælen, 2017). Furthermore, the commitments of many major emitters, including developed countries who committed to take the lead in reducing emissions and mobilizing finance to support mitigation in developing countries(UNFCCC, 2015), do not match concepts of equity that they publicly supported(Robiou du Pont et al., 2016).” Lines 42-48

I found the paragraph beginning with line 48 to be unhelpful. It touches on a number of issues, but it does not raise what would seem (given the authors’ overall strategy) to be the main point, which is the assumption that nations simply take equity positions that suit their interests within a decision space defined by author’s five approaches, and will hold those position steady while adjusting to the demands of a massive and extremely challenging increase in global ambition.

The para that begins on line 63 is a critical one. First of all, I do not agree with the authors’ claim that their exercise here is not normative. In fact, I find this claim to be so unsupportable

that it makes me lean toward the “reject outright” recommendation. In a nutshell, I believe that their method is strongly skewed in favour of wealthier, higher-emitting countries.

Consider the sentence that begins on line 64, a sentence that strikes me as problematic, if not simply wrong. Briefly, and despite citations 26 and 27, the issue is not merely “national preference.” The issue is also “relative fair shares” – the real core of the effort-sharing debate – and it inevitably raises normative challenges that should not be perfunctorily dismissed, especially not by way of blithe claims to be avoiding normative judgment.

Ultimately, what is at stake here is a neo-realist claim that we should finesse normative judgment. In fact, the claim that countries will do just this when faced with the overwhelming imperatives of climate stabilization. In effect, the authors are arguing that the “agreement to disagree on equity concepts to achieve a common temperature goal” is key to finding a path forward. It may be – I don’t claim to know – but I do not find the argumentation here, the method, to be particularly convincing. In particular, I do not see any defense for the claim made at the end of this para, that the authors’ “hybrid approach” can inform the PA’s ratcheting’s mechanism. Have I missed it?

Thanks for raising this raising this critical point. We stated that this hybrid approach is not normative in the sense that it can let each country pick the most favourable modelling of equity, and could let a country pick any modelling of equity. However, we agree with the reviewer that any modelling applicable to all is normative in some sense. We have removed the term normative:

“The ‘hybrid’ approach does not constitute an ‘equitable’ operationalization of the CBDR-RC principle where all countries seek to maximize ‘absolute gain(Keohane, 1984)’ by agreeing on a common approach of equity.” Lines 70-72

We agree with the authors that progressing towards an agreement on equity requires dealing with normative challenges. As a first step, this manuscript explores a situation in the absence of progress towards an agreement on equity, which serves as a warning against such a situation.

As a second step the manuscript introduces the ‘hybrid’ approach, which is not presented as an approach itself, as stated above. The manuscript does not present the hybrid approach as the “key to finding a path forward”. Rather it is presented as an alternative while progress is made on a common understanding of fairness. The current absence of a commonly agreed equity metric to assess NDCs should not leave observers, experts and courts without a tool reflective of current agreements, to assess countries’ ambitions.

Thanks to the reviewers’ comments, we have clarified the contribution of this manuscript and reshaped entirely the paragraph:

“The ‘hybrid’ approach does not constitute an ‘equitable’ operationalization of the CBDR-RC principle where all countries seek to maximize ‘absolute gain(Keohane, 1984)’ by agreeing on a common approach of equity. Rather, it reflects national preferences for ‘relative gain(Krasner, 1991)’, i.e. a country’s inclination to measure the fairness of its contribution to the global mitigation effort by looking at other countries’ efforts, rather than to domestic indicators alone. Despite claims that discussions of justice are “irrelevant or dangerous in a post-Paris world” (Robert Keohane quoted in ref.(Klinsky et al., 2017)), equity is fundamental climate policy research(Dooley et al., 2018; Klinsky et al., 2017) and scientific analyses on equitable

burden-sharing is influential on the UNFCCC processes(Mace, 2016). However, the absence of agreement on an unanimous operationalisation of the CBDR-RC should not be used as an excuse for inaction(Winkler & Rajamani, 2014b) and should not leave the international community without a metric reflective of current agreements to assess the ratcheting-up process. The multiplicity of equity concepts results in a wide range of emissions allocations for countries(Pan, Elzen, Höhne, Teng, & Wang, 2017; Robiou du Pont et al., 2017) and regions(Clarke et al., 2014) that is sometimes used as an uncertainty range by non-experts. In a recent climate case, the District Court of The Hague ruled(Schiermeier, 2015; The Hague District Court, 2015) that the Dutch government has to reduce 2020 emissions by at least the least-ambitious end of the range recommended by the IPCC-AR4 for the Annex I country group based on multiple equity allocations from 16 studies(Gupta et al., 2007). The court did not pick an approach of equity and ruled for the minimum effort consistent with international treaties in light of commonly reviewed science. While the multiplication of climate litigations cases against governments(Sabin Center for Climate Change Law, 2018) contributes to the ratcheting-up process, systematic court decisions that governments must follow the least-ambitious end of an equity range is unlikely to achieve the Paris Agreement. As a first step, this manuscript models such a ‘bottom-up’ situation where each country follows the least-ambitious effort-sharing approach representing the quantified IPCC categories. As a second step it models the ‘hybrid’ approach to represent the current compromise where each country chooses an equity approach to determine its effort but cannot directly use that approach to influence other countries’ effort. Overall, this study presents an operationalization of the current ‘agreement to disagree’ on equity concepts to achieve a common temperature goal. The ‘bottom-up’ allocation of emissions under the ‘hybrid’ approach is consistent with the pledge-and-review nature of the Paris Agreement and its mitigation goals and provides a metric for the ratchetting-up process.” Lines 70-97

Line 77. Again, the reification of the “five IPCC effort-sharing categories.” (Note that they have been promoted).

Thanks for noticing this imprecision, the text has been modified to:

“using five equity emissions allocations representative of the five effort-sharing categories quantified in the latest IPCC report(Robiou du Pont et al., 2017, 2016).”
Lines 105-107

What I find particularly interesting (line 82) is the admission here that the CER approach (popularly known as “grandfathering”) is widely considered to be unfair, because just a bit later (line 117) we are told that “current national emissions ratios, at the root of the controversial CER approach, also influence near-term allocations of all other approaches.” Which is to say that CER itself is tossed out along the way (and I applaud this move), though GDR is tossed out along with it, because it is “redundant.” The result is that EPC, CPC and CAP, a group that is heavily influenced by CER (unfair CER) is taken as the core of the assessment methodology, going forward.

Thanks for raising this important point. By definition, current emissions levels of countries are having a near-term influence on any continuous emissions allocation, whether or not it is compensated later (as the historical responsibility based CPC or GDR approaches). Only a discontinuous step function would avoid this near-term influence. The modelling framework used here features continuous allocations (including the GDR). The CER approach supposes the strict preservation of current emissions ratio in the future, which no other approach supposes. Therefore, it is incorrect to say that the EPC, CPC and CAP are influenced by the CER which is modelled independently. The results of the EPC, CPC and CAP approaches are consistent with the quantification presented in the IPCC in 2030 and 2050 for the corresponding categories (summarising over 40 studies), please see supplementary information of (Robiou du Pont et al., 2017, 2016). While other modelling frameworks could certainly be used under the hybrid approach, we choose to pursue with this set-up that is consistent with a wide range of the literature. We clarify the exclusion of the GDR from the warming metric in the main text of the manuscript in the next comment.

Line 119. By the way, no convincing explanation of why the GDR approach is “redundant” is ever given. What is said is that “The GDR approach blends the concepts of historical responsibility and capability covered by the CPC and CAP approaches, respectively,” but this misses the whole point of the framework, as I understand it, which is NOT a per-capita framework, but rather one that is based upon “the right to development,” and which treats that right in explicitly progressive terms. In fact, as I understand it, modeling GDR – which by the way was long ago renamed the “climate equity reference framework” – by way of a parameterization that reduces it to a per-capita system is explicitly to caricature it. Which is to say that it is NOT redundant with EPC, CPC, or CAP and that to paint it as being so just raises the question of how GDR was modeled by the authors in the first place. In any case, given that the authors are leaning so heavily on the supposedly canonical nature of their five categories, the further simplification of the “five” into “three” seems a real methodological stretch.

Thanks for reminding that the GDR approach aims at depicting a right to development which the other approaches do not. The redundancy was an equivocal term (now removed) referring to the indicators used by the GDR approach: historical emissions and GDP per capita.

Thanks as well for raising the lack of clarity. The goal of this manuscript is not to make a philosophical contribution on the categorisation of equity. Rather, it proposes a combination of indicators.

Our choice is to convey three indicators (historical emissions, per capita emissions, and GDP per capita) that represent the three equity principles of Table 6.5 of the IPCC AR5: responsibility, capability and equality, which also are also suggested in the notion of CBDR-RC.

We have included results with the GDR approach included in the ‘hybrid’ approach in the supplementary information. The GDR approach uses reference emissions, or business as usual emissions. These hypothetical trajectories depend on contentious assumptions by each country regarding what would have happened in the absence of effort or climate measures. It seems impossible for countries to agree on such trajectories¹.

¹ see for example page 115 of <http://www.climatechangeauthority.gov.au/files/files/Target-Progress-Review/Targets%20and%20Progress%20Review%20Final%20Report.pdf>

Furthermore, while the wealth threshold of the GDR is a great idea in theory at the per capita level, it results in greater emissions allocations to unfair countries (low Gini index) compared to a fair one (high Gini index). Since this study focusses on national emissions allocation, we find this characteristic inconsistent with its fairness goal at the international level.

The inclusion of GDR approach, as it is modelled here using national business-as-usual emissions projections downscaled from RCP8.5, disfavours many developing countries, compared to using only the CPC, EPC and CAP approaches in the ‘hybrid’ setup. We updated the text to reflect these points:

“The modelling of the GDR approach relies on business-as-usual (BaU) emissions that countries do not mutually recognize. The BaU emissions used here are downscaled from RCP8.5 resulting in allocations substantially higher than other allocations for Eastern-European countries or Australia. Compared to the ‘hybrid’ setup presented in Figure 3, the inclusion of the GDR approach in the ‘hybrid’ setup results in a lower temperature assessment in favour of the NDCs of Eastern-European countries, Australia, and South Africa, and higher temperature assessment disfavours India, Brazil, Mexico and Indonesia (Supplementary Figure 8).” Lines 390-396

We have modified the manuscript to mention this more clearly, and referred to the supplementary information’s results, which include the GDR in the hybrid approach. Regarding the naming, we have modelled the GDR approach as described (Baer, Fieldman, Athanasiou, & Kartha, 2008; Kemp-Benedict, 2010; M. Meinshausen et al., 2015) and used the corresponding naming.

“Each country is attributed the equity approach with the least stringent 2030-emissions, excluding the CER and GDR approaches (Methods). Current national emissions ratios, at the root of the CER approach(Caney, 2009; Peters, Andrew, Solomon, & Friedlingstein, 2015; Robiou du Pont et al., 2017), also influence near-term allocations of any continuous allocations, reflecting some of the national circumstances mentioned in the Paris Agreement(UNFCCC, 2015). The GDR approach is based on historical emissions and GDP per capita that are covered in the CPC and CAP approaches, respectively. The GDR approach relies on hypothetical projections of Gini indices and emissions (here downscaled from RCP8.5, see methods) that can lead to large variations in emission allowances. These variations can be more determined by counterfactual input assumptions than by the effort-sharing principles themselves (<http://paris-equity-check.org/>). We therefore present a synthesis of the EPC, CPC and CAP approaches in a ‘CBDR-RC hybrid’ setup that enables the derivation of an NDC-warming assessment tool ‘applicable to all’. We also provide in the Supplementary Information results under a ‘hybrid’ setup that includes the GDR, which represents a ‘right to development’, and that uses the five effort-sharing approaches.” Lines 158-171

The real point in all this, I suppose, is that the “hybrid” method that the authors are speaking for in this paper, while it may be interesting, is not well served by being erected upon the fragile foundation of the “five approaches.” As they themselves note (line 145), results “strongly depend on the set of effort-sharing approaches included in ‘hybrid’ approach modeling.” Which is exactly why, to properly evaluate the usefulness of the approach, it would be nice to see it modeled in a fair way, one that in fact takes account of all the equity

challenges discussed in AR5, rather than just the ones that remain after the cascading reductions that seem to lie behind this paper.

Firstly, the manuscript states that the hybrid approach does not constitute a modelling of a fair situation that can be modelled “in a fair way”. However, we agree with the reviewer in that the hybrid approach combines equity approaches, which should be modelled in as fair ways as possible, given their narratives.

We agree with the reviewer that the modelling framework from (Robiou du Pont et al., 2017), as other literature, implies simplifications and interpretations of equity and cannot take account of all the equity challenges discussed in the AR5. However, these approaches align with the quantifications presented in the IPCC, which itself is based on more than 40 studies. This modelling framework represent a well-documented and peer-reviewed basis for the hybrid approach. This hybrid approach can be updated in the future with other modelling or categorisation of equity.

Finally, let me note (line 152) that the authors statement that “The emissions trajectories quantified in this study offer an alternative effort-sharing scheme that reconciles the bottom-up pledging architecture of the Paris Agreement’s with its mitigation goals. The resulting equity-based metric acknowledges countries’ self-interested positions and does not hypothesize an international agreement on a unique approach of equity.” Let me, in particular, say that I find the logic here, or at least the desire, to be quite understandable. Self-interest, after all, is a major factor in all this. Having said this, let me add that I see no reason to believe that this paper, as it stands, will be helpful. For one thing, it cleans up the “five” by dismissing the CER approach, but carries the CER idea forward under other names. And it caricatures and problematically dismisses a promising approach (GDR) as a per-capita approach, when in fact its great strength is that it is explicitly not a per-capita approach. Not, by the way, that I think GDR is without its problems. It’s hardly the answer to the riddle of history, but still. This is all supposed to be about fairness, right?

Thanks for this summary and the understanding of the logic of modelling self-interest. It is about fairness, but this manuscript does not seek to define fairness, only to offer a quantitative framework to combine multiple views. The information provided in this manuscript is novel and adds to the discussion on operationalising fairness, whether it is seen as a potential way forward or as a temporary metric while discussions continue on finding an equity principle applicable to all.

We have removed the CER from the warming metric in accordance with the position of the reviewer but disagree that the CER is carried in other names simply because we model continuous trajectories. We have better justified the exclusion of the GDR from the warming metric (which is by the way also a continuous trajectory influenced by current emissions levels) given the intent of the study to reflect basic indicators of equity. We have presented results with the inclusion of the GDR in the supplementary information and discussed its influence.

Overall, we have also reviewed the whole manuscript to improve clarity and language thanks to reviewers’ comments.

Reviewer #2 (Remarks to the Author):

Peer review of “Temperature assessment of the bottom-up Paris emissions pledges”

Reviewer: Daniel Horen Greenford

Disclaimer: Don't mean to be rough with comments so if it seems I'm harsh at times, it's only me being terse and opinionated. Things that I feel especially strong about have been emphasized and repeated. My angle is one of ethics side so gave most criticism and comments on that, and less weight on the technical side, which I hope other reviews have addressed more thoroughly. Hope these comments are helpful to your work! Good job! Lot's of great insights here! Hope this gets out soon and there's good media coverage of it. Feel free to contact me for further explanation or to talk over some points. I have chosen to disclose my identity, in keeping with the journal's philosophy of transparency. Apologies for the delay, busy time of year! Also sorry if this is a bit of a mess and for some repetition. Think I got most of what I had to say out in this sprawl of points. Keep up the great work.

General comments:

Overall excellent work. The paper is very interesting and informative, but needs to be tightened up for clarity. Understandable but a bit cumbersome and flow not as illustrative of main themes as it could be.

Thanks a lot for these comments, the thorough review and the suggestion. We very much agree, the lack of clarity regarding the goal and usefulness of this manuscript was raised by other reviewers. We have restructured the narrative, the introduction and conclusion of the manuscript to convey the points of this manuscript clearly.

I have some suggestions as how to make it more accessible to a broader audience and if I get what points you're trying to make, how to make them jump out more clearly and explicitly. I also am unsure of the best way to structure the narrative here. **In my opinion, the most interesting result is how much warming would be if all countries followed paths of analogous ambition to a chosen country.** If I understand correctly this is what you show when you talk about what “prolongating bottom-up NDCs of a given country would lead to.” Correct me if I'm wrong, but if this is what you've down, that's the most important insight for me.

Thanks for your suggestion. That is correct.

First, we model a pathway consistent with a situation where all countries follow the most advantageous of five effort sharing approaches. We find that this pathway that represents a self-interested selection of effort-sharing approaches by countries aligns with aggregated NDCs in 2030 and which exceeds 2°C of warming.

As a second step, we allocated the emissions of a range of pathways with warmings from 1.2°C to 5.1°C, attributing to each country the least-stringent off three equity approaches. This provides a metric that answers the question: **how much warming would be if all countries followed paths of analogous ambition, based on the least-stringent effort sharing approach, to a chosen country?**

I think the more technical analysis of showing what temperatures lead to successful hybrid allocations is also super interesting but less important insofar as it is pretty tough to see countries following this approach (not to criticize too severely the applicability of your work to realpolitik because pretty much all work in this field, including my own is likely never making into actual policy), so the real contribution is insight into how to conceptualize ambition and how far countries actions are from the ideal. I think the other realization that targets would have to be much lower if countries would all pick the effort-sharing method that benefited them the most is interesting but the way you frame it seems to suggest that countries would be convinced by this, and I'm pretty skeptical of this. It is a bit like trying to tell your son that he can have whatever allowance he wants but you'll only give him \$5 and he has to wait for inflation to make it \$50 for example. Maybe not a precise analogy but the same logic applies of making someone think they are choosing something/winning but in reality you're just telling them what they want to hear and then giving them the share that's due to them in first place (not exactly how your hybrid works, I know, but pretty close, looking at).

Thanks for raising the question of the usefulness of the hybrid approach. The hybrid approach is indeed not based on hypothetical inflation. It represents an alternative distribution of a given quantity and does not magically extend it. The combination of equity is not designed as an approach that countries should agree to follow. It is designed to provide a metric, reflective of current agreements, to assess the ambition of countries despite a lack of consensus on equity. Progress on the definition on a common definition of equity is still encouraged and would enable great progress towards achieving all the current global goals.

One thing that bothers me, and I'm not certain I'm interpreting this correctly: Is your "hybrid approach" just a way of tricking countries into thinking they have agency? Giving them the false impression that they're choosing what's best for them (they are to begin with, we assume, and not even picking the least stringent of equity methods for current NDC derivations) when in actuality you are just tweaking the temperature target so it works. Kind of like sleight of hand to make everyone feel like they're coming out on top. If this was proposed, would nations really think it was fair? What would you be telling them? If I'm reading this right, I see "so everyone pick the metric that gives you the biggest share of the carbon budget, then we lower the temperature target such that we get to the stated global objective." If I were a nation who was told that, I'd be scratching my chin a bit and ask, "hey are you sure this is a winning deal for me, aren't you just leading me to believe it is but then tweaking the target so it works and hence cutting my share just the same?"

Thanks for raising this. The manuscript does not intend to trick anyone. This study suggests an alternative distribution of a given amount of emissions rights. In that perspective, not all countries can be better off quantitatively. The possibility for each country to adopt the least-stringent approach logically comes at the cost of the overall target applied to the effort-sharing approaches. We have clarified the purpose and the usefulness of the study with a completely reshaped paragraph as follows:

"The 'hybrid' approach does not constitute an 'equitable' operationalization of the CBDR-RC principle where all countries seek to maximize 'absolute gain(Keohane, 1984)' by agreeing on a common approach of equity. Rather, it reflects national preferences for 'relative gain(Krasner, 1991)', i.e. a country's inclination to measure the fairness of its contribution to the global mitigation effort by looking at other countries' efforts, rather than to domestic indicators alone. Despite claims that

discussions of justice are “irrelevant or dangerous in a post-Paris world” (Robert Keohane quoted in ref.(Klinsky et al., 2017)), equity is fundamental climate policy research(Dooley et al., 2018; Klinsky et al., 2017) and scientific analyses on equitable burden-sharing is influential on the UNFCCC processes(Mace, 2016). However, the absence of agreement on an unanimous operationalization of the CBDR-RC should not be used as an excuse for inaction(Winkler & Rajamani, 2014b) and should not leave the international community without a metric reflective of current agreements to assess the ratcheting-up process. The multiplicity of equity concepts results in a wide range of emissions allocations for countries(Pan et al., 2017; Robiou du Pont et al., 2017) and regions(Clarke et al., 2014) that is sometimes used as an uncertainty range by non-experts. In a recent climate case, the District Court of The Hague ruled(Schiermeier, 2015; The Hague District Court, 2015) that the Dutch government has to reduce 2020 emissions by at least the least-ambitious end of the range recommended by the IPCC-AR4 for the Annex I country group based on multiple equity allocations from 16 studies(Gupta et al., 2007). The court did not pick an approach of equity and ruled for the minimum effort consistent with international treaties in light of commonly reviewed science. While the multiplication of climate litigations cases against governments(Sabin Center for Climate Change Law, 2018) contributes to the ratcheting-up process, systematic court decisions that governments must follow the least-ambitious end of an equity range is unlikely to achieve the Paris Agreement. As a first step, this manuscript models such a ‘bottom-up’ situation where each country follows the least-ambitious effort-sharing approach representing the quantified IPCC categories. As a second step it models the ‘hybrid’ approach to represent the current compromise where each country chooses an equity approach to determine its effort but cannot directly use that approach to influence other countries’ effort. Overall, this study presents an operationalization of the current ‘agreement to disagree’ on equity concepts to achieve a common temperature goal. The ‘bottom-up’ allocation of emissions under the ‘hybrid’ approach is consistent with the pledge-and-review nature of the Paris Agreement and its mitigation goals and provides a metric for the ratcheting-up process.” Lines 70-97

We have also reshaped the conclusion to better guide the reader on the usefulness of the results:

“The UNFCCC and Paris Agreement do not indicate how to operationalize the CBDR-RC and countries supposedly build their pledge based on their own understanding of fairness, often self-interested(Averchenkova et al., 2014; Fleurbaey et al., 2014; Lange et al., 2010; Tørstad & Sælen, 2017). As no single definition of fairness emerges from current NDCs(Winkler et al., 2017), we quantify a combination of concepts of equity that reconciles the bottom-up ‘pledge and review’ architecture of the Paris Agreement’s with its top-down mitigation goals. The resulting metric provides a temperature assessment of countries’ NDCs under the current regime and can inform the ratcheting-up process, without hypothesizing an international agreement on a single approach of equity. This ‘hybrid’ combination of countries’ least-stringent equity approaches is also relevant to climate cases where the court can only rule for the least-ambitious end of an equity based range(Urgenda, 2017). We find that most of the least-developed countries have NDCs consistent with the Paris Agreement goals. However, the NDCs of most developing countries appear insufficient, as those of developed countries who agreed to take the lead in reducing

emissions and mobilizing finance to support mitigation in developing countries(UNFCCC, 2015).”Lines 203-215

Of course the share derived with your hybrid approach may be bigger for certain countries and smaller for others (do you have some numbers showing how much the difference is between a chosen equity metric and the hybrid outcome? This is figure 2 panel b, right? Showing the difference between hybrid and average, but does average include CER?)

Yes, it includes CER for technical comparison between the average over the five effort-sharing approaches presented in (Robiou du Pont et al., 2017) and the ‘hybrid’ allocation using the same five effort-sharing approaches. We did not provide allocations under individual effort-sharing approach, these can be found in the supplementary material of (Robiou du Pont et al., 2017).

So the top-lining messaging is very important and I think this can be clearer. My main two takeaways from this paper are:

1) This is what temperature rise would be if all countries followed analogous ambition to a chosen/specified country’s ambition, e.g. if all countries acted in a manner of analogous ambition to China, we’d warm the planet by 5.1degC. Am I interpreting this correctly? This is in my option the most important takeaway and should be more plainly worded for a more general audience, try to be clear in the opening paragraph and abstract for sure. A lab-mate who is in climate adaptation did not get that from reading the abstract, so not accessible enough. I would suggest something like (instead of line 24-26): “We also (I would say ‘also’ since this is another significant finding in your study and it seems to be lumped into the previous part but I’m confused as to whether the NDC analysis is the same as the hybrid result with higher temperature thresholds) explore what temperature would be induced when using each country’s NDC as the specified level of ambition for a bottom-up approach to global mitigation efforts. In other words, we find that if all countries acting with a level of ambition of a single chosen country, emissions would result in a given amount of warming. We find that the NDCs of India ... respectively.” I think the added sentence or two of explanation is necessary.

Thanks, we have rewritten the introductory paragraph to reflect these suggestions and match the requirements of the Nature Communication Guide to Authors:

“Under the bottom-up architecture of the Paris Agreement, countries pledge Nationally Determined Contributions (NDCs). Current NDCs individually align, at best, with divergent concepts of equity and are in aggregation inconsistent with emissions scenarios to achieve the warming thresholds of the Paris Agreement. We find that the global 2030-emissions of current NDCs match the sum of each country adopting the least stringent of five effort-sharing allocations of greenhouse-gas (GHG) emissions scenarios to achieve the Paris Agreement. We estimate that extending such a self-interested ‘bottom-up’ aggregation of equity might lead to a median (>50% likelihood) warming of 2.3°C in 2100. We find that ratcheting-up the warming goal of all the individual ‘bottom-up’ effort-sharing allocations to hypothetical levels of 1.1°C and 1.3°C could achieve the Paris Agreement’s warming thresholds of 1.5°C and well below 2°C, respectively. This new ‘hybrid’ allocation that reconciles the ‘bottom-up’ pledging nature of the Paris Agreement with its ‘top-down’ warming threshold, provides a temperature metric to assess NDCs. When

taken as benchmark by other countries, the NDCs of India, the EU, the USA and China lead to warmings of 2.6°C, 3.2°C, 4°C and over 5.1°C respectively, under the current bottom-up regime.” Lines 10 to 23

2) If all countries pick the effort-sharing approach that benefits them the most, I.e. gives them the largest share of the remaining carbon budget / amount of allowable cumulative emissions, then the world would need to pursue a global warming threshold target lower than the Paris Agreement’s window, and find that in order to reach 1.5 and 2 degrees, the “aspirational” target would have to be lowered to 1.1 and 1.3 degrees.

Yes, and that indicates the ‘cost’ of disagreement. We included the sentence:

“The resulting temperature gaps between the aspirational target and effective warming reflect the necessary strengthening of global temperature aspirations to compensate for the disagreement on effort-sharing (Figure 1).” Lines 129-131

Also, does your analysis based on Climate Action Tracker’s work use a similar approach?

No, the aggregation in CAT is different and not peer reviewed. The CAT excludes the extrema-allocations for each country (these extrema-allocations can be based on different equity approaches for different countries). The approach is interesting but is not based on a narrative. We find the self-interested attribution of equity more relevant to the current state of international negotiations.

Is your estimate of the impact of abandoning the Clean Power Plan novel or just quoting their work? If the latter, it doesn’t belong in the opening paragraph, but it seems like you take their initial work and then quantify the warming impact so this is pretty interesting.

We take the emissions estimate from the Climate Action Tracker (what would the US emissions level in 2030 if the clean power plan is repealed: 6.74 GtCO_{2e}). Using the temperature metric developed in this manuscript, we translate this emissions level projected by the CAT into a temperature assessment. The CAT itself assesses the ambition of the USA as critically insufficient, leading to a warming exceeding 4°C degrees.

Curious as to how they do it, and generally how you and Meinhausen have come up with pathways for NDCs based on somewhat ambiguous descriptions of policy. I intend to read more about this, but generally very important and impressive work that’s invaluable to the community so thanks very much for this!

Thanks. We are also very thankful to be able to use this work in this manuscript.

I can’t stress this enough: I would NOT include CER in the “average” since it is clearly not equitable. Please EXCLUDE CER from your “average”, so only use the 4 methods: GDR, CAP, EPC, CPC. And to be an ethical purist, I would argue that equal per capita is NOT an equitable outcome, it’s a compromise at best, since it disregards historical emissions, and the historical overuse of the atmosphere or climate debts or however you prefer to frame it, it still is not a truly just outcome. It’s a “let’s wipe the slate clean and start from here outcome.” And what are/is the convergence date(s)? Is that where the range for EPC comes from or

that's just the range of uncertainty from the NDCs, right? In your previous work you use a 30 year convergence period. That the same parameter here?

Yes, the parameterisation comes from (Robiou du Pont et al., 2017). The CER is discarded from the metric modelled to assess NDCs' ambition. However, we choose to show it in Figure 2, and include it in the average in order to show how different the average is from the bottom-up or hybrid approach. Doing so, this comparison enables the reader to put this manuscript in the perspective of the (Robiou du Pont et al., 2017) study. This technical comparison does not imply that countries' NDCs should be assessed against the average. Instead, the metric defined later in the text to assess NDCs ambitions (Figure 3) excludes the CER. We have included the reviewer's suggestion and added Supplementary figure 6b that compares the hybrid combination to the average using the CAP, EPC and CPC approaches (so excluding the CER as the reviewer suggests).

CPC on the other hand does take historical emissions into account so I think that's just. GDR does blend history fault with present responsibility based on available wealth (what they call "capacity"), and is very nuanced and I think very equitable regardless of most parameterization, and I'm not too family with CAP but if it's similar to Kemp Benedict's method for deterring "capacity" I support it. If it's a simpler version of that, it seems just conflating financial capacity with GDP, it's a much worse metric than GDR's capacity so even if it's an IPCC standard, it doesn't add anything. I understand the appeal/benefit of going with established categories and methods so I would stick with that, but I have to voice my grievances with this effort sharing approaches even if this is not the paper to debate them. That said, I still think you have the creative license to exclude ones you think are bunk or not equitable so if you find my arguments convincing please do dispense with EPC and maybe CAP and just keep GDR and CPC. Or as a compromise, have another "average" that is only those or for a broader one that isn't pure equity and take the average of the four. Still have to be hard on the no CER "average" being represented, but if you do want to include the original one with it, be clear about it. It is analogous to something like Raupach and colleagues' "blended" equity setting (Nature Climate Change, 2014), which is the middle ground between totally unfair and climate justice since it's just the average of CPC and CER. The average of the five will be much more equitable than that since there are more equitable allocations than non-equitable ones to shift the average. I actually tried taking the data from your Nature Climate Change paper and removed CER to see how much of a difference it made and it's not huge but still significant.

Thanks for the suggestions and for taking the time to do data analysis. The goal of the manuscript is not to convey the views of the authors on fairness. The contribution of this study is to suggest a new combination of various approaches of equity. The inclusion of the CER in Figure 2 serves a comparison between the average (or blending) used in the literature, and the hybrid approach presented here. Neither the average, nor the CER, are used to assess the ambitions of countries' NDCs. We amended the manuscript to clarify the selection of the three equity approaches to build the hybrid metric. Our choice is to convey three indicators (historical emissions, per capita emissions, and GDP per capita) that represent the three equity principles of Table 6.5 of the IPCC AR5: responsibility, capability and equality, which also are also suggested in the notion of CBDR-RC.

We have included the GDR approach in the 'hybrid' approach and its results in the supplementary information. The inclusion of GDR approach, as it is modelled here using national business-as-usual emissions projections downscaled from RCP8.5, disfavors many

developing countries, compared to using only the CPC, EPC and CAP approaches in the 'hybrid' setup. We updated the text to reflect these points:

“The modelling of the GDR approach relies on business-as-usual (BaU) emissions that countries do not mutually recognize. The BaU emissions used here are downscaled from RCP8.5 resulting in allocations substantially higher than other allocations for Eastern-European countries and Australia. Compared to the 'hybrid' setup presented in Figure 3, the inclusion of the GDR approach in the 'hybrid' setup results in a lower temperature assessment in favour of the NDCs of Eastern-European countries, Australia, and South Africa, and higher temperature assessment disfavouring India, Brazil, Mexico and Indonesia (Supplementary Figure 8).” Lines 390-396

We have modified the manuscript to mention this more clearly, and referred to the supplementary information, which includes the GDR in the hybrid approach.

Regarding the naming, we have modelled the GDR approach as described (Baer et al., 2008; Kemp-Benedict, 2010; M. Meinshausen et al., 2015) and used the corresponding naming.

“Each country is attributed the equity approach with the least stringent 2030-emissions, excluding the CER and GDR approaches (Methods). Current national emissions ratios, at the root of the CER approach(Caney, 2009; Peters et al., 2015; Robiou du Pont et al., 2017), also influence near-term allocations of any continuous allocations, reflecting some of the national circumstances mentioned in the Paris Agreement(UNFCCC, 2015). The GDR approach is based on historical emissions and GDP per capita that are covered in the CPC and CAP approaches, respectively. The GDR approach relies on hypothetical projections of Gini indices and emissions (here downscaled from RCP8.5, see methods) that can lead to large variations in emission allowances. These variations can be more determined by counterfactual input assumptions than by the effort-sharing principles themselves (<http://paris-equity-check.org/>). We therefore present a synthesis of the EPC, CPC and CAP approaches in a 'CBDR-RC hybrid' setup that enables the derivation of an NDC-warming assessment tool 'applicable to all'. We also provide in the Supplementary Information results under a 'hybrid' setup that includes the GDR, which represents a 'right to development', and that uses the five effort-sharing approaches.” Lines 158-171

Title:

Abstract

Main messages/take homes

If I get this (sorry for being repetitive in the comments by the way, helps me rephrase the ideas and clarify my thinking and you can correct me at any step where I'm misreading the study)

The takeaways are:

1) What would the temperature target need to be to meet 1.5 or 2 deg C if all countries picked the equity principle (including CER) that gives them the largest share of remaining emissions.

Correct. We include the CER at this descriptive stage given that this approach is in the narrative of many countries when national circumstances are mentioned, and given that many developed countries align implicitly with that approach. The inclusion of the CER is therefore used to reflect and extend the current situation and evaluate the corresponding warming. The absence of progress on a common understanding of equity would imply a warming overshoot.

2) What would the rise in global temperature be if all countries acting in accordance with analogous level of ambition to a chosen country (or policy change).

Correct. The metric is based on the hybrid approach, which is based on CAP, EPC and CPC.

General points about study:

- I'm interested to know what the hybrid approach looks like without CER. Can you just exclude it as an option? It really isn't part of the suite of equitable choices and so let countries "pick" any other approach, just not grandfathering.

Certainly, the Supplementary Figure 8 in the Supplementary Information show the application of the hybrid approach excluding the CER only (so including the GDR).

Line-by-line:

17-19. This is what you found in your previous paper? Sounds like original research when in present tense. Should be "In a previous study, we found" if so. Also, sounds like you mean that current NDCs align with least stringent effort sharing method, and add up to 2.3 degC, but this isn't what ref [4] found (more like 3-4 degC). Needs to be clearer that this is a novel finding when you "prolongate" bottom-up approaches. Also, don't use the word "prolongate," try something simpler with equivalent meaning like "extending" or "extrapolating." Or maybe you meant "elongate?" Since prolongate is a more arcane way of saying "prolong" and refers to either extending the duration of time or distance in space. Do you mean : "When extending/extrapolating the current level of ambition for each country, we derive a "bottom-up" estimate of global mitigation resulting in 2.3degC of warming by 2100 (50% chance)." I also wouldn't use the word "equity" since you are referring to countries picking the least stringent effort sharing method which for developed countries is always grandfathering (CER).

Thanks for these comments and suggestions. Line 17 to 19 refer to the finding of this paper (hence the present tense). Reference [4] identifies the 2100 warming under an extension of the current and pledged policies (and finds a warming of 2.6°C to 3.1°C). In this manuscript, we model a situation where each country adopts the least stringent of five effort-sharing approaches. We find that this hypothetical pathway, which results in a 2.3°C warming, aligns with the aggregation of current NDCs. The aggregate NDCs align with a world where countries follow the 2°C goal, each using the least stringent of these five effort-sharing approaches. We have rephrased this sentence to (note that the absence of references in this introductory paragraph is a requirement of Nature Communication):

"We find that the global 2030-emissions of current NDCs match the sum of each country adopting the least stringent of five effort-sharing allocations of greenhouse-gas (GHG) emissions scenarios to achieve the Paris Agreement. We estimate that

extending such a self-interested ‘bottom-up’ aggregation of equity might lead to a median (>50% likelihood) warming of 2.3°C in 2100.” Lines 13-17

23. So this hybrid approach is from ref[1]? Or based on theory proposed by Mace? Haven’t read this article yet. If it’s your method, make sure to note that it’s novel.

Thanks for noticing. This reference was incorrect and is now removed.

24. To make this clearer: “By creating this range of top-down emissions allocations, we are able to infer which global temperature change matches which NDC. This is effect is equivalent to assessing what temperature change would be brought on if all countries acted with analogous ambition to a specified country’s level of ambition (as defined by the NDC). For example, we find that if the world acted in alignment with the NDCs of India, the EU..., warming would result of 2.6degC,..., respectively.” If I am interpreting this finding correctly, this is a clearer way of communicating the result (which is in my option the most interesting/profound insight form this study, since it informs the public and policy makers what the level of ambition an NDC amounts to in temperature change).

Thanks for your suggestion. We have adapted it to limit the amount of words for the abstract.

25. Curious as to the methodological approach of getting from Climate action Tracker’s work to this finding. Will see if you explain in methods.

We simply took the emissions estimate for the US in 2030 under a withdrawal of the Clean Power Plan as stated above.

41. Cite something on the upcoming Facilitative Dialogue (UN document)

Thanks for the suggestion. We cite the Paris Agreement that initiated the stocktake.

44. Insert “have” between “experts” and “suggested”

Thanks for the suggestion. Done.

47. Change “justify their negotiating positions” to something a bit more clearly about self-interest. Their negotiating positions are symptomatic of their self-interest. I know you say this later in the text so not absolutely necessary to change if you prefer this framing, but I just wanted to point out that it’s helpful to be explicit about the politics. Could say “...justice that best serve their self-interest and likewise justify their...”

Thanks for the suggestion. Done.

49. Remove “indeed” and replace with: “One way/approach to reconcile this ambiguity is to combine multiple concepts of equity into a single metric using weighting factors / by taking a weighted average of each approach / using a weighted average. Weighting allows for parameterization that can be adjusted to reflect the level of importance placed on any single approach.”

Thanks for the suggestion. We have updated this sentence to:

“The UNFCCC does not specify whether the equity and the CBDR-RC principles refer to distinct principles or to a single operationalization of equity(Winkler & Rajamani, 2014a). One way to reconcile this ambiguity is to combine multiple dimensions of equity using weighting factors(Baer et al., 2008; BASIC experts, 2011; den Elzen, Höhne, & Moltmann, 2008; Holz, Kartha, & Athanasiou, 2017; Peters et al., 2015; Raupach et al., 2014) and per-capita income thresholds(Baer et al., 2008; Holz et al., 2017) in a single effort-sharing approach applicable to all countries.”
Lines 49-52

51. Ref [19] sounds like similar research, if yours is an extension of their method, connect/relate the two by adding something like: we extend this methodology by including the full suite of IPCC effort-sharing approaches and [whatever else you feel you do above and beyond their work].”

Thanks for raising this. Our method is very different, in that the global ambition level the same for all countries, instead of being set by a given country (the ‘diversity leader’) as in the reference. We clarified the phrasing as follows:

“A recent study allocated emissions to each country using the least stringent of two equity allocations(Malte Meinshausen et al., 2015). The global level of ambition of each equity allocation is then set by a ‘diversity-aware’ leader so that the sum of all countries’ allocations matches 2°C-consistent levels(Malte Meinshausen et al., 2015). Under that approach, equity approaches are applied to trajectories leading to warmings lower than 2°C. Consequently, countries follow different equity approaches that are applied under different temperature thresholds, which may be considered unfair by Parties.

In the present study, we use a unique temperature threshold that is lowered consistently across all equity approaches until the ‘bottom-up’ allocation, where the least-stringent equity approach is attributed to each country individually(Malte Meinshausen et al., 2015), aligns with 2°C-consistent levels (Figure 1a). Hypothetically, countries could then follow different equity approaches applied to a common global aspirational temperature goal. Ultimately, the ‘hybrid’ approach follows a ‘bottom-up’ equity allocation consistent with a common ‘top-down’ warming threshold, which arguably reflects the hybrid nature of the Paris Agreement(Bodansky, 2016; Winkler et al., 2017). The key characteristic of this hybrid approach is then that a hypothetical warming threshold is adjusted downwards until the common warming target, either 2°C or 1.5°C, is achieved when the hypothetical warming threshold is pursued by each country under its most favourable equity allocation.” Lines 54-69

52. Why is ref [19] after leader and not end of sentence, necessary to cite again, seem to be talking about the same paper as last lines, not that crucial but maybe better way to cite? Something for the editor perhaps.

Thanks for the suggestion. Done.

51-56. OK I'm having a bit of trouble seeing the difference between their work and yours. So you're saying that they set temperature targets for each country differently so that they do their fair share based on each of two chosen effort sharing metrics or the weaker of the two /. Better one for them? And you also let countries pick the least stringent effort sharing metric but lower the temperature threshold for all countries until they meet 2 (or 1.5) degrees?

Exactly. The previous study applies different global goals for each of the used approaches, and therefore different ambition levels to different countries. This correspond to different temperature aspirational targets, even though these temperatures are not calculated in the previous study. We have clarified this section as mentioned above.

59. So did they do this in ref[19]? And you now did it for your set so the only difference is the number of choices they had? Maybe you just need to make the description of ref[19]'s study clearer. Work on that last paragraph. I'd be happy to take another look at that and anything else I've said could be clears after you edit again.

57-62. Again just be clear where ref[19]'s research ends and yours begins. After reading this over once and reading it again, I know, but on a first read it may be confusing to some. Maybe have a paragraph laying out the hybrid approach from "A to Z" rather than defining it implicitly in line

61. Don't know if this is nitpick and that maybe the style here is supposed to be more journalistic, but a suggestion nonetheless. If others have found this reads well and they see the definition then feel free to ignore.

Thanks for raising this lack of clarity in the three previous comments. No, the number of approaches is not the only difference. We do not regulate the global goal level through the ambition of a 'leader' country. In our study, a unique temperature goal is applied to all approaches, and thus to all countries. This section is now rephrased as mentioned above.

65. So are you getting into Pareto optimality here? Not sure but maybe this is worth elaborating here or maybe those references are enough and this isn't the place to go further into this. Maybe there's a less technical / more intuitive way of describing this optimality. Essentially, all countries agree that they want to limit warming to 1.5 to 2 degrees, (the absolute gain being saving civilization from possible destabilization and collapse), then they also want to get the most out of any arrangement supporting this for themselves. These two are generally in direct conflict and lead to a tragedy of the commons. This kind of research can be seen as a way of regulating the commons when parties agree that they want to protect the commons and will hold each other accountable for overuse of the atmosphere while being allowed to vie for additional benefits to themselves. Maybe add a line making this more concrete, about how this looks like in effect. Not sure if it's appropriate in a paper like this but I might add something like: **We start with the assumption that all countries support the 'absolute gain' which is the safeguarding of human civilization, since any individual economy's welfare is contingent/predicated upon the geopolitical stability of the global community, then are allowed to vie for whatever additional benefits they can garner/secure for themselves, the 'realtime gain,' without undermining the absolute gain.**

The distribution of a limited amount of emissions implies that higher allocations for some countries implies lower allocation for others. This is indeed the tragedy of the common. Using the suggestion, we have amended the manuscript to:

“The ‘hybrid’ approach does not constitute an ‘equitable’ operationalization of the CBDR-RC principle where all countries seek to maximize ‘absolute gain(Keohane, 1984)’ by agreeing on a common approach of equity. Rather, it reflects national preferences for ‘relative gain(Krasner, 1991)’, i.e. a country’s inclination to measure the fairness of its contribution to the global mitigation effort by looking at other countries’ efforts, rather than to domestic indicators alone. Despite claims that discussions of justice are “irrelevant or dangerous in a post-Paris world” (Robert Keohane quoted in ref.(Klinsky et al., 2017)), equity is fundamental climate policy research(Dooley et al., 2018; Klinsky et al., 2017) and scientific analyses on equitable burden-sharing is influential on the UNFCCC processes(Mace, 2016).” Lines 70-77

68. I like the “agree to disagree” framing

Thanks, it’s simplification of the current situation, even though many countries are pushing for more ambition.

73. Mention how your choice of scenarios is commensurate r made to represent/ quantify the Paris Agreement’s targets. Like “we pick two temperature scenarios that we feel best quantify/capture the meaning of the PA’s ‘well below 2C’ and ‘pursue efforts to limit warming to 1.5.’”

Thanks for the suggestion. We amended the manuscript to:

“The first step of this study is to model a self-interested approach of the Paris Agreement goals. We derive the ‘bottom-up’ allocation of two emissions pathways reflective of the Paris Agreement, excluding emissions from land-use and international shipping and aviation (Methods), the ‘2°C-scenario’ – with a likely (>66%) chance to stay below 2°C until 2100 (RCP2.6, ref.(Rogelj, Meinshausen, & Knutti, 2012)) and the ‘1.5°C-scenario’ – with a median 2100 warming below 1.5°C (Methods) – using five equity emissions allocations representative of the five effort-sharing categories quantified in the latest IPCC report(Clarke et al., 2014; Robiou du Pont et al., 2017, 2016)” Lines 100-107

78-82. So here I’m going to chime in about “equity.” I don’t feel that EPC is equitable since it doesn’t include historical consideration. I also don’t think CER should be included at all or maybe have another average without it. Sure it represents a choice, but is it a choice we want to offer nations? Sure you may argue, they’re picking it de facto right now anyway, but they’re not, it just happens that the poetically-derived targets countries are picking from themselves coincidentally fall in line with this approach. For all intents and purposes they could pick any metric out of the sky, e.g. we won’t be able to decarbonize and actually need a bigger share than our current ratio since we need more revenue from carbon intensive industries since some other country outdid us on some low carbon sector and now that’s failed.” I mean you can make up anything to justify/argue you need more emissions. I think it’s impossible to not be normative or a bit prescriptive here. I see the appeal of proposing a “non-normative” framework here, and I think CER is the furthest out you’d even consider countries being able to justify taking for themselves, so in this sense it’s accurate as a

depiction of reality, but it's tricky. If you are going this route, it's imperative that you remind the reader, this isn't a policy portal, we're not legitimating the continued colonialism of poor nations by wealthy ones and the perpetuation of this kind of international oppression, we're simply asking the question: "what would you have to tell countries to shoot for if they picked their current NDC-level of ambition." So in this sense it's a bit of trickery. You're saying sure pick what you like, then aim for this temp instead of 1.5 or 2.

Thanks for raising this. As we answered earlier in the review, the goal of Figure 2 is to proceed to a technical comparison between an average and the hybrid approach, not to recommend the inclusion of the CER, which is be removed from the hybrid before the derivation of the equity-based metric of figure 3. So first we study and compare the hybrid approach to the commonly used average, before tailoring it based on equity considerations. Please find a comparison of the hybrid and the average without the CER and GDR in Supplementary Figure 6b.

The EPC approach is included in the hybrid approach used in figure 3. We agree with the limitations of the EPC approach mentioned in the comment. An equality of resources does not imply an equality of outcome. However, the discussion on the fairness of the equity concepts is outside the scope of this study. We include the EPC approach as it is often discussed in the literature, and put forward by many countries in their NDCs (Robiou du Pont, 2017; Winkler et al., 2017).

We include the suggestion to state that this manuscript does not legitimate the perpetuation of the current inequities. As a first step, the manuscript models the outcome of the extension in time of a self-interested approach of effort-sharing (including the unfair CER approach), which may serve as a warning. As a second step, it models a self-interested approach of commonly accepted concepts of equity, under the Paris Agreement warming thresholds, consistently with the limitations of current agreements on equity. Progress on a common understanding of equity would certainly help to achieve the current goals, but it appears dangerous to wait for a potentially utopian outcome. We amended the text to clarify this point:

"Despite claims that discussions of justice are "irrelevant or dangerous in a post-Paris world" (Robert Keohane quoted in ref.(Klinsky et al., 2017)), equity is fundamental climate policy research(Dooley et al., 2018; Klinsky et al., 2017) and scientific analyses on equitable burden-sharing is influential on the UNFCCC processes(Mace, 2016). However, the absence of agreement on an unanimous operationalization of the CBDR-RC should not be used as an excuse for inaction(Winkler & Rajamani, 2014b) and should not leave the international community without a metric reflective of current agreements to assess the ratcheting-up process." Lines 75-80

82. Remove "in the literature" since it is plainly unfair, then cite the literature that supports that (those should be fine)

Thanks for the suggestion. Done.

83. Change "is reflective of" to "matches"

Thanks for the suggestion. Done.

84. Do they really declare their contributions as “fair?” Haven’t read this work of yours, but surely this definition of fairness is contingent on how other countries are conducting themselves. Maybe add a line about that. “These countries claim that their efforts are fair, but only in the context of the rest of the global commutates efforts. This definition of fairness is dynamic and bound to become more stringent as countries progressively ratchet up their own ambitions. Fairness is always seen asa. Relative measure, in a ‘I only want to do as much as my neighbour /the next country does’ kind of way.” (Make less colloquial, of course)

Many countries, but not all declare their contributions as fair (Robiou du Pont, 2017; Winkler et al., 2017). Countries are asked to explain how their contribution is fair and ambitious, as part of their NDC. We agree that their justification is in the context of global effort (or lack of effort).

We added:

“The CER, an approach considered unfair(Caney, 2009; Peters et al., 2015) that allocates equal emissions mitigation rates to all countries, is not openly supported by any country but implicitly matches many developed countries’ targets(Robiou du Pont et al., 2017) , which they often declare as fair(Robiou du Pont, 2017).” Lines 111-113

85-87. So here is the rest of your explanation of your hybrid method, I see it’s a bit spread out, if you understand what I’m trying to get at. Consolidating this earlier in the text with the definition of your method may be helpful to the reader. Again, if others feel it makes sense in this order, feel free to disregard.

Thanks for the suggestion, we have amended the text as suggested in pervious paragraphs.

91. Again, don’t like the word “prolongating.” Unless this is a technical word that’s absolutely necessary and says something more accurate than e.g. extrapolating, I’d change it.

Thanks for the suggestion. Changed to “extending”.

92. So this implies that at the moment, may countries are proposing NDCs that are even laxer than the CER allocation? (Citing “Paris Agreement’s need a boost...” paper) So this could be a good thing to point out explicitly. Maybe add a brief sentence to state explicitly.

Correct. This finding was the object of a previous publication, using the same model (Robiou du Pont et al., 2017). This is also visible in Figure 2. However, we do not see how this corresponds to Line 92.

102. Again just a note to provide an average with CER excluded for this too. It’s in there right?

Thanks for the suggestions. The CER is included in order to compare an average with the hybrid approach. Graphs of the average excluding the CER, or with a break-down for each approach are available in the Supplementary Information of (Robiou du Pont et al., 2017).

105. I think you note somewhere else to remind the reader that Brazil’s obligation would look different (have to be more ambitious) if you included LULUCF.

The obligations of many countries would likely be affected if LULUCF emissions were accounted, especially those which important positive or negative land-use emissions. We cannot conclude how the allocation of Brazil would be affected. However, we mentioned the great progress of Brazil in reducing deforestation, even though our metric cannot capture it.

116-117. change “excluding the redundant...” to “excluding the inequitable CER and redundant GDR approaches.” Also I strongly prefer GDR to CAP since it uses a much better way of calculating CAP than CAP does itself. I would opt to dispose of CAP in favour of using GDR with $R=0$ and $C=1$. (Will make another note in Methods)

Thanks for the suggestion. The GDR approach has an interesting way of integrating the capability component, mixed with historical responsibility and sub-national wealth distribution (paradoxically leading to greater allocation to unfair countries). However, its reliance on business as usual emissions and Gini projections makes its operationalisation highly contentious. Furthermore, the reason for removing it from the manuscript is that we seek to preserve the three basic dimensions of equity, reflected in the CBDR-RC principle of the UNFCCC (equality, capability and responsibility).

We have considerably amended the text to:

“The second step of this study derives a metric to assess the ambition of current NDCs and their future ratcheting-up, under the Paris Agreement mitigation goals and the CBDR-RC principle(UNFCCC, 1992, 2015; Winkler & Rajamani, 2014b). We quantify the ‘hybrid’ approach to reflect countries’ preferences, in 2030, for equity allocations based on dimensions of equality, responsibility and capability(Winkler & Rajamani, 2014b). Each country is attributed the equity approach with the least stringent 2030-emissions, excluding the CER and GDR approaches (Methods). Current national emissions ratios, at the root of the CER approach(Caney, 2009; Peters et al., 2015; Robiou du Pont et al., 2017), also influence near-term allocations of any continuous allocations, reflecting some of the national circumstances mentioned in the Paris Agreement(UNFCCC, 2015). The GDR approach is based on historical emissions and GDP per capita that are covered in the CPC and CAP approaches, respectively. The GDR approach relies on hypothetical projections of Gini indices and emissions (here downscaled from RCP8.5, see methods) that can lead to large variations in emission allowances. These variations can be more determined by counterfactual input assumptions than by the effort-sharing principles themselves (<http://paris-equity-check.org/>). We therefore present a synthesis of the EPC, CPC and CAP approaches in a ‘CBDR-RC hybrid’ setup that enables the derivation of an NDC-warming assessment tool ‘applicable to all’. We also provide in the Supplementary Information results under a ‘hybrid’ setup that includes the GDR, which represents a ‘right to development’, and that uses the five effort-sharing approaches.” Lines 154-171

121-22. Why “orthogonal,” do you mean that these dimensions are independent of each other? If so, I would say they’re not, they’re very much a combination and reflection of each other/ two-sides of the coin. Capacity was gained through the accumulation of wealth derived from the bringing of fossil fuels, which is historical fault (or what his refereed to responsibility). I would use another word or don’t qualify it at all. Seems like you’re trying to make a math analogy here, like you’re looking for the basis of a set of linear equations. Don’t know if this works well for equity-based policy. Way more messy than linear algebra!

Thanks for pointing this out. We decided to remove this imprecise term and change the text as stated above.

122-123. What's this tool, the Paris Equity Check website? Or this methodology you use to see what warming would be like if all countries acted with similar ambition as one chosen country's NDC pledge? Don't really get this sentence.

Thanks for raising the lack of clarity. The tool mentioned here the temperature assessment presented in Figure 3. We updated the text as stated above.

132-134. So are you suggesting that Brazil would get an even bigger share if deforestation were included (because the rate of deforestation has slowed? Please clarify. I would not say that since that's like rewarding them for being less bad, when rewards should only be additive, not less subtractive. Like in Canada, the new Conservative leader has said things like since we have so much forest on our territory we should be entitled to emit as much as we like since our forests offset it. He forgets that forests don't belong to anyone really and your territory is coincidental. In my opinion, countries shouldn't be allowed to claim decreases in emissions from not destroying their forests, but in keeping with the philosophy of good stewardship, they should be penalized by being allocated increases in emissions when they deforest their land. Brazil deforesting less than a previous should not grant them extra emissions allowances. But if we were to include LULUCF for everyone, and Brazil was reducing deforestation, it would make room for more emissions. Also maybe this should be at line 105?

As mentioned earlier, the allocations of many countries would likely be affected if LULUCF emissions were accounted, especially those which important positive or negative land-use emissions. We cannot conclude how the allocation of Brazil would be affected. However, we mentioned the great progress of Brazil in reducing deforestation, even though our metric cannot capture it. Given the absence of common accounting framework for LULUCF emissions, we exclude these emissions and do not discuss further equity for this sector. Other works deal with this issue (Dooley & Gupta, 2017) (<https://link.springer.com/article/10.1007/s10784-016-9331-z>).

141-142. So you're saying that the order of NDC-determined T changes doesn't change when accounting for uncertainty in how you quantify NDC pledges (uncertainty due to ambiguity in how pledges are worded/described)?

Correct. The selection of global scenarios affects all countries in the same manner (increasing or decreasing allocations of all countries). Change in countries' ranking can happen when changing to significantly different scenarios as the least stringent equity approach of a country changes. Using a linear fit instead of a quadratic one, would not change the ranking of countries significantly.

149-151. Don't quite get this. Can you elaborate on "contentious future reference emissions"? Does this mean that there's a lot of uncertainty regarding what emissions will look like for these countries? How does Australia, a developed nation with high capacity, benefit in a similar manner to eastern European nations and South Africa? There all quite different.

The reference emissions, or business as usual, are indeed hypothetical trajectories that depend on contentious assumptions by each country regarding what would have happened in the absence of effort or climate measures. It seems impossible for countries to agree on such trajectories². Here we used a downscaling of RCP8.5 to the national level. This downscaling results in high values for economies in transition. The calculation of the GDR approach was detailed in (Robiou du Pont et al., 2017) and results are visible on www.paris-equity-check.org

This has its own limitations which influence on the hybrid allocation if the GDR is include as we specified in the methods:

“The modelling of the GDR approach relies on business-as-usual (BaU) emissions that countries do not mutually recognize. The BaU emissions used here are downscaled from RCP8.5 resulting in allocations substantially higher than other allocations for Eastern-European countries and Australia. Compared to the ‘hybrid’ setup presented in Figure 3, the inclusion of the GDR approach in the ‘hybrid’ setup results in a lower temperature assessment in favour of the NDCs of Eastern-European countries, Australia, and South Africa, and higher temperature assessment disfavouring India, Brazil, Mexico and Indonesia (Supplementary Figure 8).” Lines 390-396

159-161. Needs work. You trying to say they’re especially inadequate since these are the same countries who championed deep and rapid decarbonization and climate finance to support mitigation in developing countries, and their slacking is therefore especially egregious/hypocritical? Like: not only are developed nations not pledging NDCs in line with the PA but they also have rhetoric that says they would be the leaders on rapid mitigation. (and as a footnote, also are not living up to their rhetoric on climate finance contributions). In other words: “The disconnect between lofty/ambitious rhetoric and lack of action makes the latter even more reprehensible.”

Thanks for the suggestion. Yes, that is the message. We updated the text to in two instances:

“Furthermore, the commitments of many major emitters, including developed countries who committed to take the lead in reducing emissions and mobilizing finance to support mitigation in developing countries(UNFCCC, 2015), do not match concepts of equity that they publicly supported(Robiou du Pont et al., 2016).” Lines 46-48

And reshaped the conclusion:

“We find that most of the least-developed countries have NDCs consistent with the Paris Agreement goals. However, the NDCs of most developing countries appear insufficient, as those of developed countries who agreed to take the lead in reducing emissions and mobilizing finance to support mitigation in developing countries(UNFCCC, 2015).” Lines 212-215

² see for example page 115 of <http://www.climatechangeauthority.gov.au/files/files/Target-Progress-Review/Targets%20and%20Progress%20Review%20Final%20Report.pdf>

Minor point on style: you put quotations inside punctuation. This deliberate? Typically one puts the quote outside but I think this is becoming a style preference. I sometimes think it's nicer and more representative of the meaning of a sentence. Does nature have a style guide/preference?

Thanks for raising this. The quotations seem to be inside punctuation in Nature articles. We will leave it that way.

201. typo = “substracted” change to subtracted

Thanks for noticing this obsolete spelling. We updated it to “subtracted” in two instances.

218. When you say “iterative” do you mean you repeat the calculation for every year? If so, I wouldn't use the term iterative since it refers to (in my mind, at least) numerical solving. This was confusing.

No, the iteration is over the ratchetting up mechanism described by the formula below. We amended the text to:

“The modelling of the hybrid allocation consists in iterative steps to derive a global ‘aspirational pathway’ whose ‘bottom-up’ allocation matches any chosen emissions scenarios from IAM.” Lines 274-276

277. Indeed, a 3 to 4 degree world would have highly inequitable distribution of impacts. Maybe cite Althor et al., 2016 Scientific Reports. “Global mismatch between greenhouse gas emissions and the burden of climate change”

Thank you for the very interesting reference, we have added the suggestion.

282. But you said that approaches are picked for individual countries?

Thanks for pointing out this imprecise phrasing. The sentence was corrected to:

“Removing the least-stringent approach of a country group would penalize these countries that would have to follow a more stringent approach, and would consequently favour all other countries.” Lines 372-375

283. Including ones in the country group?

Possibly since the attribution of allocation approaches was made at the national level.

292. “same rates” as what they were doing already, not the same as in they all mitigate at a given rate. Maybe reword to clarify.

Thanks for raising this. All countries mitigate at a unique rate, which is thus also the global mitigation rate. We reworded to:

“The CER approach represents a status-quo in terms of equity where all countries conserve their share of global emissions and mitigate at the unique rate, that of the global scenario.” Lines 382-384

337-338. Citation to substantiate “mitigated relatively easily compared to other GHG.” Is it that simple? Some study must elaborate on this? I’m personally curious.

Thanks for pointing this. We have increased the clarity of the phrasing as follows:

“Indeed, fossil CO₂ emissions can be mitigated more profoundly than other GHG (e.g. methane from agriculture). Carbon dioxide is the only gas where prototypes for large scale negative emissions technologies exist (even though cost and acceptability of large scale projects is uncertain).” Lines 430-433

Negative emissions can be achieved through carbon capture and storage, so reducing CO₂.

True, and there are many uncertainties regarding the feasibility, price and acceptability of the required measures (Dooley & Gupta, 2017).

Figures

Like the figures, no major criticisms or recommendations. Maybe add a point on how the diagonal in figure 2 represents matching NDC and equitable ambition.

Thanks for the suggestion. We have included it in Figure 2.

Methods

To summarize my thoughts as I wrote comments line by line:

- 1) I think it would be good to dispose of CER, or if being used to inform reader that it happens that current ambition often matches a CER share, then great, but maybe include an average with it excluded.
- 2) I strongly prefer GDR to CAP since it uses a much better way of calculating CAP than CAP does itself. I would opt to dispose of CAP in favour of using GDR with $R=0$ and $C=1$. CAP doesn’t capture actual capacity (as discussed by Kemp-Benedict in his justification for his Gini parametrized wealth distribution with development threshold) it’s subject to being misrepresented, since GDP/capita hides/asks the excessive wealth of the super rich and the aggregate wealth of lots of people living in abject poverty. I’m reminded of the old adage that illustrates how averages can be very misleading — the average person has one testicle.
- 3) I know you use “cost-optimal” pathways, but I’m of the camp that cost-optimal pathways are highly contentious as they don’t really account for future damages, and of course heavily privilege the wealthy and alive over the poor and the yet-to-born. I’m also wary of the magnitude of negative emissions in these pathways and would like to see what mitigation would be required to avoid negative emissions a) entirely and b) with some more technically-supported limit. I know there is a literature review on the technical limits and feasibility of large scale deployment (see Smith et al. 2016 NCC. “Biophysical and economic limits to negative CO₂ emissions”). Maybe there would be some clear numbers there to use. I am currently looking into this for my own work on making decarbonization pathways fro Canada that respect these constraints.

4) Convergence dates: I see you use a 30 year convergence date (as being politically viable) so in the non-notative spirit of this paper, maybe best to stick to that. Is this supported by other research? Could you include a scenario or a range provided by picking a faster convergence?
5) Cumulative emission period: Can you do the same thing for 4) but for the cumulative emissions period? Maybe try one at 1960 and even 1850 but that may be excessive, and unnecessary since the bulk of emissions is from mid 20th century onwards. See Matthews, 2016. Nature Climate Change. “Quantifying historical carbon and climate debts among nations” for rational. Just the argument of “we knew beyond a reasonable doubt” (1990) and “intent matters but it’s not everything” (1960).

Thanks for this summary. We have answered to this comment above, but here is a summary of the answers.

1) The CER is only included in the average for technical comparison but is then removed when deriving the temperature assessment of countries’ NDCs. We have clarified this point in the manuscript, thanks to the review’s suggestions.

2) We choose to use the framework published in (Robiou du Pont et al., 2017), which uses the modelling of (Jacoby, Babiker, Paltsev, & Reilly, 2008) for the capability approach. The goal of this study is not to revolve around a variation of the GDR. The GDR can be set to represent capability only, or even responsibility only. However, the limitations of the GDR (reliance on business as usual and Gini projections that are not commonly agreed) would impact the results too strongly. Furthermore, while the wealth threshold of the GDR is a great idea in theory at the per capita level, it results in greater emissions allocations to unfair countries (low Gini index) compared to a fair one (high Gini index). Since this study focusses on national emissions allocation, we find this characteristic inconsistent with its fairness goal at the international level.

The hybrid approach is actually an alternative to using averages (as in (Robiou du Pont et al., 2017)) and provides a rationale for the combination of effort-sharing approaches.

3) That is a very good point. The IAM scenarios represent a commonly accepted framework that provides useful global information. We agree that they also feature many shortcomings and do not feature equity considerations in their technological or economic assumptions. Thanks to this suggestion, we added a mention of this shortcoming in the manuscript:

“These scenarios from Integrated Assessment Models (IAM) represent a commonly used framework to discuss global mitigation under various Shared Socioeconomic Pathways (SSP) that represent possible futures with different equity settings (O’Neill et al., 2013). However, many technologic assumptions used in these scenarios can adversely impact vulnerable populations, depending on their implementation (for example land-based mitigation to achieve negative emissions (Dooley & Gupta, 2017).” Lines 226-231

4)&5) The convergence period is indeed 30 years, and the starting date for accounting historical emissions is 1990, the date of the first IPCC report and second World Climate Conference. As the reviewer notes, this study builds on the framework published in (Robiou du Pont et al., 2017). The purpose of this current manuscript is to present a novel way to combine views of equity (the hybrid approach), not to discuss the influence of the parameterisation of equity. The influence of the parametrisation is discussed in the supplementary information of (Robiou du Pont et al., 2016) and in the Poster (DOI:

10.13140/RG.2.2.22758.63049,

https://www.researchgate.net/publication/311130635_Quantifying_equitable_allocations_of_a_2C_consistent_emissions_pathway).

Studying the influence of parametrisation in depth, under a given global target would certainly be an interesting and useful exercise that would represent a study on its own.

Congratulations on the submission and good luck with the revisions!

Thanks a lot for the support, the feedback of the reviewers contributes to making this work clear and relevant to a broad audience. We hope that this manuscript now addresses the concerns raised in the reviews.

- Daniel

Reviewer #3 (Remarks to the Author):

The manuscript “Temperature assessment of the bottom-up Paris emissions pledges” introduce a new method to evaluate the individually NDCs and can inform the ratchetting-up process of the Paris Agreement.

Interestingly, the paper follows the architecture of the Paris Agreement, with a global mitigation goal and bottup-up pledges. The paper presents an operationalization of an ‘agreement to disagree’ on equity concepts to achieve a common temperature goal. Each country can choose an equity approach to determine its effort. The least stringent equity approach are chosen for each country when allocating global scenario emissions to each country. This acknowledges the countries’ self-interested positions. The global scenarios, used to allocate global emissions to countries, are adjusted such as the temperature goal is reached (1.5/2.0 degrees or to follow IAM scenarios).

The topic of the paper is NDCs and global temperature targets. A method to assess the ambition of the current NDCs is presented in the paper, and would be of interests in a wide community.

I feel that the paper is not very well presented and some clarifications are needed before possible publications. My main comments are related to Figure 3. See below.

Thanks for your feedback and interest. We have improved the clarity of the manuscript thanks to these suggestions and those of the other reviewers. The introductory paragraph, the rationale and the conclusion have been reshaped.

Here are my specific comments:

Line 14-16: “Current NDCs align,...” consider rephrasing.

Line 17: Which emission scenarios are you referring to?

Thanks for these two points. We corrected the sentences and reshaped the introductory paragraph to:

“Under the bottom-up architecture of the Paris Agreement, countries pledge Nationally Determined Contributions (NDCs). Current NDCs individually align, at best, with divergent concepts of equity and are in aggregation inconsistent with emissions scenarios to achieve the warming thresholds of the Paris Agreement. We find that the global 2030-emissions of current NDCs match the sum of each country adopting the least stringent of five effort-sharing allocations of greenhouse-gas (GHG) emissions scenarios to achieve the Paris Agreement. We estimate that extending such a self-interested ‘bottom-up’ aggregation of equity might lead to a median (>50% likelihood) warming of 2.3°C in 2100. We find that ratcheting-up the warming goal of all the individual ‘bottom-up’ effort-sharing allocations to hypothetical levels of 1.1°C and 1.3°C could achieve the Paris Agreement’s warming thresholds of 1.5°C and well below 2°C, respectively. This new ‘hybrid’ allocation that reconciles the ‘bottom-up’ pledging nature of the Paris Agreement with its ‘top-down’ warming threshold, provides a temperature metric to assess NDCs. When taken as benchmark by other countries, the NDCs of India, the EU, the USA and China lead to warmings of 2.6°C, 3.2°C, 4°C and over 5.1°C respectively, under the current bottom-up regime.” Lines 10-23

Line 73: “First,..” What is the “second/then”?

Thanks for pointing this out. We reshaped the text to have two clear steps. The second step appears as the start of the paragraph starting line 154.

Line 96: Do you mean that the global temperature goal should be lower than 1.5? Consider rephrasing.

Thanks for pointing this out. This study suggests aspirational temperature targets, lower than the current Paris Agreement targets. These aspirational targets would be missed and overshoot under a self-interested approach of effort sharing. The result is that the overshoot of the aspirational targets would correspond to the Paris Agreement targets (1.5°C or 2°C).

We corrected this part to:

“A ‘bottom-up’ allocation consistent with more stringent ‘aspirational’ global scenarios that limit warming to 1.1°C and 1.3°C results in global emissions consistent with 1.5°C and 2°C respectively (Figure 1, Methods). Under such a self-interested approach of effort-sharing, the international community needs to aim at an aspirational target of 1.3°C to effectively stay well below 2°C, and 1.1°C to effectively return to 1.5°C. The resulting temperature gaps between the aspirational target and effective warming reflect the necessary strengthening of global temperature aspirations to compensate for the disagreement on effort-sharing (Figure 1).” Lines 124-131

Line 133: “Our assessment finds lower warming assessments for Ethiopia and Philippines” Lower than what? The previous assessment in ref 5? These two countries could then be included in the list at Line 132.

Thanks for raising this. We reformulated the comparison with the results of an existing online assessment tool (now ref. 46) to:

“An assessment for 32 countries(Climate Action Tracker, 2017) finds similar results for the NDCs of USA, Russia, and Indonesia, higher warming assessments for Ethiopia and the Philippines, and lower warming assessments for Australia, Brazil, Canada, China, Japan, India and the EU. However, the progresses of Brazil on deforestation are not accounted in this study as land-use emissions are excluded.” Lines 178-182

Line 213-217: Here all Figures 1-3 are related to “while preserving a likely change to limit global warming to 2 degrees”. The figures are related to both 1.5 degree, 2 degrees and the different temperatures in Fig. 3. This need to be rephrased. To be clearer, maybe include a reference to Fig 3 after these words: “The modelling of the hybrid allocation”. It would be useful if you are clearer in the description of the method if you are talking about the 1.5/2 degree scenario and the IAMs results used in Fig. 3. That would help the reader a lot.

Thanks a lot for noticing this imprecise and confusing language. Indeed, Figure 3 does not refer to a likely chance of staying below 2°C. We have corrected this sentence to:

“Under the ‘hybrid’ approach, every country picks the least stringent approach, in terms of cumulative emissions over 2010-2100 (Figure 1 and 2) or 2030 emissions levels (Figure 3), while staying below a warming threshold.” Lines 272-274
We also update the caption of figure 3, as the other comments below suggest.

Line 258-260: I do not understand this sentence or how Fig. 3 show this. Please explain.
Thanks a lot for pointing this out. This sentence is mistakenly present in this version of the manuscript and is now deleted.

Line 261-263: Consider rephrasing.

Thanks for the suggestion. We have rephrased this section to:

“For 36 countries, relationships between 2030 emissions and 2100 warming non-strictly monotonous and oscillate. In each case, the inflection points are at 4.8°C or more, higher than their NDC assessments of the corresponding countries. The national emissions allocations at high temperature are only indicative for these countries. In the absence of interpolation, inflection points are found for 14 countries, and are at lower temperature than the NDCs of: Iraq, Trinidad and Tobago, Jordan, Brunei Darussalam and the Maldives.” Lines 338-343

Line 313-314: Is aerosol forcing also one of the uncertain model parameters? What is the range of the aerosol forcing?

The aerosol forcing is one of the models of the MAGICC model, we have not explored the influence of that parameter as the paper focuses on the methodological contribution of combining multiple equity approaches. The parametrisation for aerosol forcing is that of the underlying study (M. Meinshausen, Raper, & Wigley, 2011; M. Meinshausen, Wigley, & Raper, 2011).

Other comments:

In the figures, warming of X degrees. Relative to what? What is the reference period? 1750 or 1850 or another period? Please specify.

Thanks for raising this. Warming refers to pre-industrial levels as the Paris Agreement. We amended the caption of figure 3, and the first paragraph of the methods.

Related to the scenarios, what is the start year of the mitigation? I presume it is before 2017. Since we are at the end of 2017, how will it affect the results?

The starting year of mitigation (or other trajectories, depending on the scenario) is 2010. The main reason is that this manuscript uses scenarios from (Robiou du Pont et al., 2017) with this starting date. Note that the UNFCCC inventories currently only stretch until 2015 for Annex I countries (http://di.unfccc.int/time_series). The data for non-Annex I countries stops at even older dates as these do not have the same reporting requirements. Emissions data for non-Annex I countries that is not self-reported, is available until 2014 (ref.(Gütschow et al., 2016)). The GDP scenarios and development narratives, as well as the business as usual

emissions projections (and other parameters) have been harmonised accordingly and we do not know of more up to date data.

It is likely that the mitigation requirements presented here would be slightly greater, and that as a result, the temperature assessment of current NDCs would be higher. Nonetheless, we believe that adding a few years of data would not strongly affect the results of this study regarding the ranking of countries' ambition.

Line 195: The aggregation of Kyoto-GHG emissions follows the 'SAR GWP-100'. Why do you still use SAR GWP and not AR5 GWPs?

LULUCF CO₂ emissions are excluded from the analysis, but the other Kyoto GHGs are included? What about aerosols and other short lived components?

We are using the GWP SAR to match the reporting requirement of the UNFCCC (http://unfccc.int/ghg_data/items/3825.php). The mitigation percentages calculated in this study could be applied to emissions accounted with other GWP without introducing significant bias. In this manuscript GHG emissions refer to the Kyoto GHG. All Kyoto GHG are therefore accounted, but not other gases. Emissions from the LULUCF sector are accounted in the global scenarios, when deriving global warming responses, but are excluded from the allocation to countries.

We clarified this point in the methods as:

“The aggregation of Kyoto-GHG emissions follows the ‘SAR GWP-100’ (Global Warming Potential for a 100-year time horizon), consistently with the reporting under UNFCCC (http://unfccc.int/ghg_data/items/3825.php).” Lines 253-255

Comments to the Figure:

Fig 1a: Hybrid allocation, Equity approach 2: Impossible to read “Aspirational 2°C scenario”.
Figure caption line 468: “Small island developing states by their maritime zones.” Does this sentence belong to Fig. 1?

Thanks for pointing this out. We have made the label more readable and corrected the caption.

Fig 2: Suggest to add “(rest of the world)” below “Other Economies” as (G8+China) is added below “Major economies”. Rest of the world are used in the figure legend and in the main text.

Thanks for this suggestion. We have changed the figure accordingly.

Fig 3: I am struggling a bit with this figure and the method used for making it. Below is my comments.

Why do you include the map when the name of the countries are written? Do you need the map? Is it only for visualization? E.g. Mexico and Brazil have the same color, but in the method used to derive the global temperature response they are treated as separate countries, right?

That is correct, the map offers a visualisation of the warming assessment of the data for the countries presented in Figure 3a. The map also shows the warming assessment of the NDCs of all the other countries part of this study.

The colour of Mexico is more orange, and that of Brazil redder, consistently with Figure 3a. We are aware that a colour scale has limitations, we provide the numeric values for all countries in a separate file as supplementary material.

You say in the text that you use 9 scenarios with 2100 warming of 1.2°C to 5.1°C. In the figure caption you write: “Coloured dots represent allocations under each of the 121 global scenarios”. And further in the caption on line 484 you write: “coloured lines and patches range over the interpolations of degree one, two and three (Methods)” I am struggling a bit here. Are both the lines and the patches determined by the range over the interpolations of degree one, two and three? The patches are not determined by the individual scenarios (121 global scenarios) consistent with the range of the NDC? I am struggling a bit on the method here, and it would be useful with more clarification.

Thanks for pointing this out. We have rectified the caption to:

“For the top-fifteen emitters, 2030-emissions as a function of 2100 median warming above pre-industrial levels under the hybrid approach (coloured lines). Disks indicate the ‘average’ NDC assessment and their sizes are proportional to 2010 emissions. Vertical ranges (coloured rectangles) indicate NDC assessment ranges (Malte Meinshausen & Alexander, 2015). The horizontal uncertainty ranges (coloured rectangles) over the warming of the IAM scenarios (coloured dots) that lie within the vertical NDC assessment range. Colours indicate countries’ world regions (see map inset).” Lines 609-614

The relationship between countries’ emissions and global temperature outcome is modelled by a second degree fit of a sub-selection of nine scenarios selected to be representative of the 121 IAM scenarios. The methodology is detailed as follows in the methods section. We updated the methods section to:

“The relationship between 2030 national emissions levels and the 2100 temperature responses presented in Figure 3 is derived from a representative sub-selection of global emissions scenarios. We standardize the data across both dimensions (2030 emissions, excluding LULUCF and bunkers, and 2100 warming) and derive the third-degree polynomial fit (Supplementary Figure 1). Using a second-degree polynomial fit would result in a plateau where high global warming hardly depends on 2030 emissions levels. We then select a subset of scenarios with the least standardized distance to the fit, starting at the lowest 2100 temperature and every 0.5°C (nine scenarios, Supplementary Table 1, Supplementary Figure 2).” Lines 237-244

and

“The bijectivity between NDCs ambition and their temperature assessment relies on the strict monotony of the relationship between global scenario’s 2030 emissions and the 2100 warming. We selected nine global scenarios every 0.5°C to achieve such strict monotony at the global level. The 2030 emissions levels dependency to the NDC temperature assessment of Figure 3 is a second-degree polynomial fit based the allocations derived from the selected nine global scenarios. A second-degree polynomial approach smoothens the variability while preserving the greater sensitivity of national 2030 emissions allocations at lower 2100 warmings.” Lines 331-337

We have also clarified the legend of figure 3a.

Figure 3a is messy around the disks of China, Russia, Canada etc. As I understand, these countries NDCs does not correspond to any of the IAM scenarios. Maybe extend the x-axis with a > 5.1 to make this clearer.

Thanks for this suggestion. We have modified the figure accordingly.

And I do not understand the dotted lines from these disks towards the main part of the figure. It would be very useful if the figure caption of Fig. 3b could be extended. So far only: "Global warming responses (median assessment) following NDC ambitions." And in the figure "Warming in degrees" One or two sentences what have actually been done (maybe related to Line 216-217).

Thanks for this suggestion. The dotted line schematically connected the countries disk, which are outside the horizontal scale as noticed, with the countries curve that stops at 5.1°C of warming. We have modified the figure to address the issue, countries whose NDCs leads to a warming over 5.1°C have the symbol "*". The caption for Figure 3b now reads:

"Global warming assessment (50% likelihood, compared to pre-industrial levels) of 'average' NDC ambitions for 169 countries as calculated in panel a. (maps with 'high' and 'low' NDC quantifications in Supplementary Figures 9 and 10). The assessment ranges from 1.2°C to 5.1°C , NDCs outside this range are not differentiated. Small island developing states by their maritime zones." Lines 614-618

"Other countries": Use the same definition as in the rest of the figures.

Thanks for this pointing this out. We have modified the figure accordingly.

Maybe specify in the figure caption that the darkest green color and the darkest red color are outside countries with NDCs that correspond to global mean temperature change below or above the scenario range. Or use an array in the color bar. Now the color bar is rounded at the ends covering several temperature levels.

Thanks for these suggestions, we have modified the figure and caption accordingly.

I like that the figure show the large spread in median warming of the scenario, and as mentioned in the text, the emission change over the period 2010 to 2030 can lead to large spread in the 2100 warming. Maybe you can include two map figures in the supplement with the max and min scenario warming.

Thank you. Conveying the uncertainty specific to this methodology is important. Thanks as well for the suggestion. We have added two maps reflecting the temperature assessment under the 'high' and 'low' NDC quantifications ('low' and 'high' end of the vertical ranges of Figure 3a, respectively), instead of the 'average' assessment used in Figure 3 (see supplementary figure 9 and 10). Note that the same method as for Figure 3b is used. The NDC warming assessments are taken from the fitting-curve when its value is equal to the NDC emissions quantification ('high' or 'low').

Comments to the supplementary:

The title of the supplementary text is not equal to the title of the main article.

Thanks a lot for pointing this out. This was corrected.

Line 34 and 40: grew -> gray

Thanks for seeing this. Now corrected.

FigS5: How does this map plot show uncertainty?

Thanks for noticing this mistake in the caption. The maps do not show an uncertainty, only an alternative paramterisation (using different effort-sharing approaches). The caption is corrected.

Supplementary table: Columns D to OH contain the national 2030 emissions levels associated with the global 2100-warming temperature of Row 1 using the 'Hybrid approach'[in GgCO₂eq]. Is the unit correct? It should not be in %?

Again, thanks a lot for noticing this. Now corrected.

Thanks a lot for all the very constructive comments that greatly improve the quality of this manuscript. We hope that could address the issues these comment in a satisfying manner.

- Arneson, R. (2013). Egalitarianism. Retrieved January 1, 2017, from <https://plato.stanford.edu/entries/egalitarianism/>
- Averchenkova, A., Stern, N., & Zenghelis, D. (2014). *Taming the beasts of “burden-sharing”: an analysis of equitable mitigation actions and approaches to 2030 mitigation pledges*. Centre for Climate Change Economics and Policy - Grantham Research Institute on Climate Change and the Environment. London. Retrieved from <http://www.lse.ac.uk/GranthamInstitute/publication/taming-the-beasts-of-burden-sharing-an-analysis-of-equitable-mitigation-actions-and-approaches-to-2030-mitigation-pledges/>
- Baer, P., Fieldman, G., Athanasiou, T., & Kartha, S. (2008). Greenhouse Development Rights: towards an equitable framework for global climate policy. *Cambridge Review of International Affairs*, 21(August 2014), 649–669. <https://doi.org/10.1080/09557570802453050>
- BASIC experts. (2011). *Equitable access to sustainable development: Contribution to the body of scientific knowledge*. (H. Winkler, T. Jayaraman, J. Pan, Y. Santhiago de Oliveira, Adriano Zhang, G. Sant, J. D. Gonzalez Miguez, ... S. Raubenheimer, Eds.), BASIC expert group: Beijing, Brasilia, Cape Town and Mumbai. Beijing, Brasilia, Cape Town and Mumbai. Retrieved from <http://gdrights.org/wp-content/uploads/2011/12/EASD-final.pdf>
- Bodansky, D. (2016). The Paris climate change agreement: a new hope? *The American Journal of International Law*, 110(2), 288–319. Retrieved from <https://doi.org/10.5305/amerjintelaw.110.2.0288>
- Caney, S. (2009). Justice and the distribution of greenhouse gas emissions. *Journal of Global Ethics*, 5(2), 125–146. <https://doi.org/10.1080/17449620903110300>
- Clarke, L., Jiang, K., Akimoto, K., Babiker, M., Blanford, G., Fisher-Vanden, K., ... van Vuuren, D. P. (2014). *Chapter 6 Assessing Transformation Pathways*. In: *Climate Change 2014: Mitigation of Climate Change*. Cambridge, United Kingdom and New York, NY, USA. Retrieved from http://www.ipcc.ch/pdf/assessment-report/ar5/wg3/ipcc_wg3_ar5_chapter6.pdf
- Climate Action Tracker. (2017). Climate Action Tracker. Retrieved January 1, 2017, from <http://www.climateactiontracker.org/>
- den Elzen, M., Höhne, N., & Moltmann, S. (2008). The Triptych approach revisited: A staged sectoral approach for climate mitigation. *Energy Policy*, 36(3), 1107–1124. <https://doi.org/10.1016/j.enpol.2007.11.026>
- Dooley, K., & Gupta, A. (2017). Governing by expertise: the contested politics of (accounting for) land-based mitigation in a new climate agreement. *International Environmental Agreements: Politics, Law and Economics*, 17(4), 483–500. <https://doi.org/10.1007/s10784-016-9331-z>
- Dooley, K., Gupta, J., & Patwardhan, A. (2018). INEA editorial: Achieving 1.5 °C and climate justice. *International Environmental Agreements: Politics, Law and Economics*, 18(1), 1–9. <https://doi.org/10.1007/s10784-018-9389-x>
- Fleurbay, M., Kartha, S., Bolwig, S., Chee, Y. L., Chen, Y., E., C., ... Sagar, A. (2014). *Chapter 4. Sustainable Development and Equity*. In: *Climate Change 2014: Mitigation of Climate Change. Contribution of Working Group III to the Fifth Assessment Report of the Intergovernmental Panel on Climate Change*. Cambridge, United Kingdom and New York, NY, USA. Retrieved from http://www.ipcc.ch/pdf/assessment-report/ar5/wg3/ipcc_wg3_ar5_chapter4.pdf
- Gupta, S., Tirpak, D. A., Burger, N., Gupta, J., Höhne, N., Boncheva, A. I., ... Sari, A. (2007). Working Group III to the IPCC AR4 - Chapter 13: Policies, Instruments and Co-operative Arrangements, 746–807. <https://doi.org/10.1088/0264-9381/22/1/L01>

- Gütschow, J., Jeffery, M. L., Gieseke, R., Gebel, R., Stevens, D., Krapp, M., & Rocha, M. (2016). The PRIMAP-hist national historical emissions time series. *Earth System Science Data*, 8(2), 571–603. <https://doi.org/10.5194/essd-8-571-2016>
- Höhne, N., den Elzen, M., & Escalante, D. (2013). Regional GHG reduction targets based on effort sharing: a comparison of studies. *Climate Policy*, 14(1), 122–147. <https://doi.org/10.1080/14693062.2014.849452>
- Holz, C., Kartha, S., & Athanasiou, T. (2017). Fairly sharing 1.5: national fair shares of a 1.5 °C-compliant global mitigation effort. *International Environmental Agreements: Politics, Law and Economics*, 1–18. <https://doi.org/10.1007/s10784-017-9371-z>
- Jacoby, H. D., Babiker, M. H., Paltsev, S., & Reilly, J. M. (2008). *Sharing the Burden of GHG Reductions. MIT Joint Program on the Science and Policy of Global Change*. Retrieved from <https://globalchange.mit.edu/publication/14428>
- Kemp-Benedict, E. (2010). *Calculations for the Greenhouse Development Rights Calculator*. Somerville, MA, USA. Retrieved from <https://www.sei-international.org/mediamanager/documents/Publications/Climate/calculations-gdr-calculator.pdf>
- Keohane, R. O. (1984). *After Hegemony: Cooperation and Discord in the World Political Economy*. Princeton: Princeton University Press. Retrieved from <http://press.princeton.edu/titles/1322.html>
- Klinsky, S., Roberts, T., Huq, S., Okereke, C., Newell, P., Dauvergne, P., ... Bauer, S. (2017). Why equity is fundamental in climate change policy research. *Global Environmental Change*, 44, 170–173. <https://doi.org/10.1016/j.gloenvcha.2016.08.002>
- Krasner, S. D. (1991). Global Communications and National Power: Life on the Pareto Frontier. *World Politics*, 43(3), 336–366. <https://doi.org/10.2307/2010398>
- Lange, A., Löschel, A., Vogt, C., & Ziegler, A. (2010). On the self-interested use of equity in international climate negotiations. *European Economic Review*, 54(3), 359–375. <https://doi.org/10.1016/j.eurocorev.2009.08.006>
- Mace, M. J. (2016). Mitigation Commitments Under the Paris Agreement and the Way Forward. *Climate Law*, 6(1–2), 21–39. <https://doi.org/10.1163/18786561-00601002>
- Meinshausen, M., & Alexander, R. (2015). INDC Factsheets. Retrieved January 1, 2015, from <http://climatecollege.unimelb.edu.au/indc-factsheets/>
- Meinshausen, M., Jeffery, L., Guetschow, J., Robiou du Pont, Y., Rogelj, J., Schaeffer, M., ... Meinshausen, N. (2015). National post-2020 greenhouse gas targets and diversity-aware leadership. *Nature Climate Change*, 5(12), 1098–1106. <https://doi.org/10.1038/nclimate2826>
- Meinshausen, M., Jeffery, L., Guetschow, J., Robiou Du Pont, Y., Rogelj, J., Schaeffer, M., ... Meinshausen, N. (2015). National post-2020 greenhouse gas targets and diversity-aware leadership. *Nature Climate Change*, 5(12). <https://doi.org/10.1038/nclimate2826>
- Meinshausen, M., Raper, S. C. B., & Wigley, T. M. L. (2011). Emulating coupled atmosphere-ocean and carbon cycle models with a simpler model, MAGICC6 – Part 1: Model description and calibration. *Atmospheric Chemistry and Physics*, 11(4), 1417–1456. <https://doi.org/10.5194/acp-11-1417-2011>
- Meinshausen, M., Wigley, T. M. L., & Raper, S. C. B. (2011). Emulating atmosphere-ocean and carbon cycle models with a simpler model, MAGICC6 - Part 2: Applications. *Atmospheric Chemistry and Physics*, 11(4), 1457–1471. <https://doi.org/10.5194/acp-11-1457-2011>
- O'Neill, B. C., Kriegler, E., Riahi, K., Ebi, K. L., Hallegatte, S., Carter, T. R., ... van Vuuren, D. P. (2013). A new scenario framework for climate change research: the concept of shared socioeconomic pathways. *Climatic Change*, 36(4), 147–149. <https://doi.org/10.1007/s10584-013-0905-2>

- Pan, X., Elzen, M. Den, Höhne, N., Teng, F., & Wang, L. (2017). Exploring fair and ambitious mitigation contributions under the Paris. *Environmental Science and Policy*, 74(August), 49–56. <https://doi.org/10.1016/j.envsci.2017.04.020>
- Peters, G. P., Andrew, R. M., Solomon, S., & Friedlingstein, P. (2015). Measuring a fair and ambitious climate agreement using cumulative emissions. *Environmental Research Letters*, 10(10), 105004. <https://doi.org/10.1088/1748-9326/10/10/105004>
- Raupach, M. R., Davis, S. J., Peters, G. P., Andrew, R. M., Canadell, J. G., Ciais, P., ... Le Quéré, C. (2014). Sharing a quota on cumulative carbon emissions. *Nature Climate Change*, 4(10), 873–879. <https://doi.org/10.1038/nclimate2384>
- Robiou du Pont, Y. (2017). *The Paris Agreement global goals: What does a fair share for G20 countries look like?* Melbourne. <https://doi.org/10.13140/RG.2.2.11932.69762>
- Robiou du Pont, Y., Jeffery, M. L., Gütschow, J., Christoff, P., & Meinshausen, M. (2016). National contributions for decarbonizing the world economy in line with the G7 agreement. *Environmental Research Letters*, 11(5), 54005. <https://doi.org/10.1088/1748-9326/11/5/054005>
- Robiou du Pont, Y., Jeffery, M. L., Gütschow, J., Rogelj, J., Christoff, P., & Meinshausen, M. (2017). Equitable mitigation to achieve the Paris Agreement goals. *Nature Climate Change*, 7(January), 38–43. <https://doi.org/10.1038/NCLIMATE3186>
- Rogelj, J., Meinshausen, M., & Knutti, R. (2012). Global warming under old and new scenarios using IPCC climate sensitivity range estimates. *Nature Climate Change*, 2(4), 248–253. <https://doi.org/10.1038/nclimate1385>
- Sabin Center for Climate Change Law. (2018). Climate Case Chart. Retrieved from <http://climatecasechart.com/>
- Schiermeier, Q. (2015). Landmark court ruling tells Dutch government to do more on climate change. *Nature News*, (June). <https://doi.org/10.1038/nature.2015.17841>
- The Hague District Court. ECLI:NL:RBDHA:2015:7196 (2015). Retrieved from <http://deeplink.rechtspraak.nl/uitspraak?id=ECLI:NL:RBDHA:2015:7196>
- Tørstad, V., & Sælen, H. (2017). Fairness in the climate negotiations : what explains variation in parties' expressed conceptions? *Climate Policy*, 3062(August), 1–13. <https://doi.org/10.1080/14693062.2017.1341372>
- UNFCCC. (1992). *United Nations Framework Convention on Climate Change. Report No. FCCC/INFORMAL/84. Report No. FCCC/INFORMAL/84.* Rio. Retrieved from <https://unfccc.int/resource/docs/convkp/conveng.pdf>
- UNFCCC. (2012). *Report on the workshop on equitable access to sustainable development. - Report: FCCC/AWGLCA/2012/INF.3/Rev.1.* Bonn. Retrieved from <http://unfccc.int/resource/docs/2012/awglca15/eng/inf03r01.pdf>
- UNFCCC. (2015). *Adoption of the Paris Agreement. Report No. FCCC/CP/2015/L.9/Rev.1* (Vol. 21932). Paris, France. Retrieved from <http://unfccc.int/resource/docs/2015/cop21/eng/109r01.pdf>
- Urgenda. (2017). Urgenda Climate Case. Retrieved January 1, 2017, from <http://www.urgenda.nl/en/climate-case/>
- Winkler, H., Höhne, N., Cunliffe, G., & Maria, J. de V. C. (2017). Countries start to explain how their climate contributions are fair : more rigour needed. *International Environmental Agreements: Politics, Law and Economics*, 1–17. <https://doi.org/10.1007/s10784-017-9381-x>
- Winkler, H., & Rajamani, L. (2014a). CBDR&RC in a regime applicable to all. *Climate Policy*, 14(1), 102–121. <https://doi.org/10.1080/14693062.2013.791184>
- Winkler, H., & Rajamani, L. (2014b). CBDR & RC in a regime applicable to all. *Climate Policy*, 14(1), 102–121. <https://doi.org/10.1080/14693062.2013.791184>

Reviewers' comments:

Reviewer #1:

This reviewer only left comments to editor)

Reviewer #2 (Remarks to the Author):

Peer review by Daniel Horen Greenford (Round 2)

I have read all the responses to the review comments by Dr. Robiou du Pont and am pleased with his work addressing them. I recommend that the manuscript be accepted with minor revisions. See my comments, criticisms, questions, and recommendations below.

General comments:

Big fan of Figure 3, great way of visualizing data in panel a, and the choropleth in b is an excellent summary and way of communicating the ambition (and relative difference in ambitions) of countries. These findings are still my favourite part of the paper.

I am much happier with the introductory paragraph and the general messaging of the manuscript. That said, I have maintained some of my original criticisms. Also, I am still a bit confused as to how CER is included (in which parts of the analysis). Is it still part of the "average" 1.5 and 2 degree scenarios, and/or bottom-up and hybrid approaches?

I would really try to use "grandfathering" instead of "Constant Emissions Ratio" (CER), since I feel that latter is too much of a euphemism which sounds technical and scientific, which in turn endows it with authority. It just is too loaded and evasive and will mislead lay readers (and as this is a general audience publication, as part of Nature Communications, I would try to keep the language as accessible as possible). I know you use the CER convention in all your previous work, but you could link it to that and just say "in previous work we use CER but the community uses "grandfathering" to describe the perpetuation of the status quo, where countries continue to use a share of remaining emissions proportional to their historic usage." Even "grandfathering" sounds a bit neutered but at least it's common parlance. IMO I would prefer to use terms like "status quo".

As for the equity definitions and the effort sharing methods. One thing worries me greatly about how you set up Equal Per Capita (EPC). Correct me if I'm wrong, but when I looked at the ERL paper SI, it seems that countries per capita emissions converge to the same amount at a fixed year in the future. Is this not just Contraction and Convergence (C&C)? If so, why do you call it EPC, and not just C&C? If this is the case, I find it misleading to name it EPC when to me, EPC means equal per capita emissions started immediately. This works with "resource sharing" (i.e. sharing what's left of cumulative emissions), but understandably, instantaneous change is impossible with effort-sharing and so there must be a convergence date, but then it's not really EPC...

I think it would be more appropriate to swap the analysis in the SI that excludes CER from the suite of possible equity choices with the main body analysis that includes it. You could keep the w/CER analysis to show what happens when countries default to gathering when it best suits them, but I maintain that CER/grandfathering cannot ever be considered even de facto "equitable" and is never argued as such (with a normative basis, even though countries try to rationalize how it's doing their "fair share" post hoc).

OK just to debate myself here, and anticipate the author's response to this, I do agree that there is validity to providing analysis of a hypothetical situation that is deemed to be a realistic representation of current international policy environment. In this case, countries *are* choosing to use de facto grandfathering allocations, and hence CER represents the lower end of the equity

spectrum, and therefore should be included in the main analysis. The fact that courts are employing grandfathering as this lower end is also reason for it to be included. However if this is the case, then this study represents a snapshot of the current state of policy rather than a possible or desirable policy avenue for the future. Here I find the response to be equivocating between “yes, we recognize that” and still has language suggesting that this approach can be a policy solution. Not sure how to best rectify this. Maybe we are being overly harsh but it is only because we are worried about the message here. We are aware that the authors do not advocate moral subjectivism or “post-ethics”/neorealist policy, however the concern that work will be used out of context in this highly politicized realm of climate justice-oriented policy is very real.

So are you proposing that the IPCC recommend lower temperature targets in order to achieve the 1.5 to well-below 2 degree target?

One thing that I must point out, is the reality that countries do not seem to concoct policies with much consideration of effort-sharing literature, or even any consideration of top-down shares of the remaining global carbon budget. So when we posit here that increasing global ambition via an aspirational temperature target will ensure that countries follow suit and ultimately, even if acting in a self-interested manner (within the constraints set by a lower limit of grandfathering, barring even more egregious behaviour/overuse), we can still succeed in achieving the Paris goals... This seems dubious. Is there not a risk that countries will just continue deriving policies politically? Why would this entice them to only now start considering the implications of top-down scientific insights? As someone engaged in this field of work myself, I understand that the merit of our work is not necessarily in its political viability, but in its illustrative power, and how then this can be transformed into political currency, and so in an indirect way, we can hope to affect policy, but the route is long and convoluted. I still cannot resist playing the devil’s advocate here so please forgive me!

Would be interesting to see a little bit of sensitivity analysis to test what happens when the bounds of the effort sharing approaches are changed. Like using a less stringent than grandfathering approach (i.e. greater than historical proportion) and a more stringent high-equity approach (e.g. high progressivity setting Climate Equity Reference Project (CERP) approach).

Reflections on reviewer comments:

Just a few notes here after reading the responses the authors made to reviewer comments.

In the response to Reviewer 1, the authors assert that the “hybrid approach” is not put forward as a replacement for a common normative definition of equity or equitable approach to effort—sharing, however it can still be interpreted as this by the reader. After reading the amendments and restructuring of the crucial components of the manuscript, I am still somewhat dissatisfied with the clarity of this message. Perhaps adding a line in the conclusion would help to remedy this. Something to the effect of: “While our approach serves as an alternative perspective to reconcile disagreements on a common definition of equity in effort-sharing, this should not be interpreted as an endorsement of moral subjectivism. In fact, the hybrid approach serves as model the extreme self-interested approach that countries must strive to escape if global mitigation efforts are to become sufficient to successfully limit warming to 1.5 to well below 2 degrees. Rather than settling on an “agreement to disagree”, nations must continue their debate as to what “fairness” or “justice” encompasses in their national contribution to global mitigation efforts.”

Questions:

Q1: So when you let countries pick the least stringent effort sharing choice, then lower the aspirational temperature target until the actual temperature target is achieved, does that mean that the balance of equity is preserved? or does that change the ranking of relative shares? In other words, if countries then examined the equity outcome, their relative share, after the

aspirational temperature was defined, would they still find that their share was the least stringent allowed?

Q2: Is the iterative process you use a common one, or derived for this specific use? I'm not too familiar with the mechanics of bootstrapped iterative/numerical solutions, so just inquiring.

Q3: So you use all five ES approaches for figure 1, but only the 3 independent/base ones for figure 3?

Lines:

17-19: Phrasing better but would change "could achieve" to "would achieve" in line 18. I think this is how it should be put: it is a hypothetical outcome of a bottom-up scenario, not a top-down policy outcome. I don't think the idea should be that we're going to dictate a more stringent temperature target in order to achieve the 1.5 to well-below 2 degrees target. It still feels like some sort of weird trickery. Really want to see this framed as "what would it take for us to make Paris work if all countries acted in a fully self-interested manner" (with a lower limit of grandfathering their emissions, since they could act even worse; perhaps this should also be acknowledged and noted somewhere in the text). I see it as an interesting thought experiment that can help people understand the magnitude of the challenge and the implications of self-interested policymaking, not as a viable international governance framework.

42-43: a bit unclear, or could be reworded. Are you trying to say that: "Some countries don't pick the effort sharing metric that benefits them most, or even speak explicitly about equity implications of their NDCs, most countries have attempted too justify their NDC choices with some arguments of how they are equitable (as required by the criteria of the INDC submission process under the Paris Agreement) [Winkler et al., 2018], however this doesn't imply that NDCs are equitable or much less that countries will convene upon a common definition of equity and a common effort-sharing method."? SO some will go above and beyond the call of duty, but these nations are the minority and often are at a significant political/economic disadvantage (like LDCs). Some developed countries who espouse "climate leadership" are doing better than grandfathering or not? Think it's the latter, right? So could note that here too, that others have found this (e.g. CAT) and that you found this in PEC.

47-48: Are you saying adaptation finance pledges are inadequate, not up to equity standards? Please clarify further. Just mentioning as characteristic of climate leadership, and how these hough pledgers don't exhibit similar ambition when it comes to domestic abatement policy. Would be good to mention that contributions to GCF are also insufficient and that don't represent fair share. Not sure what to cite here but maybe something? If you find a paper, please let me know. I'm working on similar analysis now but too early to cite.

55-57: Still not sure I like this, really don't see this as a viable policy outcome. It still sounds like deception to say to countries: pick the effort sharing method that most benefits you, including grandfathering (but like I said before, why stop there? countries in theory could occupy even more than their historical emissions space!). Surely, we have to cut the equity limit off somewhere, and for me that would be equal per capita starting immediately.

81-82: This is why CER must be excluded from the suite of choices. Including it in the main analysis serves to endorse it as a legitimate option. The worry here is that by including it in the main analysis, it will be taken out of context, read off the main figure, and cited as a legitimate choice, just like during these court proceedings.

89: Is this the IPCC convention? i.e. does "unlikely" here mean <33% chance? Worth noting, if so (for non-expert audience).

110-113: Again, I am advocating for the unfair metric to be removed from primary analysis, since it is no way tenable/defensible as a definition of equity or an equitable outcome.

129-131: Again, are you suggesting that this is a viable option (re: increasing aspirational temperature target to compensate for a country's tendency to overuse their fair share of emissions). At points it seems you do not want to comment on whether this is feasible (technically, politically, etc.) however it still seems like this may be implied, even though the reader should be aware that this is a modelled outcome of a hypothetical world, resembling certain simplified aspects of the real world (as with all modelling). I would suggest adding a caveat to the end, perhaps in the conclusion or later in the discussion, to note that "...although this is suggested as a feasible way of avoiding agreement on effort sharing approaches, we do not explore the specifics that are required to make this approach possible. Assessment of the technological, economic, and political barriers to increasing ambition via an 'aspirational' target are beyond the scope of this study, and may be so grave to render this method inviable. Regardless of this exercise's real world /real politic applicability, the analysis presented still provides an illustration of the ambition gap, quantified in temperature terms, when all countries follow self-interested domestic mitigation policies."

132-136: Can the language here be simpler? I had to read through the whole paper and come back to this. Wasn't clear if e.g. China or Russia was using more or less of their least stringent fair share. Are you saying higher ambition or higher emissions in absolute terms (lower ambition)?

140: I would hesitate to say that how countries NDCs line up with effort-sharing schemes is an indication of or that it "reflects individual countries' views on equity". Countries do not consult effort-sharing literature before crafting policies, hence not having an a priori knowledge of equity, they at best try to justify their decisions / root them in equity arguments post hoc. I would be careful with language here. Saying they coincidentally match / align with effort-sharing approach x, y, or z is fine but not that they are representative of a country's views on equity.

162-163: Not really, as reviewer 1 had pointed out, CERP (f.k.a. GDR) contains the dimensions of wealth and population, but does not use per capita values and hence is not a linear combination of per capita emissions and GDP. In this sense, excluding GDR/CERP from the set of base indicators is not legitimate. However, from the perspective of keeping the base set to just including dimensions of population and GDP, it is fine to exclude GDR. I understand the appeal here, the additional simplicity is very attractive.

163-166: Is the sensitivity analysis for this somewhere in the documentation of older papers? I'm interested in what the spread is for the median CERP under different BAU projections vs the spread of the outcomes under the range of effort sharing principles (outlined here for example).

174-176: That's astonishing. Can this be reiterated in simpler terms as a take home message like: "In other words, the NDCs of Canada, Russia and China are much less than what would be required of them if they had the least stringent / most beneficial share of any global emissions scenario we analyzed." Also what was that exactly? The 5.1degC trajectory? Or RCP 8.5? Or the least stringent trajectory to achieve the Paris Agreement (e.g. 66% chance of 2degC)?

Just to confirm again (and I won't share these findings yet): So does this mean that if all countries pledged analogous NDCs to Canada, China or Russia, warming would be equal to or in excess of 5.1 degrees? This is a pretty powerful message. Once again, I really like the tidiness and how informative this single number metric is for describing a country's ambition. Figure 3 is excellent, a lot of information in the panel a, but worth including, the choropleth in panel b is a great figure for picking up the essentials at a glance.

181-182: Thanks for including this note. Might have mentioned this before. Important to highlight

how the picture here would change if including LULUCF.

199-201: Although I'm not a member of the CERP research group, I will speak on their behalf a bit more: it has been explained to me that the way PEC uses the GDR metric fundamentally changes the outcome, due to the choice of NAU projections used mostly I think. Not sure about all the intricacies here behind the scenes, but I do know that the outcome of CERP for these countries (developed European and Canada etc.) are NOT favoured from their work but somehow that switches under the PEC methodology. In this sense, calling your it 'GDR' here may be misnomer? You seem to take the overarching manipulation but change the fundamental inputs and achieve entirely different results (hence the sensitivity to inputs you mention above and indeed that outcome can be greater than changing the effort sharing principle itself). Perhaps to reconcile this you could add a line in the methods if not done already saying you use the method from GDR but achieve a fundamentally different result and call it something else like the 'Right to Development (RTD)' principle?" This way we could distinguish between the differing methodologies between the PEC and CERP projects. Now there is a risk that readers will conflate the two. I know this has happened already with the PEC website.

204: Once again, I don't think that countries really do 'pledge based on their own understanding of fairness' — this seems like wishful thinking. Could this be rephrased to say that countries usually pledge in a self-interested fashion, where they try to get away with doing as little as possible without looking worse than their peers, then justify their pledge with some contorted equity argument post hoc? Of course, change the language to not be so polemical. This is my take on international negotiations, I think it's better than speculation. Not sure if those references also support this take. Though I can't see how it happens the other way around. By citing work, you're saying this isn't your speculative analysis but even though I am not familiar with the literature to support my argument, I haven't read anything to support the argument as it stands presently. Maybe I am being overly pedantic about the language used, but it means something entirely different to me. Will look more into this later on my own time.

240-241: Is there a physical reasons for using this or just common practice?

300-301: not sure what you mean by this? The CPC was giving the wrong sign before? What did you do exactly? I thought is just shares the annual emissions trajectory, starting at e.g. 1990, equally per capita between countries.

349-350: because of impact at 2+ degrees?

354: typo: change 'ta' to 'the'

368: 'sufficiencatrian': never seen this word before, coined in this paper? Could you explain it or use another word?

381-382: once again not to nitpick, but I would refrain from saying "supported implicitly" even, since you either do support equity or you don't... It is clear that countries who have NDCs that fall in the CER or worse range and hence want to perpetuate the status quo or even a graver injustice of using more than their already unfair share, patently do not support equitable outcomes. Maybe this is a matter of option, but using the term 'equity' implies for me, some modicum of equity. While in the sense the word is being used here, it is totally ambiguous, like how the word 'quality' is satirized in the comic strip 'Dilbert' — If you use it in isolation, like saying we produce 'quality goods', you aren't really saying anything. They could be of terrible quality, but it still is an effective marketing play.

410: so does this mean that the warming actually would be higher or have you composted already for LULUCF?

434-435: So LULUCF CO2 is included? But not non-CO2 gases?

Closing thoughts:

I would advocate for Figure 1 being redone with either CAP, CPC, and EPC, only, or the three base ones plus GDR, i.e. excluding CER.

I think a more plain-language summary of what you do in the second part of your analysis (depicted results in figure 3) would be nice for the lay reader. Something with the words, "warming resulting if all countries acted in an analogous manner", could go in the abstract.

I think your approach, rather than be used as a hypothetical 'agree to disagree' scenario, could be instead or also be framed as a predictive tool / thought experiment for what would happen if all countries 1. Pick the least stringent equity setting allowed, and 2. Act all like a given country. This for me sums up the paper entirely. The results of this are a novel and significant contribution to the climate equity literature. I look forward to seeing this in print soon.

Reviewers' comments:

Reviewer #1:

This reviewer only left comments to editor)

Reviewer #2 (Remarks to the Author):

Peer review by Daniel Horen Greenford (Round 2)

I have read all the responses to the review comments by Dr. Robiou du Pont and am pleased with his work addressing them. I recommend that the manuscript be accepted with minor revisions. See my comments, criticisms, questions, and recommendations below.

General comments:

Big fan of Figure 3, great way of visualizing data in panel a, and the choropleth in b is an excellent summary and way of communicating the ambition (and relative difference in ambitions) of countries. These findings are still my favourite part of the paper.

I am much happier with the introductory paragraph and the general messaging of the manuscript. That said, I have maintained some of my original criticisms. Also, I am still a bit confused as to how CER is included (in which parts of the analysis). Is it still part of the “average” 1.5 and 2 degree scenarios, and/or bottom-up and hybrid approaches?

Thanks a lot for this review and for letting us know that it is not only us who think of Figure 3 as an intuitive and new illustration of currently pledged national ambitions.

We understand the concern of the author that the CER approach that is not based on fairness is included as recommendation or used in a metric.

Following the reviewer's comments, we now implemented three further changes:

- 1) Figure 2 does not include any more the CER approach. The so-called CBDR-RD Bottom up approach is different from the previous “Bottom-up” approach by excluding the CER.
- 2) To clarify in which parts of the study the CER (Constant Emission Ratio) approach is used, we now provide an overview Table in the SI.
- 3) Also, we now include in the figure captions an explicit explanation of which underlying approaches are used for the Hybrid and Bottom-up approaches.

The CER approach is used in Figure 1 (and does not present national allocations), which offers a modelling of a self-interested situation and highlights the difference between the continuation of this situation and the global climate objectives. The presence of the CER and the results displayed in this figure reflects the importance it had in shaping the first round of NDCs and warns against a continuation of its utilisation. Current aggregated NDCs align with such a self-interested situation when status quo is considered as an option for countries. Prolongating such a situation would lead to warming inconsistent with the Paris Agreement. This is an important finding. Thus, we hope that the reviewer agrees with us that including CER in this status quo analysis adds to the usefulness of the study. We also take into account the reviewer's comment below which nicely starts with “OK just to debate myself here...” Following this finding, the article suggests a metric to assess countries' ambition. This metric is based on fairness context, is called ‘CBDR-RC Hybrid’ and exclude the CER approach.

For a better comparison with the bottom-up approach, i.e. when the global temperature goal is kept at 2C or 1.5C, the ‘CBDR-RC bottom-up’ approach is consistently limited to just reflect approaches EPC, CAP. And CPC, but not CER. Hence, the CER is completely absent from Figure 3.

Following the reviewer’s suggestion, we have removed the CER approach from the analysis in Figure 2.

I would really try to use “grandfathering” instead of “Constant Emissions Ratio” (CER), since I feel that latter is too much of a euphemism which sounds technical and scientific, which in turn endows it with authority. It just is too loaded and evasive and will mislead lay readers (and as this is a general audience publication, as part of Nature Communications, I would try to keep the language as accessible as possible). I know you use the CER convention in all your previous work, but you could link it to that and just say “in previous work we use CER but the community uses “grandfathering” to describe the perpetuation of the status quo, where countries continue to use a share of remaining emissions proportional to their historic usage.” Even “grandfathering” sounds a bit neutered but at least it’s common parlance. IMO I would prefer to use terms like “status quo”.

Thanks for the suggestion. We agree that clarity is important to a wide audience. The suggestion has been implemented in the text.

As for the equity definitions and the effort sharing methods. One thing worries me greatly about how you set up Equal Per Capita (EPC). Correct me if I’m wrong, but when I looked at the ERL paper SI, it seems that countries per capita emissions converge to the same amount at a fixed year in the future. Is this not just Contraction and Convergence (C&C)? If so, why do you call it EPC, and not just C&C? If this is the case, I find it misleading to name it EPC when to me, EPC means equal per capita emissions started immediately. This works with “resource sharing” (i.e. sharing what’s left of cumulative emissions), but understandably, instantaneous change is impossible with effort-sharing and so there must be a convergence date, but then it’s not really EPC...

Thanks for the suggestions. We avoid the use of the C&C acronym, as we are not seeing the advantage of C&C versus another term, such as EPC or Per-capita convergence. We were contacted more than once that the usage of the term C&C in the literature incurs license fees (as somebody seems to have trademarked it, [http://www.gci.org.uk/trade mark.html](http://www.gci.org.uk/trade_mark.html)), which is somewhat not in line with our understanding of open science. Thus, ECP or Per-capita convergence do just as fine and might be more appropriate as they do not follow the exact same settings that the trade mark holder might have chosen.

The modelling and naming of equality-based approaches varies in the literature. For example, many equal per capita approaches are budget based and distributed over time following arbitrary (often linear) trajectories that never result in equal per capita annual emissions. We have followed the naming of the study it is based on. Other literature has referred to this approach as ‘equity’ (Peters, Andrew, Solomon, & Friedlingstein, 2015), which sounds subjective and controversial to us.

The text has been modified as follows:

“Similar modelling to the EPC is also named per-capita convergence (Meinshausen et al., 2015), equity (Peters et al., 2015) or similar.” (Line 301)

I think it would be more appropriate to swap the analysis in the SI that excludes CER from the suite of possible equity choices with the main body analysis that includes it. You could keep the w/CER analysis to show what happens when countries default to gathering when it best suits them, but I maintain that CER/grandfathering cannot ever be considered even de facto “equitable” and is never argued as such (with a normative basis, even though countries try to rationalize how it’s doing their “fair share” post hoc).

Thanks for suggestions. We have implemented the suggestion and swapped the Supplementary Figure 6 and Figure 2 (and added a panel in the SI to compare the results of these two setups). The new Figure 2 in the main text does not include the CER approach. Note that this section only serves as a technical analysis and a comparison between averaging and ‘hybridizing’ several burden-sharing approaches. In any case, the CER is never referred to as equitable and we have made sure not to refer to it as an equity approach, only as a burden sharing approach in the manuscript.

OK just to debate myself here, and anticipate the author’s response to this, I do agree that there is validity to providing analysis of a hypothetical situation that is deemed to be a realistic representation of current international policy environment. In this case, countries *are* choosing to use de facto grandfathering allocations, and hence CER represents the lower end of the equity spectrum, and therefore should be included in the main analysis. The fact that courts are employing grandfathering as this lower end is also reason of it to be included. However if this is the case, then this study represents a snapshot of the current state of policy rather than a possible or desirable policy avenue for the future. Here I find the response to be equivocating between “yes, we recognize that” and still has language suggesting that this approach can be a policy solution. Not sure how to best rectify this. Maybe we are being overly harsh but it is only because we are worried about the message here. We are aware that the authors do not advocate moral subjectivism or “post-ethics”/neorealist policy, however the concern that work will be used out of context in this highly politicized realm of climate justice-oriented policy is very real.

We basically agree with the reviewer’s reasoning. As a result, we use Figure 1 to describe the current situation implied by NDCs and project a continuation of this situation to show the inconsistency with global climate objectives. This is a crucial finding.

The metric proposed to assess global ambition is present in Figure 3 and completely excludes the CER/grandfathering approach. This metric only combines equity approaches based on ‘equality’, ‘historical responsibility’ and ‘capability’.

To avoid confusion, we have also moved the technical analysis of Figure 2, which included the CER approach in the previous version of the manuscript, to the SI. In the new version of the manuscript, Figure 2 exclude the CER approach.

Moreover, all the national allocations presented in the paper exclude the CER approach.

We hope that these changes satisfy reviewer #2, as they are extensive. Among the two authors, there is however also the view (not universally shared 😊), though, that CER could indeed be an indirect equity criteria that emphasizes national circumstances. In particular given the falling cost curves of new zero emission technology undercutting new fossil fuel investments in the electricity sector, it is less clear that fairness and development space is monotonously correlated with emissions. In fact, the least cost and lowest emission development is these days often one and the same in the majority of regions and in the

dominant electricity sector at least. Thus, while ‘grandfathering’ as a resource sharing approach has no moral standing – simplified speaking – it might start to have a moral relevance in the new world, where zero emission technology is cheaper. However, given that this is stuff for another article, we leave that debate out of the current one. We just hope that the reviewer will agree with our strong efforts to de-emphasize CER, calling it more clearly ‘grandfathering’ and swapping the main analysis with the SI as per the reviewer’s suggestion.

So are you proposing that the IPCC recommend lower temperature targets in order to achieve the 1.5 to well-below 2 degree target?

No, this study aligns with the existing objectives of the Paris Agreement. This study uses virtual/hypothetical temperature targets as a tool to determine national emissions reductions in line with the global climate objectives (1.5°C and 2°C) under differentiated approaches of fairness. These virtual/aspirational targets only reflect the “cost” of disagreeing on equity when aiming at the Paris goals.

One thing that I must point out, is the reality that countries do not seem to concoct policies with much consideration of effort-sharing literature, or even any consideration of top-down shares of the remaining global carbon budget. So when we posit here that increasing global ambition via an aspirational temperature target will ensure that countries follow suit and ultimately, even if acting in a self-interested manner (within the constraints set by a lower limit of grandfathering, barring even more egregious behaviour/overuse), we can still succeed in achieving the Paris goals... This seems dubious. Is there not a risk that countries will just continue deriving policies politically? Why would this entice them to only now start considering the implications of top-down scientific insights? As someone engaged in this field of work myself, I understand that the merit of our work is not necessarily in its political viability, but in its illustrative power, and how then this can be transformed into political currency, and so in an indirect way, we can hope to affect policy, but the route is long and convoluted. I still cannot resist playing the devil’s advocate here so please forgive me!

Having been ourselves part of the science-policy nexus in UNFCCC and IPCC for a while, we understand and basically agree with this view. We have no illusion of this becoming the new approach that all nations will abide by. Nor can it be the realistic purpose of any of these equity and burden sharing approaches to hope for their implementation. Scientific studies on the question (just as our study here) will only ever be useful to inform the decision making around politically set targets. Our study will be no different.

However, as with other studies, some countries clearly use scientific studies as a starting point and potentially strong influencing factor when deriving their politically set targets (for example the EU mitigation targets are based on the IPCC AR4 recommendation for developed countries, Chapter 13, Box 13.7). Additionally, judges may enforce IPCC recommendations (Urgenda, 2017). Finally, NGOs and civil society pressure governments inform their work by the scientific literature. The question here is not whether this study will solve the mitigation gap, but whether it can make a useful contribution to the debate of how and whom could/should ratchet-up global ambition. We are convinced that this study provides a unique and useful metric, reflective of the bottom-up architecture of the Paris

Agreement and will help the multiple actors of this process as one additional piece of information (among many).

Would be interesting to see a little bit of sensitivity analysis to test what happens when the bounds of the effort sharing approaches are changed. Like using a less stringent than grandfathering approach (i.e. greater than historical proportion) and a more stringent high-equity approach (e.g. high progressivity setting Climate Equity Reference Project (CERP) approach).

Thank you for this suggestion. We added a section to address the influence of this indirect sensitivity, related to the sensitivity of burden-sharing approaches themselves but not related to the ‘hybrid’ combination of approaches. We added in the main manuscript a description of the influence of the parameterisation of the underlying approaches on the results.

“Additionally, an indirect source of sensitivity arises from the sensitivity of the underlying effort-sharing approaches to their input data and parameters (Robiou du Pont, 2018; Robiou du Pont, Jeffery, Gütschow, Christoff, & Meinshausen, 2015). While this sensitivity is not linked to the ‘hybrid’ combination of effort-sharing approaches, the warming assessment and ranking of NDC’s ambition can be affected by political choices such as the choice of period to account for historical emissions, convergence periods and technical assumptions regarding projections of population, GDP and BaU emissions (Methods).” (Line 220)

In the Methods, we added a discussion of the influence of the parameters on the allocation results.

“The sensitivity of effort-sharing approaches to their input data and parameters indirectly affects the results of their ‘hybrid’ combination and thus of the NDC’s warming-assessments presented in Figure 3. The sensitivity of the effort-sharing framework used in this study (Robiou du Pont et al., 2017) was studied under a range of parameters consistently across the five effort-sharing approaches (Robiou du Pont, 2018; Robiou du Pont et al., 2015). The sensitivity analysis was performed by quantifying 3 to 81 combinations of parameters, depending on the approach. Because of their flexible parameterization, the most sensitive approaches are the CPC (sensitive to the period covering historical emissions) and GDR approaches (also sensitive to business-as-usual emissions projections, and internal parameters such as wealth threshold and responsibility-capability ratio (Baer, Fieldman, Athanasiou, & Kartha, 2008)). The GDR approach is included in the ‘Complete hybrid’ quantifications, but not in the NDC assessment of Figure 3. In addition to the allocation approaches’ uncertainty, various ‘exogenous’ assumptions have equity implications. For example, an earlier (later) convergence date for the EPC and CAP approaches, and earlier (later) starting date to account for historical emissions is expected to favour – result in a lower NDC warming-assessment – developing (developed) countries compared to the current NDCs.” (Line 408)

Additionally, the present manuscript contains a sensitivity analysis directly related to the process of combining multiple approaches. Figure 3 and the Methods/SI provide analyses of

respectively the dependence of the results on the considered approaches and on the underlying uncertainty of global warming under global emissions scenarios.

We did, however, not include the quantification of the sensitivity suggested by the author as it would provide incomplete information. The suggestion of picking only two approaches (the grandfathering and the GDR/CERP, which cannot be unambiguously described as the most and least stringent approaches – given that this varies from country to country (Robiou Du Pont et al., 2017)), would create a number of interpretative challenges. Instead, we believe that an independent study should explore the sensitivity of all the five burden-sharing approaches to the whole range of parameters. This work is part of an independent study that is in progress and involves an in-depth exploration of all the burden-sharing approaches' parameters. Initial results (Robiou du Pont et al., 2015) scope 129 scenarios across the five approaches (3 to 81 scenarios depending on the burden sharing approach and its parameters <http://hdl.handle.net/11343/213527> Table 5.1, page 72). A meaningful and comprehensive analysis of the sensitivity across the combination of these scenarios under the calculation-intensive Hybrid approach would require yet another study (and quite a bit more computing time).

Reflections on reviewer comments:

Just a few notes here after reading the responses the authors made to reviewer comments.

In the response to Reviewer 1, the authors assert that the “hybrid approach” is not put forward as a replacement for a common normative definition of equity or equitable approach to effort—sharing, however it can still be interpreted as this by the reader. After reading the amendments and restructuring of the crucial components of the manuscript, I am still somewhat dissatisfied with the clarity of this message. Perhaps adding a line in the conclusion would help to remedy this. Something to the effect of: “While our approach serves as an alternative perspective to reconcile disagreements on a common definition of equity in effort-sharing, this should not be interpreted as an endorsement of moral subjectivism. In fact, the hybrid approach serves as model the extreme self-interested approach that countries must strive to escape if global mitigation efforts are to become sufficient to successfully limit warming to 1.5 to well below 2 degrees. Rather than settling on an “agreement to disagree”, nations must continue their debate as to what “fairness” or “justice” encompasses in their national contribution to global mitigation efforts.”

Thanks for the suggestions. We have implemented the suggestions with some amendment as we do not feel to be in a position to directly tell countries what they ‘should’ do. We have added to the conclusion:

“The ‘hybrid’ approach should not be interpreted as an endorsement of moral subjectivism, and only a commonly accepted operationalisation of equity would result in a fair and enduring mitigation (Klinsky et al., 2017). The ‘hybrid’ approach serves as an additional benchmark, reflective of the current Paris Agreement architecture, to assess the ratcheting-up process and avoid the most unfair outcome: unmitigated climate change.” (Line 238)

Questions:

Q1: So when you let countries pick the least stringent effort sharing choice, then lower the aspirational temperature target until the actual temperature target is achieved, does that mean that the balance of equity is preserved? or does that change the ranking of relative shares? In other words, if countries then examined the equity outcome, their relative share, after the aspirational temperature was defined, would they still find that their share was the least stringent allowed?

That is an important question and we have added the answer in the Methods. Yes, countries can follow different approaches throughout the iterative process. Their least stringent approach under the initial bottom up approach may not be the same as the one under the final Hybrid approach. Supplementary Figure 5 shows that this is the case for few countries.

Q2: Is the iterative process you use a common one, or derived for this specific use? I'm not too familiar with the mechanics of bootstrapped iterative/numerical solutions, so just inquiring.

I have not used it previously, but the convergence is rather quick (10 to 15 iterations). This is fortunate since each iteration requires substantial computational time.

Q3: So you use all five ES approaches for figure 1, but only the 3 independent/base ones for figure 3?

That is correct.

And following suggestions from reviewer 2, we now also restrict figure 2 to just include 3 independent approaches.

Lines:

17-19: Phrasing better but would change “could achieve” to “would achieve” in line 18. I think this is how it should be put: it is a hypothetical outcome of a bottom-up scenario, not a top-down policy outcome. I don't think the idea should be that we're going to dictate a more stringent temperature target in order to achieve the 1.5 to well-below 2 degrees target. It still feels like some sort of weird trickery. Really want to see this framed as “what would it take for us to make Paris work if all countries acted in a fully self-interested manner” (with a lower limit of grandfathering their emissions, since they could act even worse; perhaps this should also be acknowledged and noted somewhere in the text). I see it as an interesting thought experiment that can help people understand the magnitude of the challenge and the implications of self-interested policymaking, not as a viable international governance framework.

Thanks, it has been addressed following the referee's suggestion. In accordance with the reviewer's opinion, we avoid suggesting that an objective should be imposed to any country. We checked the language throughout the text. The sentence line 18 has been changed to:

“Tightening the warming goal of each country's effort-sharing approach to aspirational levels of 1.1°C and 1.3°C could achieve the 1.5°C and well-below 2°C thresholds, respectively.” Lines 17

42-43: a bit unclear, or could be reworded. Are you trying to say that: “Some countries don’t pick the effort sharing metric that benefits them most, or even speak explicitly about equity implications of their NDCs, most countries have attempted too justify their NDC choices with some arguments of how they are equitable (as required by the criteria of the INDC submission process under the Paris Agreement) [Winkler et al., 2018], however this doesn’t imply that NDCs are equitable or much less that countries will convene upon a common definition of equity and a common effort-sharing method.”? SO some will go above and beyond the call of duty, but these nations are the minority and often are at a significant political/economic disadvantage (like LDCs). Some developed countries who espouse “climate leadership” are doing better than grandfathering or not? Think it’s the latter, right? So could note that here too, that others have found this (e.g. CAT) and that you found this in PEC.

Thanks for the suggestion. We added a sentence that highlights how most large economies have made insufficient contributions compared to multiple visions of equity and that the most ambitious countries are LDCs or developing countries.

“Developed countries who committed to take the lead in reducing emissions and mobilizing finance for developing countries (UNFCCC, 2015) often submitted NDCs that do not match the concepts of equity that they publicly supported (Robiou du Pont, Jeffery, Gütschow, Christoff, & Meinshausen, 2016) and leave the Green Climate Fund poorly funded (Althor, Watson, & Fuller, 2016). Their NDCs often imply a status-quo in terms of global emission shares (Robiou du Pont et al., 2016) while most of the very ambitious NDCs are from smaller developing countries (Robiou du Pont & Jeffery, 2017).” (Line 46)

47-48: Are you saying adaptation finance pledges are inadequate, not up to equity standards? Please clarify further. Just mentioning as characteristic of climate leadership, and how these hough pledgers don’t exhibit similar ambition when it comes to domestic abatement policy. Would be good to mention that contributions to GCF are also insufficient and that don’t represent fair share. Not sure what to cite here but maybe something? If you find a paper, please let me know. I’m working on similar analysis now but too early to cite.

Thanks for pointing this out. This paper does not tackle issues related to adaptation. A sentence has been added to refer to the GCF and compare developed and developing countries’ ambitions.

“Developed countries who committed to take the lead in reducing emissions and mobilizing finance for developing countries (UNFCCC, 2015) often submitted NDCs that do not match the concepts of equity that they publicly supported (Robiou du Pont et al., 2016) and leave the Green Climate Fund poorly funded (Althor et al., 2016). Their NDCs often imply a status-quo in terms of global emission shares (Robiou du Pont et al., 2016) while most of the very ambitious NDCs are from smaller developing countries (Robiou du Pont & Jeffery, 2017).” (Line 46)

55-57: Still not sure I like this, really don’t see this as a viable policy outcome. It still sounds like deception to say to countries: pick the effort sharing method that most benefits you,

including grandfathering (but like I said before, why stop there? countries in theory could occupy even more than their historical emissions space!). Surely, we have to cut the equity limit off somewhere, and for me that would be equal per capita starting immediately.

These lines refer to a different study (Meinshausen et al., 2015). Regarding the reviewer's comments, we note however that the selection of the different allocation regimes included in our study largely follow the ones in the political debate and reflect the effort-sharing categories presented in the IPCC AR5. And yes, if a country or group of countries would have presented another "equity" approach that would then have become part of the broader discourse, we would have included it in this study. Given that grandfathering is the implicit approach that is in line with many (large) countries' NDCs does not make that approach morally right, but makes it suitable for this study's purpose, i.e. to investigate the consequence of our continuous disagreement on equity.

We would like to stress that the manuscript does not mention grandfathering or any specific approach at this stage. This section describes the theoretical framework of how to combine different effort-sharing approaches. The question of the selection of the approaches occurs later.

81-82: This is why CER must be excluded from the suite of choices. Including it in the main analysis serves to endorse it as a legitimate option. The worry here is that by including it in the main analysis, it will be taken out of context, read off the main figure, and cited as a legitimate choice, just like during these court proceedings.

We can see the argument of the reviewer that the CER must be excluded from recommendations on fair share given its lack of equity basis. As a result, it is not used as part of the metric to assess the fairness and ambition of countries (Figure 3) and it is not included in any of the national results discussed in this paper. Following the reviewer's suggestions we have also excluded the CER from the technical analysis in Figure 2, but keep it in Figure 1 as it reflects the current status (and one end of the range) of international debate.

89: Is this the IPCC convention? i.e. does "unlikely" here mean <33% chance? Worth noting, if so (for non-expert audience).

We did not use 'unlikely' following the IPCC convention in this intense. We corrected the wording to 'insufficient'.

"While the multiplication of climate litigations cases against governments (Sabin Center for Climate Change Law, 2018) can contribute to the ratcheting-up process, systematic court decisions that governments must follow the least-ambitious end of an equity range would be insufficient to achieve the Paris Agreement." (Line 85)

110-113: Again, I am advocating for the unfair metric to be removed from primary analysis, since it is no way tenable/defensible as a definition of equity or an equitable outcome.

The CER is not part of the fairness metric (Figure 3) in this analysis and it is not used to derive any of the national results presented in the study.

129-131: Again, are you suggesting that this is a viable option (re: increasing aspirational temperature target to compensate for a country's tendency to overuse their fair share of emissions). At points it seems you do not want to comment on whether this is feasible (technically, politically, etc.) however it still seems like this may be implied, even though the reader should be aware that this is a modelled outcome of a hypothetical world, resembling certain simplified aspects of the real world (as with all modelling). I would suggest adding a caveat to the end, perhaps in the conclusion or later in the discussion, to note that "...although this is suggested as a feasible way of avoiding agreement on effort sharing approaches, we do not explore the specifics that are required to make this approach possible. Assessment of the technological, economic, and political barriers to increasing ambition via an 'aspirational' target are beyond the scope of this study, and may be so grave to render this method inviable. Regardless of this exercise's real world /real politic applicability, the analysis presented still provides an illustration of the ambition gap, quantified in temperature terms, when all countries follow self-interested domestic mitigation policies."

The technical feasibility or viability of following the hybrid approach is that of implementing the underlying IAM scenario (for example RCP2.6). The more ambitious aspirational target is only used to calculate each country's fair share, but the sum of all countries' emissions matches that of the initial IAM scenario.

The political feasibility of implementing this approach or any equity-based approach – and the implicit transfers of allocated emissions to match them with real emissions – is the challenge posed but not solved here. It will depend on the support that this approach gains. We are confident that the hybrid approach metric presented here, which reflects the bottom up nature of the Paris Agreement, will make a useful contribution to a debate on equity that is often presented as a deadlock preventing a progress of negotiations (Zenghelis, 2017). As mentioned in our reply to reviewer 2, we are however neither under the illusion or wish that this approach will be implemented. It is an analysis tool to inform the debate and allow to compare relative stringency of national pledges from a new angle.

We have modified the manuscript to clarify this point.

"The national emissions trajectories resulting from the 'hybrid' allocation add up to the targeted IAM scenario and can be met through combination of domestic mitigation, internationally traded mitigation outcomes (UNFCCC, 2015) and financial contributions. As a result, the technical feasibility of the hybrid approach follows that of the underlying IAM scenario." (Line 134)

132-136: Can the language here be simpler? I had to read through the whole paper and come back to this. Wasn't clear if e.g. China or Russia was using more or less of their least stringent fair share. Are you saying higher ambition or higher emissions in absolute terms (lower ambition)?

Thanks for pointing this out. We have clarified the sentence.

"At the national level, the current NDCs of the G8 countries (including the 28 EU countries) and China imply higher 2030-emissions than even the most favourable of the three equity approaches applied to the 2°C-scenario. In other words, their NDCs are less ambitious than the 'CBDR-RC bottom-up' allocation of the 2°C-scenario, unlike the 'other economies' as a group (Figure 2a)." (Line 159)

140: I would hesitate to say that how countries NDCs line up with effort-sharing schemes is an indication of or that it “reflects individual countries’ views on equity”. Countries do not consult effort-sharing literature before crafting policies, hence not having an a priori knowledge of equity, they at best try to justify their decisions / root them in equity arguments post hoc. I would be careful with language here. Saying they coincidentally match / align with effort-sharing approach x, y, or z is fine but not that they are representative of a country’s views on equity.

Thanks for this comment. The comment probably refers to another line, as it does not seem to apply to the content of line 140. However, we will take the opportunity to answer. We agree that the alignment of an NDC with an effort sharing approach does not necessarily indicate a preference of the country for that approach. We do not think that we have made such statements. Regarding the importance of the literature on effort-sharing, we would like to emphasise the adoption by the EU of the mitigation target recommended by the IPCC AR4 (Chapter 13, Box 13.7).

162-163: Not really, as reviewer 1 had pointed out, CERP (f.k.a. GDR) contains the dimensions of wealth and population, but does not use per capita values and hence is not a linear combination of per capita emissions and GDP. In this sense, excluding GDR/CERP from the set of base indicators is not legitimate. However, from the perspective of keeping the base set to just including dimensions of population and GDP, it is fine to exclude GDR. I understand the appeal here, the additional simplicity is very attractive.

The modelling used in this study follows that of the studies that introduce the GDR approach (Baer et al., 2008; Kemp-Benedict, 2010). The papers explicitly use per capita values for emissions and income, as well as per capita income thresholds. The GDR approach is indeed not a linear combination of per capita emissions and GDP, but neither is the CAP. Our justification for removing the GDR approach (aside from its strong dependence on hypothetical business as usual emissions) approach is that it is based on a combination of responsibility and capability, which are here covered by the CPC and CAP approaches. The GDR is explicitly a combination of the responsibility and capacity principles as stated the abstract of the study that first introduced it (Baer et al., 2008):
“In this article we present a new framework called ‘Greenhouse Development Rights’ (GDRs): a formula for the calculation of national obligations on the basis of quantified capacity (wealth) and responsibility (contribution to climate change). GDRs”

We have clarified the text as follows:

“The GDR approach is based on principles of capacity and responsibility (Baer et al., 2008) that are covered in the CAP and CPC approaches, respectively.” (Line 148)

163-166: Is the sensitivity analysis for this somewhere in the documentation of older papers? I’m interested in what the spread is for the median CERP under different BAU projections vs the spread of the outcomes under the range of effort sharing principles (outlined here for example).

An initial sensitivity analysis across different parameters and data assumptions (including three BaU projections: SSP2, SSP5 and SSP3) is available on a poster online (Robiou du Pont et al., 2015) and in my recently published PhD thesis (<https://minerva-access.unimelb.edu.au/handle/11343/213527>).

174-176: That's astonishing. Can this be reiterated in simpler terms as a take home message like: "In other words, the NDCs of Canada, Russia and China are much less than what would be required of them if they had the least stringent / most beneficial share of any global emissions scenario we analyzed." Also what was that exactly? The 5.1degC trajectory? Or RCP 8.5? Or the least stringent trajectory to achieve the Paris Agreement (e.g. 66% chance of 2degC)?

Implementing the suggestion of the author could be misleading. The hybrid allocation of the 5.1°C scenario does not correspond to the least stringent share of it. It corresponds to the least stringent share of the corresponding hypothetical lower scenario. To avoid confusion, we will keep the current wording.

Just to confirm again (and I won't share these findings yet): So does this mean that if all countries pledged analogous NDCs to Canada, China or Russia, warming would be equal to or in excess of 5.1 degrees? This is a pretty powerful message. Once again, I really like the tidiness and how informative this single number metric is for describing a country's ambition. Figure 3 is excellent, a lot of information in the panel a, but worth including, the choropleth in panel b is a great figure for picking up the essentials at a glance.

Thank you very much for your appreciation.

181-182: Thanks for including this note. Might have mentioned this before. Important to highlight how the picture here would change if including LULUCF.

The rationale for excluding LULUCF emissions is that emissions of the LULUCF sector are not considered by all parties as part of the emissions scope to be negotiated. Moreover, no universal accounting method of positive or negative LULUCF emissions is currently in place. Therefore, we exclude LULUCF emissions from the global scenarios before allocating their emissions across countries

199-201: Although I'm not a member of the CERP research group, I will speak on their behalf a bit more: it has been explained to me that the way PEC uses the GDR metric fundamentally changes the outcome, due to the choice of NAU projections used mostly I think. Not sure about all the intricacies here behind the scenes, but I do know that the outcome of CERP for these countries (developed European and Canada etc.) are NOT favoured from their work but somehow that switches under the PEC methodology. In this sense, calling your it 'GDR' here may be misnomer? You seem to take the overarching manipulation but change the fundamental inputs and achieve entirely different results (hence the sensitivity to inputs you mention above and indeed that outcome can be greater than changing the effort sharing principle itself). Perhaps to reconcile this you could add a line in the methods if not done already saying you use the method from GDR but achieve a fundamentally different result and call it something else like the 'Right to Development (RTD)' principle?" This way we could distinguish between the differing methodologies between the PEC and CERP projects. Now there is a risk that readers will conflate the two. I know this has happened already with the PEC website.

Thanks for the suggestion. We have followed the modelling of the GDR (Baer et al., 2008) and have therefore preserved the name. The GDR refers to a specific modelling, not a set of

results, which should be updated as new datasets (population, GDP, Gini index, BaU...) are released. We have used different BaU projections (the commonly accepted RCP8.5) and do not think this is as a reason to change the name. Furthermore, other studies have since used the GDR name under various parameterisation (Kemp-Benedict, 2010; Mattoo & Subramanian, 2012; Meinshausen et al., 2015; Robiou du Pont et al., 2017, 2016). We recommend that further work should investigate the sensitivity of the GDR/CERP approach to these parameters.

204: Once again, I don't think that countries really do 'pledge based on their own understanding of fairness' — this seems like wishful thinking. Could this be rephrased to say that countries usually pledge in a self-interested fashion, where they try to get away with doing as little as possible without looking worse than their peers, then justify their pledge with some contorted equity argument post hoc? Of course, change the language to not be so polemical. This is my take on international negotiations, I think it's better than speculation. Not sure if those references also support this take. Though I can't see how it happens the other way around. By citing work, you're saying this isn't your speculative analysis but even though I am not familiar with the literature to support my argument, I haven't read anything to support the argument as it stands presently. Maybe I am being overly pedantic about the language used, but it means something entirely different to me. Will look more into this later on my own time.

As we have stated earlier, some countries do clearly take in account concepts of fairness to shape their commitment. However, the mismatch between current NDCs and the global goals shows that this is not systematic. We agree with the reviewer's comments and would like to point out that the sentence carefully states that "countries **supposedly** build their pledge based on their own understanding of fairness, often self-interested".

We do not wish to include a statement that countries try to get away with doing as little as possible. Being polemical is not an issue, but we simply try to expose the NDC process and not to judge the intentions of countries.

240-241: Is there a physical reasons for using this or just common practice?

We chose a third degree fit to represent the relationship between 2030 emissions and temperature. Using a lower degree fit would lose information on the dynamics, while using a higher degree would result in a non-strictly increasing and thus non-bijective relationship.

300-301: not sure what you mean by this? The CPC was giving the wrong sign before? What did you do exactly? I thought is just shares the annual emissions trajectory, starting at e.g. 1990, equally per capita between countries.

The CPC approach was initially designed and only used to allocate the emissions of ambitious mitigation scenarios resulting in negative emissions by the end of the century (Robiou du Pont et al., 2017, 2016). In that case, countries without negative emissions reach zero emissions in 2100, and the other countries shared the negative emissions necessary to fit to the underlying IAM scenario.

In this study we apply the CPC to unambitious or even BaU scenarios with positive global emissions in 2100. Following the modelling described above would result in positive emissions in 2100 for countries that had previously negative emissions. Instead, with the modelling chosen here, countries do not have oscillating emissions scenarios and either increase or decrease monotonously.

349-350: because of impact at 2+ degrees?

Because of impacts, and the fact that BaU implies not mitigation effort (and thus no effort to be shared).

354: typo: change 'ta' to 'the'

Thank you very much. This has been corrected.

368: 'sufficiencatrian': never seen this word before, coined in this paper? Could you explain it or use another word?

This word comes from the study quoted in the text (Arneson, 2013) and that states: "The doctrine of sufficiency holds that it is morally valuable that as many as possible of all who shall ever live should enjoy conditions of life that place them above the threshold that marks the minimum required for a decent (good enough) quality of life".

For clarity, we modified the text to:

"Other approaches of distributive justice (for example based on sufficiency (Arneson, 2013), or using the Human Development Index) and other metrics (for example accounting for consumption-based emissions or exported emissions), not currently used in the IPCC report or under the UNFCCC, are not modelled here but would bring useful perspectives that could be integrated in the 'hybrid' approach." (Line 403)

381-382: once again not to nitpick, but I would refrain from saying "supported implicitly" even, since you either do support equity or you don't... It is clear that countries who have NDCs that fall in the CER or worse range and hence want to perpetuate the status quo or even a graver injustice of using more than their already unfair share, patently do not support equitable outcomes. Maybe this is a matter of option, but using the term 'equity' implies for me, some modicum of equity. While in the sense the word is being used here, it is totally ambiguous, like how the word 'quality' is satirized in the comic strip 'Dilbert' — If you use it in isolation, like saying we produce 'quality goods', you aren't really saying anything. They could be of terrible quality, but it still is an effective marketing play.

We agree with the reviewer that NDCs in line with a status-quo approach may reflect a support a preference for a status-quo. However, in the absence of a clear statement from a country representative, we cannot state that countries official supported the status quo (especially since they adhered to the UNFCCC and thus the CBDR-RC principle). We also agree with the reviewer that the term 'equity approach' does not apply to the CER, which is only a burden sharing approach. We have corrected this sentence, and checked throughout the text:

"The choice of the five effort-sharing approaches includes the 'grandfathering' approach (CER) that is only implicitly supported by some countries through their pledges." (Line 430)

410: so does this mean that the warming actually would be higher or have you composted already for LULUCF?

The warming is calculated with the inclusion of LULUCF emissions, bunker emissions and all the necessary data to calculate a temperature response. These emissions are simply not allocated/distributed across countries.

434-435: So LULUCF CO2 is included? But not non-CO2 gases?

We add LULUCF CO2 emissions directly from the IAM scenarios. The non-CO2 emissions, which by the way mostly occur in the agricultural sector with only a minor fraction in the LULUCF category, are included in the national emissions that we subject to the allocation approaches. Our final pathways include all the emissions derived by the underlying IAM scenarios, and the warming assessment is thus consistent with those of IAM scenarios presented in the literature.

Closing thoughts:

I would advocate for Figure 1 being redone with either CAP, CPC, and EPC, only, or the three base ones plus GDR, i.e. excluding CER.

We understand the concern with the inclusion of the CER in any recommendation or metric for NDCs. We agree and the CER is absent from the metric and suggested national emissions target. That is the case, and the CER is not used in any of the national results in the text and in Figure 2 and 3.

However, we believe that it is important to keep the CER in Figure 1 precisely to warn against the dangerous consequences of pursuing a situation where countries implicitly follow such status quo. The CER approach is used in Figure 1, which offers a modelling of a self-interested situation and highlights the difference between the continuation of this situation and the global climate objectives. The presence of the CER and the results displayed in this figure reflects the importance it had in shaping the first round of NDCs and warns against a continuation of its utilisation. Current aggregated NDCs align with such a self-interested situation when status quo is considered as an option for countries. Prolongating such a situation would lead to warming inconsistent with the Paris Agreement. This is an important finding. Figure 1 only present results at the global scale and does not present national allocations.

I think a more plain-language summary of what you do in the second part of your analysis (depicted results in figure 3) would be nice for the lay reader. Something with the words, "warming resulting if all countries acted in an analogous manner", could go in the abstract.

Thanks for this suggestion. We have gone through the text again to ensure clarity.

I think your approach, rather than be used as a hypothetical 'agree to disagree' scenario, could be instead or also be framed as a predictive tool / thought experiment for what would happen if all countries 1. Pick the least stringent equity setting allowed, and 2. Act all like a given county. This for me sums up the paper entirely. The results of this are a novel and significant contribution to the climate equity literature. I look forward to seeing this in print soon.

Thanks for the contribution and time spent on the revisions.

References included in the response to the reviewer

- Althor, G., Watson, J. E. M., & Fuller, R. A. (2016). Global mismatch between greenhouse gas emissions and the burden of climate change. *Scientific Reports*, *6*, 1–6. <https://doi.org/10.1038/srep20281>
- Arneson, R. (2013). Egalitarianism. Retrieved January 1, 2017, from <https://plato.stanford.edu/entries/egalitarianism/>
- Baer, P., Fieldman, G., Athanasiou, T., & Kartha, S. (2008). Greenhouse Development Rights: towards an equitable framework for global climate policy. *Cambridge Review of International Affairs*, *21*(August 2014), 649–669. <https://doi.org/10.1080/09557570802453050>
- Kemp-Benedict, E. (2010). *Calculations for the Greenhouse Development Rights Calculator*. Somerville, MA, USA. Retrieved from <https://www.sei-international.org/mediamanager/documents/Publications/Climate/calculations-gdr-calculator.pdf>
- Klinsky, S., Roberts, T., Huq, S., Okereke, C., Newell, P., Dauvergne, P., ... Bauer, S. (2017). Why equity is fundamental in climate change policy research. *Global Environmental Change*, *44*, 170–173. <https://doi.org/10.1016/j.gloenvcha.2016.08.002>
- Mattoo, A., & Subramanian, A. (2012). Equity in Climate Change : An Analytical Review. *World Development*, *40*(6), 1083–1097. <https://doi.org/10.1016/j.worlddev.2011.11.007>
- Meinshausen, M., Jeffery, L., Guetschow, J., Robiou Du Pont, Y., Rogelj, J., Schaeffer, M., ... Meinshausen, N. (2015). National post-2020 greenhouse gas targets and diversity-aware leadership. *Nature Climate Change*, *5*(12). <https://doi.org/10.1038/nclimate2826>
- Peters, G. P., Andrew, R. M., Solomon, S., & Friedlingstein, P. (2015). Measuring a fair and ambitious climate agreement using cumulative emissions. *Environmental Research Letters*, *10*(10), 105004. <https://doi.org/10.1088/1748-9326/10/10/105004>
- Robiou du Pont, Y. (2018). *Climate justice: can we agree to disagree? Operationalising competing equity principles to mitigate global warming*. The University of Melbourne. Retrieved from <https://minerva-access.unimelb.edu.au/handle/11343/213527>
- Robiou du Pont, Y., & Jeffery, M. L. (2017). Paris-Equity-Check.org - How fair are countries' climate pledges? Retrieved January 1, 2017, from <http://paris-equity-check.org/>
- Robiou du Pont, Y., Jeffery, M. L., Gütschow, J., Christoff, P., & Meinshausen, M. (2015). Quantifying equitable allocations of a 2 ° C consistent emissions pathway. In *Our Common Future Under Climate Change* (p. 1). Paris, France. <https://doi.org/10.13140/RG.2.2.22758.63049>
- Robiou du Pont, Y., Jeffery, M. L., Gütschow, J., Christoff, P., & Meinshausen, M. (2016). National contributions for decarbonizing the world economy in line with the G7 agreement. *Environmental Research Letters*, *11*(5), 054005. <https://doi.org/10.1088/1748-9326/11/5/054005>
- Robiou du Pont, Y., Jeffery, M. L., Gütschow, J., Rogelj, J., Christoff, P., & Meinshausen, M. (2017). Equitable mitigation to achieve the Paris Agreement goals. *Nature Climate Change*, *7*(January), 38–43. <https://doi.org/10.1038/NCLIMATE3186>
- Robiou Du Pont, Y., Jeffery, M. L., Gütschow, J., Rogelj, J., Christoff, P., & Meinshausen, M. (2017). Equitable mitigation to achieve the Paris Agreement goals. *Nature Climate Change*, *7*(1). <https://doi.org/10.1038/nclimate3186>

Sabin Center for Climate Change Law. (2018). Climate Case Chart. Retrieved from <http://climatecasechart.com/>

UNFCCC. (2015). *Adoption of the Paris Agreement. Report No. FCCC/CP/2015/L.9/Rev.1* (Vol. 21932). Paris, France. Retrieved from <http://unfccc.int/resource/docs/2015/cop21/eng/l09r01.pdf>

Urgenda. (2017). Urgenda Climate Case. Retrieved January 1, 2017, from <http://www.urgenda.nl/en/climate-case/>

Zenghelis, D. (2017). Equity and national mitigation. *Nature Climate Change*, 7(January), 9–10. <https://doi.org/10.1038/nclimate3192>

REVIEWERS' COMMENTS:

Reviewer #2 (Remarks to the Author):

General Comments (does not require response from author):

Overall, I am very happy with the revised and expanded version of this study and I would like to recommend the manuscript for immediate publication.

I am more than satisfied with the revised and expanded version of the paper, and am satisfied with the changes made to the manuscript. I thank the authors for their patience, tenacity and diligence. Their efforts have vastly improved the analysis of this study, as well as the clarity and cogency of their arguments and communication. I hope to see this study in print soon enough to be included in the upcoming international climate policy community discussion, specifically in the forthcoming Conference of the Parties, as it will greatly inform the conversation, and I hope that it will enlighten thinking on fairness and influence debate surrounding scale of ambition inherent in NDCs. I know it will make a valuable conversation to the broader discussion as well, and hope its takeaways are disseminated widely via the press and social media.

I would like to thank the author(s) for their extensive revisions at my request. I feel that the manuscript has improved in clarity and the study is now much stronger, both in terms of its attention to ethical nuances, political implications, and its context within the larger body of effort sharing literature. I require no formal responses to the above or following comments except one minor revision noted below.

Reproducibility (does not require response from author):

Although I do not model these kinds of experiments myself, I have a working knowledge of climate models and IAMs, and found that the description of the methodology is sufficiently detailed and clear to ensure reproducibility, if one should attempt it.

Novel contributions (does not require response from author):

The author's summary metric of benchmarking ambition according to temperature is an invaluable and novel contribution to the literature. The accompanying analysis ensures that this metric, as well as their 'hybrid' model for testing various equity ambition scenarios, provides sound and robust work that is thorough enough to be published in any high caliber journal. I feel that they have more than satisfied the criteria of this or any journal. Below are my final comments for authors to take into consideration. I require no response to any of the above comments.

Minor revision (requires response from author):

So does your 'Equal Per Capita' end up allotting equal cumulative emissions over the whole chosen time interval? If so, could you add a line to this effect? If not, please specify that cumulative emissions do not necessarily result in equal per capita shares, and would only do so by increasing ambition after convergence dates, so that emissions of countries who held emissions debts would be less than than credited with emissions allowances. I will refer to the cited literature, as the author(s) made reference to in their rebuttal, however it would ideal to have a few words in the main text of the manuscript that explain this method succinctly, so that the reader does not have to delve into the referenced literature to find out for themselves (in the interest of making the text self-contained for the benefit of a more general audience).

Best of luck and thanks again for your cooperation and hard work. I look forward to seeing this work in print.

Daniel Horen Greenford

REVIEWERS' COMMENTS:

Reviewer #2 (Remarks to the Author):

General Comments (does not require response from author):

Overall, I am very happy with the revised and expanded version of this study and I would like to recommend the manuscript for immediate publication.

I am more than satisfied with the revised and expanded version of the paper, and am satisfied with the changes made to the manuscript. I thank the authors for their patience, tenacity and diligence. Their efforts have vastly improved the analysis of this study, as well as the clarity and cogency of their arguments and communication. I hope to see this study in print soon enough to be included in the upcoming international climate policy community discussion, specifically in the forthcoming Conference of the Parties, as it will greatly inform the conversation, and I hope that it will enlighten thinking on fairness and influence debate surrounding scale of ambition inherent in NDCs. I know it will make a valuable conversation to the broader discussion as well, and hope its takeaways are disseminated widely via the press and social media.

I would like to thank the author(s) for their extensive revisions at my request. I feel that the manuscript has improved in clarity and the study is now much stronger, both in terms of its attention to ethical nuances, political implications, and its context within the larger body of effort sharing literature. I require no formal responses to the above or following comments except one minor revision noted below.

Reproducibility (does not require response from author):

Although I do not model these kinds of experiments myself, I have a working knowledge of climate models and IAMs, and found that the description of the methodology is sufficiently detailed and clear to ensure reproducibility, if one should attempt it.

Novel contributions (does not require response from author):

The author's summary metric of benchmarking ambition according to temperature is an invaluable and novel contribution to the literature. The accompanying analysis ensures that this metric, as well as their 'hybrid' model for testing various equity ambition scenarios, provides sound and robust work that is thorough enough to be published in any high caliber journal. I feel that they have more than satisfied the criteria of this or any journal. Below are my final comments for authors to take into consideration. I require no response to any of the above comments.

Minor revision (requires response from author):

So does your 'Equal Per Capita' end up allotting equal cumulative emissions over the whole chosen time interval? If so, could you add a line to this effect? If not, please specify that cumulative emissions do not necessarily result in equal per capita shares, and would only do so by increasing ambition after convergence dates, so that emissions of countries who held emissions debts would be less than than credited with emissions allowances. I will refer to the cited literature, as the author(s) made reference to in their rebuttal, however it would ideal to

have a few words in the main text of the manuscript that explain this method succinctly, so that the reader does not have to delve into the referenced literature to find out for themselves (in the interest of making the text self-contained for the benefit of a more general audience).

Best of luck and thanks again for your cooperation and hard work. I look forward to seeing this work in print.

Daniel Horen Greenford

Deep thanks to all the reviewers for the time spent improving this study, and to Daniel Horen Greenford for his encouragements.

Regarding the reviewers' request to provide more details on the modelling of the dynamic EPC approach, and its difference with the equal cumulative per capita (CPC) approach we have amended the main text as follows:

“These five categories include notions of: capability to pay (CAP approach), equality with the dynamic Equal Per Capita (EPC) approach, responsibility-capability-need with the Greenhouse Development Rights (GDR), historical responsibility with the Equal Cumulative Per Capita (CPC), and national circumstances regarding current emissions levels with the 'grandfathering' approach (also named 'constant-emissions ratio' or CER).” (Line 106)

We also added to the methods:

“The EPC dynamically shares the emissions of the global scenario across countries based on their projected population trajectories and thus does not result in equal cumulative per capita emissions (that is equal cumulative emissions over cumulative populations). Comparing two countries with equal given cumulative population, a country with increasing shares of the global population will have lower allocations under decreasing global emissions scenarios, and higher allocations under increasing global emissions scenarios, than a country with decreasing shares of the global population.” (Line 307)

Yann Robiou du Pont, corresponding author, and Malte Meinshausen.